# STABILITY UNDER SCRUTINY: BENCHMARKING REPRESENTATION PARADIGMS FOR ONLINE HD MAPPING

**Hao Shan**[1,2*], **Ruikai Li**[1,2*], **Han Jiang**[1,2†], **Yizhe Fan**[1,2], **Ziyang Yan**[1,2], **Bohan Li**[3,4],
**Xiaoshuai Hao**[5‡], **Hao Zhao**[5,6], **Zhiyong Cui**[1,2], **Yilong Ren**[1,2], **Haiyang Yu**[1,2]
[1] State Key Lab of Intelligent Transportation System
[2] School of Transportation Science and Engineering, Beihang University
[3] Shanghai Jiao Tong University
[4] Ningbo Institute of Digital Twin, Eastern Institute of Technology
[5] Beijing Academy of Artificial Intelligence (BAAI)
[6] Institute for AI Industry Research (AIR), Tsinghua University
[*]Equal contribution        [†]Corresponding author        [‡]Project leader

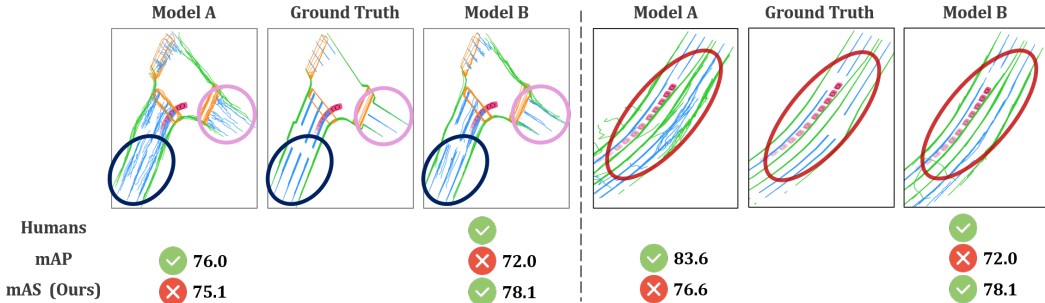

Figure 1: **Evaluating trustworthiness of online mapping models using human judgment, traditional mAP, and our mAS metric.** In each case, the standard accuracy metric (mAP) fails to align with human judgment because it evaluates only single-frame precision, disregarding stability across time. To address this limitation, we propose the first stability benchmark for online vectorized map construction and present a large-scale analysis of contemporary models.

## ABSTRACT

As one of the fundamental modules in autonomous driving, online high-definition (HD) maps have attracted significant attention due to their cost-effectiveness and real-time capabilities. Since vehicles always cruise in highly dynamic environments, spatial displacement of onboard sensors inevitably causes shifts in real-time HD mapping results, and such instability poses fundamental challenges for downstream tasks. However, existing online map construction models tend to prioritize improving each frame's mapping accuracy, while the mapping stability has not yet been systematically studied. To fill this gap, this paper presents the first comprehensive benchmark for evaluating the temporal stability of online HD mapping models. We propose a multi-dimensional stability evaluation framework with novel metrics for Presence, Localization, and Shape Stability, integrated into a unified mean Average Stability (mAS) score. Extensive experiments on 42 models and variants show that accuracy (mAP) and stability (mAS) represent largely independent performance dimensions. We further analyze the impact of key model design choices on both criteria, identifying architectural and training factors that contribute to high accuracy, high stability, or both. To encourage broader focus on stability, we will release a public benchmark. Our work highlights the importance of treating temporal stability as a core evaluation criterion alongside accuracy, advancing the development of more reliable autonomous driving systems. The benchmark toolkit, code, and models will be available at https://stablehdmap.github.io/.

# 1 INTRODUCTION

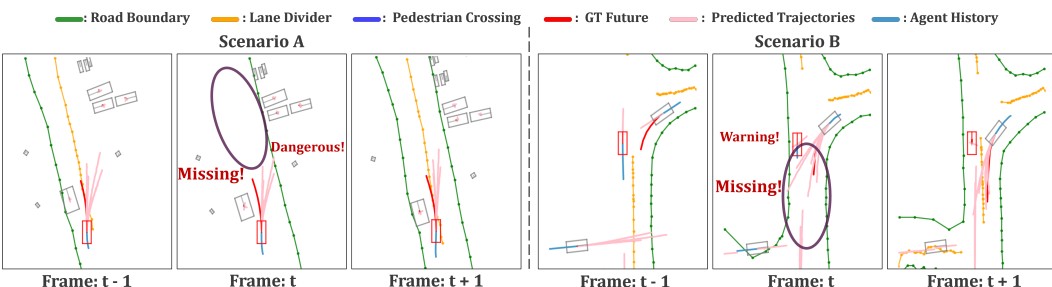

Figure 2: **The Impact of Unstable Map Elements on Downstream Tasks.** In Scenario A, the ego vehicle attempts to overtake, but the forward lane divider suddenly disappears during the maneuver, causing the ego vehicle to steer toward the curb. In Scenario B, another vehicle attempts to change lanes, but due to flickering lane dividers in the ego vehicle's perception, the ego vehicle interprets the other vehicle's action as a collision course.

High-definition (HD) map is one of the fundamental component of autonomous driving, offering centimeter-level environmental details such as precise coordinates of map elements and vectorized topological structures (Hu et al., 2023; Jiang et al., 2023; Liao et al., 2025a; Cao et al., 2025; Wei et al., 2026; Hao et al., 2025a). Although traditional pre-built HD map provides highly accurate representations, its substantial production and maintenance costs, coupled with limited adaptability to dynamic road conditions, severely restrict large-scale deployment. To address these limitations, online HD mapping has recently emerged as a promising alternative (Li et al., 2022; Liao et al., 2022). By leveraging onboard sensors to perceive the environment in real time, this approach dynamically constructs local vectorized maps, thereby reducing dependence on offline HD maps and paving the way toward scalable and generalizable autonomous driving systems.

Recent advances in online mapping have primarily aimed at improving accuracy and efficiency, giving rise to a diverse set of approaches with distinct representational paradigms (Lilja et al., 2024; Liao et al., 2022).The community typically evaluates these methods using metrics such as mean Average Precision (mAP) on benchmark datasets, which has driven continuous improvements in state-of-the-art performance. However, a critical yet underexplored issue in traditional evaluation is the stability of model outputs, a property essential for the safe deployment of autonomous driving systems, as illustrated in Fig.2. A model that achieves high average precision but produces flickering map boundaries or fails entirely at complex intersections, acting like an "intermittently blind" guide, poses substantial safety risks (Gu et al., 2024; Zhang et al., 2025; Wan et al., 2025b; Zeng et al., 2025b). Despite its importance, the field currently lacks dedicated benchmarks and metrics to quantitatively assess stability in online HD mapping. This gap hinders systematic evaluation of how different representational paradigms respond to real-world disturbances, ultimately slowing progress toward more reliable next-generation mapping systems.

To bridge this gap, we present the first systematic investigation and benchmark for stability in online HD mapping, under the theme "Beyond Accuracy: Under Scrutiny of Stability". Our key contributions are threefold:

- **A multi-dimensional stability evaluation framework.** We propose novel temporal stability metrics, including Presence, Localization, and Shape Stability, to quantitatively capture the consistency of map elements across consecutive frames. These are integrated into a comprehensive mean Average Stability (mAS) score, enabling holistic model assessment.

- **Comprehensive benchmarking and analysis.** We conduct large-scale experiments across diverse state-of-the-art models, revealing that accuracy (mAP) and stability (mAS) are largely independent performance dimensions. Our analysis examines how design choices in sensors, 2D backbones, BEV encoders, temporal fusion, and training regimens influence accuracy and stability as distinct evaluation aspects.

- **The first stability centric benchmark.** We establish and will release a public benchmark to catalyze community-wide focus on stability, providing the foundation for developing safer and more robust online mapping systems.

## 2 RELATED WORK

**Online HD Mapping Models.** Online HD mapping has become a critical and extensively studied subtask in autonomous driving (Xie et al., 2025a; Zeng et al., 2024). Depending on the choice of sensor input, existing methods can be broadly categorized into camera-only (Qiao et al., 2023; Ding et al., 2023; Zhang et al., 2023; Liu et al., 2024a;b), LiDAR-only (Wang et al., 2023), and camera–LiDAR fusion (Li et al., 2022; Liu et al., 2023; Liao et al., 2022; 2025b; Yuan et al., 2024; Zhang et al., 2024c) paradigms, each offering distinct strengths and weaknesses in perception capability and environmental adaptability (Hao et al., 2024a; Kim et al., 2025; Yan et al., 2025; Li et al., 2025c; Kong et al., 2025). Generative frameworks have also been explored for large-scale map construction from multi-modal data (Yuan et al., 2025). Persistent autoregressive mapping with traffic rules has also been explored (Liang et al., 2025). Online HD-SD map association frameworks and benchmarks have also been explored (Wan et al., 2025a). Recent advances in point cloud Transformers further provide strong 3D feature extractors that can benefit LiDAR-based perception (Wan et al., 2025c; Zhang et al., 2024a). While these paradigms have driven notable progress in mapping accuracy and efficiency, current evaluation frameworks remain narrowly focused on mean Average Precision (mAP), overlooking the critical dimension of stability. This omission substantially limits the practical reliability and deployment of online mapping systems in downstream driving tasks.

**Robustness in Autonomous Driving.** Robustness to real-world perturbations has been extensively explored in core autonomous driving tasks. Established benchmarks exist for 2D (Wang et al., 2020) or 3D detection (Dong et al., 2023; Zhu et al., 2023; Paek et al., 2022), segmentation (Hong et al., 2022), depth estimation (Kong et al., 2023), and vision-language navigation (Zeng et al., 2025c), where models are evaluated under conditions such as corruption, adverse weather, and occlusion. More recently, RoboBEV has extended to Bird's-Eye-View (BEV) perception, revealing vulnerabilities in view transformation techniques such as LSS and transformers (Xie et al., 2023; 2025b). In the context of online HD mapping, early efforts have examined sensor level robustness, demonstrating that mapping systems are highly sensitive to corrupted inputs (Hao et al., 2024b; 2025c;b). However, these studies are restricted to static, single frame analyses and sensor-specific faults. Crucially, the temporal stability of mapping models under sequential perturbations and the comparative robustness of different representation paradigms remain unexplored, a gap which our benchmark aims to address.

**Evaluation Metrics for Online HD Mapping.** Current evaluation metrics in the field of online HD map construction are often designed based on single frame geometric accuracy (Li et al., 2022; Liao et al., 2022), primarily focusing on the geometric similarity between the predicted map and ground truth in a given frame. Among typical existing metrics, mean Intersection over Union (mIoU) measures the spatial overlap between the predicted map and the ground truth, while mean Average Precision (mAP) comprehensively considers both classification accuracy and the localization precision of map elements. However, a critical yet previously overlooked issue is that the impact of online mapping on downstream planning tasks depends not only on per-frame geometric accuracy, but also on the inter-frame dynamic stability of the vectorized map. Jitter in map elements across frames can significantly impair the decision-making of autonomous driving systems (Zhang et al., 2025; Gu et al., 2024; Jiang et al., 2023). More seriously, existing metrics completely ignore the temporal geometric stability of map elements, such as the magnitude of polyline edge jitter and the frequency of shape mutations, which are crucial safety factors. To the best of our knowledge, our work is the first to establish a publicly available benchmark dedicated to stability evaluation for online mapping.

## 3 MULTI-DIMENSIONAL MAP STABILITY EVALUATION FRAMEWORK

This section details the proposed framework for multi-dimensional stability evaluation in online HD mapping. The framework quantifies temporal stability through instance-level matching across consecutive frames, specifically designed to assess three critical dimensions: detection consistency, geometric jitter, and shape preservation. The entire pipeline, consists of four main stages: (1) temporal sampling of frame pairs, (2) cross-frame instance matching, (3) geometric alignment and resampling, and (4) stability metric computation.

## 3.1 TEMPORAL SAMPLING

The temporal sampling stage constructs pairs of frames for analyzing stability over varying time intervals. Given a sequence of $L$ consecutive model output frames $\{D_1, D_2, \ldots, D_L\}$ and a pre-defined maximum temporal interval $M$, the process is as follows: for each anchor frame $D_t$ (where $t \leq L - M$), a subsequent frame $D_{t+k}$ is randomly sampled from the future window $\{D_{t+1}, \ldots, D_{t+M}\}$, forming an evaluation pair $(D_t, D_{t+k})$. Repeating this procedure for every valid anchor frame $t$ results in a comprehensive sample set $S$ of size $|S| = L - M$, which provides the foundational inputs for subsequent stability analysis. Testing multiple $M$ values ($M \in \{2, 3, 5, 10\}$) allows our framework to adapt to different application scenarios. Detailed analysis of temporal sampling design and frame rate considerations is provided in Appendix C.4.3.

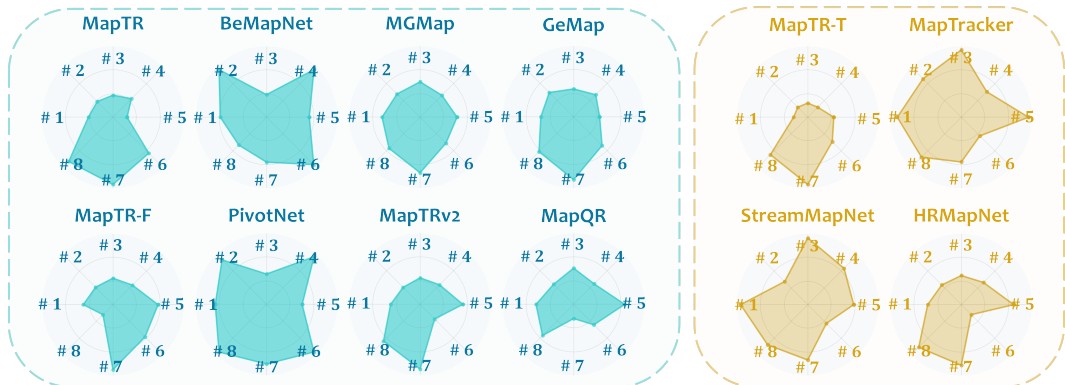

Figure 3: **Radar chart for Basic HD map constructors covering eight evaluation metrics.** The axes of the radar chart correspond to: #1 mAS, #2 Shape, #3 Loc, #4 Presence, #5 mAP, #6 Inference Memory Cost, #7 Parameter Count, #8 FPS.

## 3.2 CROSS-FRAME INSTANCE MATCHING

Establishing accurate correspondence between map elements across temporal frames is essential for stability assessment. Given the inherent inconsistencies in model predictions, a direct matching approach is prone to error. Instead, a robust indirect strategy is utilized, leveraging the consistent annotations of ground truth (GT) data as a reliable intermediary for association.

For each frame pair $(D_t, D_{t+k})$, the matching process comprises two steps:

1. **Frame-to-GT Matching:** Predictions in each frame are independently matched to their respective GT instances using the Hungarian algorithm, which optimizes a cost function based on geometric and semantic similarity.

2. **GT-based Association:** The persistent identification of GT elements across frames enables the linkage of corresponding predictions. Specifically, predictions matched to the same GT instance in different frames are paired, thereby transferring the temporal consistency of the GT to the model outputs.

This procedure yields a set of matched instance pairs $\{(\text{poly}_{t+k}(e), \text{poly}_t(e)) \mid e \in E\}$ for each frame pair, where $E$ represents the set of successfully tracked map elements. The complete algorithmic details are provided in Algorithm 2 of the Appendix. GT serves as a matching medium rather than an absolute geometric benchmark. Minor GT annotation jitter does not affect stability assessment, as matching is based on persistent GT instance IDs. Detailed discussion on GT matching robustness and alignment noise handling is provided in Appendix B.

## 3.3 GEOMETRIC ALIGNMENT AND RESAMPLING

Geometric alignment ensures a fair and spatially consistent comparison between matched polylines $(\text{poly}_{t+k}(e), \text{poly}_t(e))$ by transforming them into a common coordinate system and resampling them uniformly. This process consists of three sequential operations.

**Coordinate Transformation.** The historical polyline $\text{poly}_t(e)$ is first transformed from the ego coordinate system of its original frame $D_t$ into the ego coordinate system of the current frame $D_{t+k}$. This spatial normalization is computed as:

$$\text{poly}_{t \to t+k}(e) = T_{\text{world} \to t+k} \cdot T_{t \to \text{world}} \cdot \text{poly}_t(e),$$

where $T_{t \to \text{world}}$ and $T_{\text{world} \to t+k}$ denote the transformation matrices from frame $D_t$ to the world frame and from the world frame to frame $D_{t+k}$, respectively.

**Perception Range Filtering.** The transformed polyline $\text{poly}_{t \to t+k}(e)$ is then clipped to the perception range of the model in frame $D_{t+k}$. A point $p = (x, y)$ is retained for subsequent analysis if and only if it satisfies:

$$x_{\min} \leq x \leq x_{\max}, \quad \text{and} \quad y_{\min} \leq y \leq y_{\max},$$

where $[x_{\min}, x_{\max}, y_{\min}, y_{\max}]$ defines the operational perceptual boundaries, ensuring evaluation consistency with the model's design.

**Uniform Resampling.** Finally, to enable precise point-wise comparison, both the current polyline $\text{poly}_{t+k}(e)$ and the transformed historical polyline $\text{poly}_{t \to t+k}(e)$ are resampled uniformly. We employ a dynamic axis selection mechanism that adaptively determines the primary sampling axis based on local geometric orientation, rather than using a fixed axis. This ensures robust resampling for polylines of arbitrary orientations. Detailed implementation is provided in Appendix B.

## 3.4 STABILITY METRIC COMPUTATION

Based on the aligned and resampled point sets $\text{poly}_{t+k}^{\text{sample}}(e)$ and $\text{poly}_t^{\text{sample}}(e)$, the temporal stability of each matched map element $e$ is quantified from three perspectives.

**Presence Stability.** This metric evaluates the detection consistency of an element across frames. Let $\text{score}(e)$ denote the model's confidence score for element $e$ and $\tau$ be a detection threshold. The presence stability is defined as:

$$\text{Presence}(e) = \begin{cases} 1, & \text{if score}_{t+k}(e) \geq \tau \text{ and score}_t(e) \geq \tau, \\ & \text{or score}_{t+k}(e) < \tau \text{ and score}_t(e) < \tau; \\ 0.5, & \text{otherwise (flickering).} \end{cases}$$

A higher average value across instances indicates better detection consistency.

**Localization Stability.** This metric quantifies the point-wise positional jitter of an element. For the resampled polylines, we compute the average $L1$ distance in the $y$-coordinate and map it to a stability score:

$$\text{Loc}(e) = 1 - \frac{1}{\beta} \cdot \frac{1}{N} \sum_{i=1}^{N} |y_{t+k}(x_i) - y_t(x_i)|,$$

where $\beta$ is a scaling parameter. The selection of $\beta = 15$ corresponds to the map's short-range radius, representing the distance threshold for complete instability. Ablation studies and detailed justification are provided in Appendix C. The formula maps the average deviation to a score between 0 (unstable) and 1 (stable).

**Shape Stability.** This metric assesses the consistency of an element's geometric shape by comparing the curvature of the resampled polylines. We approximate the curvature $\kappa$ of a polyline as the average angle between consecutive segments:

$$\kappa(\text{poly}) = \frac{1}{N-1} \sum_{j=1}^{N-1} \theta_j, \quad \text{where} \quad \theta_j = \cos^{-1}\left( \frac{\vec{v_j} \cdot \vec{v_{j+1}}}{|\vec{v_j}| \cdot |\vec{v_{j+1}}|} \right).$$

The shape stability is then defined as the normalized difference in curvature:

$$\text{Shape}(e) = 1 - \frac{\left| \kappa(\text{poly}_{t+k}^{\text{sample}}(e)) - \kappa(\text{poly}_t^{\text{sample}}(e)) \right|}{\pi}.$$

**Comprehensive Stability Index.** The overall stability for a single instance $e$ is computed by combining the three metrics:

$$\text{Stability}(e) = \text{Presence}(e) \cdot [\omega \cdot \text{Loc}(e) + (1 - \omega) \cdot \text{Shape}(e)],$$

where $\omega \in [0, 1]$ is a weighting parameter (default: 0.7). The class-wise stability is the average over all instances of that class:

$$\text{Stability}_{\text{class}} = \frac{1}{|\mathcal{I}_{\text{class}}|} \sum_{e \in \mathcal{I}_{\text{class}}} \text{Stability}(e).$$

Finally, the overall model stability, **mean Average Stability (mAS)**, is the mean of the stability scores across all classes:

$$\text{mAS} = \frac{1}{|\mathcal{C}|} \sum_{\text{class} \in \mathcal{C}} \text{Stability}_{\text{class}}.$$

This single score provides a holistic measure of a model's temporal stability.

## 4 EXPERIMENTAL ANALYSIS

In this section, we present a comprehensive empirical evaluation of our proposed stability assessment framework. Our experiments are designed to answer the following key questions:

- **RQ1:** How do state-of-the-art online HD mapping models perform in terms of both conventional accuracy (mAP) and our newly proposed temporal stability (mAS)? Is there an implicit correlation between them?
- **RQ2:** How do different representational paradigms influence model stability?
- **RQ3:** What are the specific strengths and weaknesses of each paradigm under temporal scrutiny, as revealed by our fine-grained stability metrics (Presence, Localization, Shape)?

Table 1: **Basic Benchmarking of HD Map Constructors.** Performance comparison of online HD mapping methods on nuScenes val set. Models grouped by temporal fusion mechanisms, input modality, BEV encoder and training epochs. "Temp" denotes the injection of temporal information. "L" and "C" represent LiDAR and camera respectively, while the 2D and 3D backbones employ ResNet50 (He et al., 2016) and SECOND (Yan et al., 2018), correspondingly.

| Method | Venue | Temp | Modal | BEV Encoder | Epoch | mAP↑ | Presence↑ | Loc↑ | Shape↑ | mAS↑ |
|---|---|---|---|---|---|---|---|---|---|---|
| MapTR (Liao et al., 2022) | ICLR'23 | ✗ | C | GKT | 24 | 44.1 | 91.2 | 65.4 | 90.6 | 71.6 |
| MapTR (Liao et al., 2022) | ICLR'23 | ✗ | C & L | GKT | 24 | 62.8 | 91.5 | 68.6 | 91.0 | 74.0 |
| BeMapNet (Qiao et al., 2023) | CVPR'23 | ✗ | C | IPM-PE | 30 | 61.4 | 100.0 | 65.8 | 97.9 | 81.9 |
| PivotNet (Ding et al., 2023) | ICCV'23 | ✗ | C | PersFormer | 30 | 57.1 | 100.0 | 71.4 | 97.2 | 84.3 |
| MapTRv2 (Liao et al., 2025b) | IJCV'24 | ✗ | C | BEVPool | 24 | 61.4 | 91.5 | 68.6 | 90.9 | 73.9 |
| GeMap (Zhang et al., 2024c) | ECCV'24 | ✗ | C | BEVFormer-1 | 24 | 51.3 | 92.3 | 69.7 | 92.6 | 75.5 |
| MGMap (Liu et al., 2024a) | CVPR'24 | ✗ | C | BEVFormer-1 | 24 | 57.9 | 92.2 | 75.0 | 92.3 | 78.0 |
| MapQR (Liu et al., 2024b) | ECCV'24 | ✗ | C | BEVFormer-3 | 24 | 66.4 | 91.8 | 75.6 | 91.6 | 77.8 |
| MapTR (Liao et al., 2022) | ICLR'23 | ✓ | C | GKT | 24 | 51.3 | 88.61 | 59.7 | 89.3 | 66.6 |
| StreamMapNet (Yuan et al., 2024) | WACV'24 | ✓ | C | BEVFormer-1 | 30 | 63.3 | 96.6 | 97.7 | 92.3 | 91.9 |
| MapTracker (Chen et al., 2024) | ECCV'24 | ✓ | C | BEVFormer-2 | 72 | 75.95 | 93.3 | 98.1 | 95.8 | 90.4 |
| HRMapNet (Zhang et al., 2024b) | ECCV'24 | ✓ | C | BEVFormer-1 | 24 | 67.2 | 92.3 | 70.5 | 91.5 | 75.9 |

### 4.1 BENCHMARK CONFIGURATION

**Benchmark and Models.** In this work, we conduct a comprehensive evaluation of **42** online HD map constructors and their variants, covering representative methods following diverse representation paradigms, including BeMapNet (Qiao et al., 2023), PivotNet (Ding et al., 2023), MapTR (Liao et al., 2022), MapTRv2 (Liao et al., 2025b), StreamMapNet (Yuan et al., 2024), MGMap (Liu et al., 2024a), GeMap (Zhang et al., 2024c), MapQR (Liu et al., 2024b), MapTracker (Chen et al., 2024), and HRMapNet (Zhang et al., 2024b). These models represent diverse design choices across input modalities, backbone architectures, BEV encoders, temporal fusion mechanisms, historical priors, and training epochs, allowing for a holistic analysis of representational paradigms. Model weights are sourced from official code repositories or retrained using default settings to ensure fairness. Unfortunately, due to the unavailability of source code for several online mapping approaches, we were unable to include them in our full assessment.

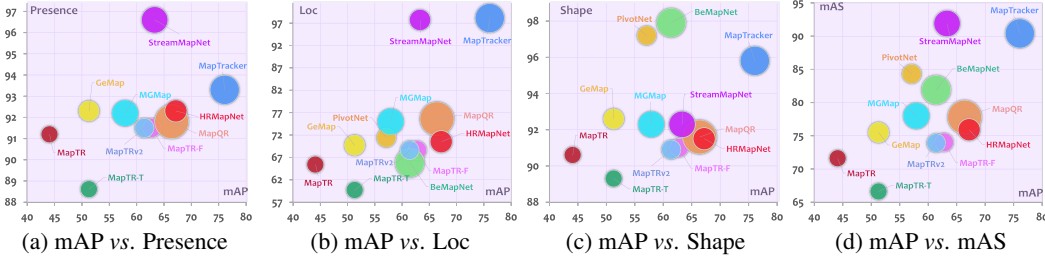

(a) mAP *vs.* Presence    (b) mAP *vs.* Loc    (c) mAP *vs.* Shape    (d) mAP *vs.* mAS

Figure 4: The correlations between the single-frame accuracy metrics mAP and the stability metrics Presence, Loc, Shape, and mAS. The bubble size represents the model's parameter count.

**Evaluation Metrics.** We evaluate each model using both the conventional mean Average Precision (mAP) and our proposed multi-dimensional stability metrics: Presence, Localization (Loc), Shape stability, and the comprehensive mean Average Stability (mAS), as defined in Section 3.4. Additional metrics related to inference performance have been incorporated into the evaluation framework, As shown in Fig.3 More detailed evaluation metrics are provided in Section D of the Appendix. We also evaluate model stability under adverse weather conditions and provide comprehensive ablation studies on hyperparameters. Results are detailed in Appendix C.

## 4.2 BASIC PERFORMANCE BENCHMARKING (RQ1)

The basic benchmarking results are summarized in Table 1. Our analysis reveals two key findings that challenge the sole reliance on accuracy for model evaluation:

**Stability constitutes a distinct and critical performance dimension.** A primary observation from our benchmark, as illustrated in Fig.4, is the imperfect correlation between conventional accuracy (mAP) and temporal stability (mAS). We observe that models with higher mAP do not necessarily achieve superior mAS, indicating that temporal stability is not an automatic byproduct of high accuracy but rather a unique aspect of model performance. This aspect is crucial for real-world deployment yet is overlooked by conventional metrics.

**Significant stability gaps exist among mainstream paradigms.** We observe that the mAS scores span a wide range from 71.6 (MapTR (Liao et al., 2022)) to 91.9 (StreamMapNet (Yuan et al., 2024)), indicating that the choice of representational paradigm profoundly impacts the consistency of the generated map. A majority of existing models, cluster in the lower to mid-range of mAS (71.6 - 78.0). This clustering suggests a common challenge faced by current approaches in maintaining output stability across consecutive frames.

## 4.3 IN-DEPTH ANALYSIS OF REPRESENTATIONAL PARADIGMS (RQ2 & RQ3)

Table 2: Ablation on the Input Modality.

| Method | Modal | mAP | Presence | Loc | Shape | mAS |
|--------|-------|-----|----------|-----|-------|-----|
| MapTR ○ | C | 44.1 | 91.2 | 65.4 | 90.6 | 71.6 |
| MapTR ● | C & L | **62.8** | **91.5** | **68.6** | **91.0** | **74.0** |
| GeMap ○ | C | 62.7 | **91.1** | 67.5 | **94.5** | **74.7** |
| GeMap ● | C & L | **66.5** | 89.1 | 66.3 | 92.7 | 71.8 |

Table 3: Ablation on the BEV Encoder.

| Method | Encoder | mAP | Presence | Loc | Shape | mAS |
|--------|---------|-----|----------|-----|-------|-----|
| MapTR ○ | BEVFormer | 41.6 | 89.6 | 69.7 | **90.6** | 71.3 |
| MapTR ○ | GKT | 44.1 | **91.2** | 65.4 | **90.6** | 71.6 |
| MapTR ● | BEVPool | **50.1** | 89.3 | **69.8** | 88.5 | **71.9** |

**Impact of Sensor Modality.** Our analysis reveals a nuanced relationship between sensor modality and temporal stability. As shown in Table 2, while LiDAR fusion consistently improves perception accuracy, increasing MapTR's (Liao et al., 2022) mAP by 42.6% (from 44.1 to 62.8) and GeMap's (Zhang et al., 2024c) mAP by 6.1% (from 62.7 to 66.5), its effect on temporal stability demonstrates significant model dependence. MapTR (Liao et al., 2022) benefits from sensor fusion with a 3.4% improvement in mAS (71.6 to 74.0), suggesting that LiDAR's precise depth measurements can enhance temporal consistency. In contrast, GeMap (Zhang et al., 2024c) experiences a 3.9% decrease

in mAS (74.7 to 71.8) despite accuracy gains, indicating potential architectural limitations in leveraging multi-modal signals for stable predictions. This divergence highlights that additional sensors alone cannot guarantee improved stability.

**Influence of BEV Encoding Strategies.**   Our ablation study on MapTR (Liao et al., 2022) demonstrates that different BEV encoders achieve similar overall temporal stability, with mAS scores ranging from 71.3 to 71.9, despite significant variations in accuracy, where mAP values span from 41.6 to 50.1, as summarized in Table 3. Further analysis reveals distinct specialization patterns among encoders. The GKT (Chen et al., 2022) encoder achieves superior Presence Stability at 91.2, ensuring consistent detection of map elements across frames. In comparison, BEVFormer (Li et al., 2024b) and BEVPool (Liu et al., 2022) excel in Localization Stability, with scores of 69.7 and 69.8 respectively, indicating their stronger capability in mitigating geometric jitter. These results highlight that BEV encoders embody characteristic preferences for different aspects of temporal stability, even within the same model architecture.

Table 4: Ablation on Temporal Fusion.

| Method | Temp | Initial Map | Back. | BEV Encoder | Epoch | mAP↑ | Presence↑ | Loc↑ | Shape↑ | mAS↑ |
|---|---|---|---|---|---|---|---|---|---|---|
| MapTR ○ | ✗ | ✗ | R50 | GKT | 24 | 44.1 | **91.2** | 65.4 | 90.6 | 71.6 |
| MapTR ● | ✓ | ✗ | R50 | GKT | 24 | 51.3 | 88.6 | 59.7 | 89.3 | 66.6 |
| MapTR ○ | ✗ | ✗ | R50 | BEVFormer | 24 | 41.6 | 89.6 | **69.7** | 90.6 | 71.3 |
| MapTR ● | ✓ | ✗ | R50 | BEVFormer | 24 | 53.3 | 90.4 | 69.5 | **91.2** | 73.0 |
| StreamMapNet ○ | ✗ | ✗ | R50 | BEVFormer-1 | 30 | 51.7 | 87.0 | **97.8** | 95.1 | 83.8 |
| StreamMapNet ● | ✓ | ✗ | R50 | BEVFormer-1 | 30 | 63.3 | **96.6** | 97.7 | 92.3 | **91.9** |
| MapTracker ○ | ✗ | ✗ | R18 | BEVFormer-2 | 72 | 62.8 | **95.3** | 97.3 | 85.9 | 87.4 |
| MapTracker ● | ✓ | ✗ | R18 | BEVFormer-2 | 72 | 71.9 | 92.9 | **98.5** | 94.8 | 89.9 |
| MapTracker ○ | ✗ | ✗ | R50 | BEVFormer-2 | 72 | 68.3 | 94.5 | 97.9 | 93.8 | 90.8 |
| MapTracker ● | ✓ | ✗ | R50 | BEVFormer-2 | 72 | 75.95 | 93.3 | **98.1** | 95.8 | 90.4 |
| HRMapNet ○ | ✓ | ✗ | R50 | BEVFormer-1 | 24 | 67.2 | 92.3 | 70.5 | 91.5 | 75.9 |
| HRMapNet ○ | ✓ | Testing Map | R50 | BEVFormer-1 | 24 | 73.0 | **94.9** | 71.4 | 93.0 | **78.4** |
| HRMapNet ● | ✓ | Training Map | R50 | BEVFormer-1 | 24 | 83.6 | 89.9 | **75.9** | 93.2 | 76.7 |

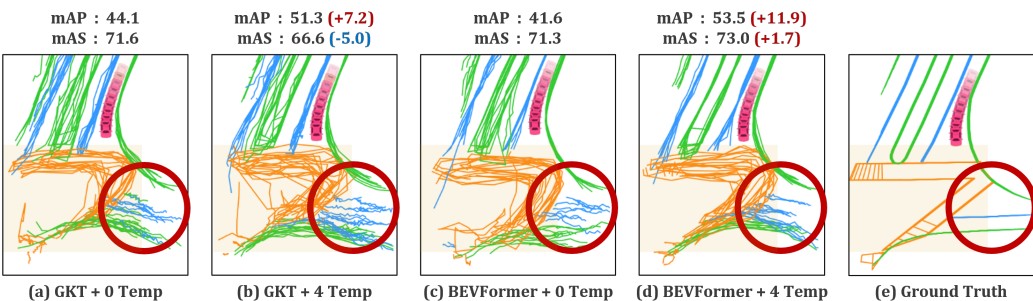

(a) GKT + 0 Temp    (b) GKT + 4 Temp    (c) BEVFormer + 0 Temp    (d) BEVFormer + 4 Temp    (e) Ground Truth

Figure 5: The dual effect of temporal fusion on MapTR with different BEV encoders.

**Discussion of Temporal Fusion.**   As shown in Table 4, the effectiveness of temporal fusion is highly dependent on architectural compatibility. Models with native temporal designs demonstrate robust performance: StreamMapNet (Yuan et al., 2024) achieves exceptional temporal stability (mAS: 91.9), while MapTracker (Chen et al., 2024) maintains strong stability (mAS: 90.4) alongside significant mAP improvement (+11.4%). In contrast, adding temporal fusion to architectures not originally designed for temporal processing yields inconsistent results. MapTR (Liao et al., 2022) exhibits divergent behaviors depending on its BEV encoder. With the GKT (Chen et al., 2022) encoder, temporal fusion degrades stability (mAS: -7.0%), whereas with BEVFormer (Li et al., 2024b), it provides balanced improvement (mAS: +2.4%, mAP: +28.1%). This contrast highlights the critical influence of the encoder's representation capacity on temporal integration. Furthermore, HRMapNet (Zhang et al., 2024b) demonstrates that while map priors substantially boost accuracy (mAP: +24.4% with training map priors), their impact on stability is more limited (mAS: +1.1%). This suggests that dynamic temporal modeling contributes more significantly to consistency than static priors. These findings collectively emphasize that effective temporal fusion requires co-design of architectural components rather than simply appending temporal modules.

Table 5: Ablation on the 2D Backbone.

| Method | Back. | mAP | Presence | Loc | Shape | mAS |
|---|---|---|---|---|---|---|
| MapTR ○ | R18 | 32.4 | 87.8 | **75.0** | 88.5 | **72.8** |
| MapTR ● | R50 | 44.1 | **91.2** | 65.4 | **90.6** | 71.6 |
| MapTRv2 ○ | R18 | 57.2 | 91.0 | **73.2** | 91.2 | 75.6 |
| MapTRv2 ● | R50 | 61.4 | **91.5** | 68.6 | 91.0 | 74.0 |
| MapQR ○ | R18 | 62.3 | 88.2 | 73.1 | 92.5 | 74.1 |
| MapQR ● | R50 | 66.4 | **91.8** | 75.6 | 91.6 | 77.8 |
| BeMapNet ○ | Effi-B0 | 60.7 | **100.0** | 67.9 | 97.9 | 82.9 |
| BeMapNet ○ | R50 | 63.6 | **100.0** | 65.8 | 97.9 | 81.9 |
| BeMapNet ● | Swin-T | 64.1 | **100.0** | 62.8 | **98.0** | 80.4 |
| PivotNet ○ | Effi-B0 | 57.8 | **100.0** | 71.8 | **97.2** | 84.5 |
| PivotNet ○ | R50 | 57.1 | **100.0** | 71.4 | **97.2** | 84.3 |
| PivotNet ● | Swin-T | 61.6 | **100.0** | 71.6 | **97.2** | 84.4 |
| GeMap ○ | R50 | 62.7 | 91.1 | 67.5 | **94.5** | 74.7 |
| GeMap ○ | Swin-T | 72.0 | 92.2 | **74.9** | 93.3 | **78.1** |
| GeMap ○ | V2-99 | 72.0 | 89.2 | 71.5 | 82.6 | 74.2 |
| GeMap ● | V2-99* | 76.0 | **93.4** | 67.0 | 93.7 | 75.1 |

Table 6: Ablation on the Training Epochs.

| Method | Epoch | mAP | Presence | Loc | Shape | mAS |
|---|---|---|---|---|---|---|
| MapTR-18 ○ | 24 | 32.4 | **87.8** | 75.0 | 88.5 | **72.8** |
| MapTR-18 ● | 110 | **45.5** | 86.0 | 71.7 | **94.8** | 71.9 |
| MapTR-50 ○ | 24 | 44.1 | **91.2** | 65.4 | 90.6 | **71.6** |
| MapTR-50 ● | 110 | **50.5** | 89.8 | 63.2 | 91.0 | 68.2 |
| MapQR-50 ○ | 24 | 66.4 | 91.8 | 75.6 | 91.6 | 77.8 |
| MapQR-50 ● | 110 | **72.6** | 92.4 | 75.9 | 96.4 | 80.3 |
| GeMap-50 ○ | 24 | 51.3 | 92.3 | 69.7 | 92.6 | 75.5 |
| GeMap-50 ● | 110 | 62.9 | 91.1 | 67.5 | 94.5 | 74.7 |
| BeMapNet ○ | 30 | 64.1 | 100.0 | 62.8 | 98.0 | 80.4 |
| BeMapNet ● | 110 | **68.3** | 100.0 | 64.0 | 98.2 | 81.1 |
| PivotNet ○ | 30 | 61.6 | 100.0 | 71.6 | 97.2 | 84.4 |
| PivotNet ● | 110 | **66.4** | 100.0 | 72.1 | 97.4 | 84.8 |
| MapTracker-18 ○ | 48 | 69.3 | 94.8 | 98.2 | 93.8 | **90.8** |
| MapTracker-18 ● | 72 | **71.9** | 92.9 | 98.5 | 94.8 | 89.9 |
| MapTracker-50 ○ | 48 | 72.96 | 91.8 | 98.5 | 96.0 | **91.7** |
| MapTracker-50 ● | 72 | **75.95** | 93.3 | 98.1 | 95.8 | 90.4 |

**Influence of The 2D Backbone.** The impact of the 2D backbone is model-specific as indicated in Table 5. A more powerful backbone consistently improves accuracy (mAP), as seen in MapTR (+36.1%) (Liao et al., 2022) and MapQR (+6.6%) (Liu et al., 2024b). However, its effect on stability (mAS) is less predictable, ranging from a slight decrease in MapTR (-1.6%) (Liao et al., 2022) to an increase in MapQR (+5.0%) (Liu et al., 2024b). We observe a recurring trade-off: stronger backbones often enhance Presence Stability (e.g., +3.4% for MapTR (Liao et al., 2022)) but can reduce Localization Stability (-12.8% for MapTR (Liao et al., 2022)), suggesting a potential focus on semantic over geometric consistency.

**Impact of Training Regimen.** Our analysis reveals distinct patterns in how extended training affects model performance across different architectures. As shown in Table 6, while longer training epochs consistently improve accuracy (mAP increases ranging from +4.3% to +40.1%), the effects on temporal stability vary significantly. We observe three distinct learning behaviors. First, models like MapTR (Liao et al., 2022) exhibit stability erosion, where accuracy gains (+22.8% for MapTR-50) come with stability degradation (-4.7% mAS). Second, architectures such as MapQR (Liu et al., 2024b) and PivotNet (Ding et al., 2023) demonstrate stability saturation, maintaining or slightly improving mAS (+3.2% and +0.5% respectively) while achieving accuracy improvements. Third, complex temporal models like MapTracker (Chen et al., 2024) show optimization sensitivity, where extended training improves mAP (+3.7% to +4.2%) but leads to slight mAS reductions (-1.0% to -1.4%). These patterns underscore that temporal stability responds differently to extended training based on architectural inductive biases, suggesting that stability should be explicitly optimized rather than expected to emerge from accuracy-focused training alone.

## 4.4 GENERAL DISCUSSION

Our benchmark reveals that **temporal stability (mAS) is an independent performance dimension** from accuracy (mAP), challenging the prevailing focus on single-frame precision. Models with high mAP can exhibit significant instability, underscoring the need for dual optimization. **Architectural choices induce distinct stability profiles.** Multi-sensor fusion improves accuracy but affects stability model-dependently. BEV encoders specialize differently: GKT (Chen et al., 2022) favors detection consistency while BEVFormer (Li et al., 2024b) variants reduce geometric jitter. Temporal fusion effectiveness hinges on architectural compatibility, with native designs outperforming retrofitted modules. **Training dynamics diverge by architecture.** Extended training improves accuracy consistently but affects stability variably, revealing three patterns: erosion (MapTR (Liao et al., 2022)), saturation (MapQR (Liu et al., 2024b)), and sensitivity (MapTracker (Chen et al., 2024)). This indicates stability requires explicit optimization rather than emerging implicitly from accuracy-focused training.

These findings advocate for **co-equal treatment of stability and accuracy** in evaluation and design. The substantial stability gaps among models (mAS: 66.6–91.9) highlight critical improvement opportunities. Future work should develop architectures that explicitly joint-optimize both criteria for trustworthy autonomous driving systems. mAS is designed to complement mAP, not replace it.

Models with high mAS but low mAP indicate pseudo-stability. Detailed discussion is provided in Appendix C.

## 5  CONCLUSION

In this work, we address the critical yet overlooked aspect of temporal stability in online HD mapping evaluation. While significant progress has been made in single-frame accuracy, the consistency of model outputs across sequential frames, which is essential for safe deployment, has remained largely unquantified. To bridge this gap, we introduce a multi-dimensional stability evaluation framework with novel metrics for presence, localization, and shape stability, integrated into a unified mean Average Stability (mAS) score. Extensive benchmarking demonstrates that accuracy (mAP) and stability (mAS) represent independent performance dimensions, challenging the assumption that accuracy optimization alone ensures real-world reliability. Our analysis further reveals how architectural choices, including temporal fusion strategies, sensor modality, training regimens, backbone designs, and BEV encoders, distinctly influence both accuracy and stability. By establishing the first stability-centric benchmark, we aim to shift community focus beyond accuracy alone and inspire the development of next-generation online HD mapping systems that achieve both high accuracy and temporal consistency, thereby advancing more robust and trustworthy autonomous driving.

## ACKNOWLEDGMENTS

This work was supported in part by the the Beijing Natural Science Foundation under Grant L243008, in part by the Shandong Provincial Natural Science Foundation under Grant ZR2024LZN010, in part by the National Natural Science Foundation of China (NSFC) under Grant 52441202, in part by the Ganwei Program of Beihang University under Grant WZ2024-2-16, and in part by the Fundamental Research Funds for the Central Universities.

**Ethics Statement.** This work presents a benchmarking framework for evaluating temporal stability in online HD mapping. All experiments are conducted on the publicly available nuScenes dataset (Caesar et al., 2020) under standard research protocols, with no human subjects involved. While the study itself does not collect sensitive data, we recognize that broader deployment of HD mapping technology raises important societal considerations: privacy risks from detailed environmental sensing, safety implications if models are overtrusted in autonomous systems, computational and environmental costs of large-scale model training, and potential performance disparities across diverse driving environments. We encourage the community to address these concerns through responsible data practices, robust safety validation, efficient algorithms, and inclusive evaluation. This work adheres to the ICLR Code of Ethics and aims to promote safer and more reliable autonomous systems through transparent and reproducible evaluation.

**Reproducibility statement.** To ensure the reproducibility of our work, we provide a comprehensive benchmark toolkit, including evaluation code and visualization script complete with full technical documentation, at https://stablehdmap.github.io/. The implementation details of all models, training configurations, and hyperparameters are described in Section 4.1 and the Appendix. Our stability evaluation framework is fully detailed in Section 3, with algorithmic steps and metric formulations explicitly defined. The nuScenes dataset (Caesar et al., 2020) used in this study is publicly available, and all preprocessing steps follow standard practices as cited. We encourage the community to use our released resources to facilitate fair comparisons and further research.

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

# Appendix

In the appendix, we supply further details on the proposed stability evaluation framework, the benchmark setup, experimental analyses, and visualizations that are omitted from the main paper for brevity. The appendix is structured as follows:

- Sec. A provides additional statements on the use of Large Language Models (LLMs) in this work.

- Sec. B presents implementation details of the stability evaluation algorithm pipeline.

- Sec. C offers supplementary experimental setups and related ablation studies.

- Sec. D presents additional analyses from 10 different models.

- Sec. E displays supplementary visualizations, encompassing additional mAP *vs.* mAS comparisons , and illustrations of how unstable predictions affect downstream tasks.

- Sec. F discusses the limitations of our work and provides an outlook on future work.

## A    STATEMENT ON THE USE OF LARGE LANGUAGE MODELS (LLMS)

In the preparation of this paper, large language models (LLMs) were used solely as an assistive tool for writing refinement and polishing. The core research ideas, theoretical framework, experimental design, data analysis, and result interpretation were entirely conceived and conducted by the human authors. The LLM was employed after the intellectual substance of the work was fully established, specifically to assist with improving grammatical correctness, sentence fluency, and overall clarity of expression in certain parts of the manuscript. It did not contribute to the scientific ideation, methodological development, or conclusions of the research. All final content was thoroughly reviewed, verified, and approved by the authors.

## B    STABILITY EVALUATION ALGORITHM WITH ADDITIONAL DETAILS

This section provides comprehensive algorithmic details for the multi-dimensional map stability evaluation framework introduced in Section 3. The complete pipeline consists of four main stages: temporal sampling, cross-frame instance matching, geometric alignment and resampling, and stability metric computation. Each stage is implemented through carefully designed algorithms that ensure robust and reproducible evaluation of temporal stability in online HD mapping.

### B.1    TEMPORAL SAMPLING

Algorithm 1 implements the temporal sampling stage that constructs frame pairs for stability analysis. The algorithm randomly selects subsequent frames within a predefined maximum temporal interval $M$ for each anchor frame, ensuring comprehensive coverage of temporal variations while maintaining computational efficiency. This approach generates a sampling set $S$ of size $|S| = L - M$, providing the foundational inputs for subsequent stability analysis.

---

**Algorithm 1** Temporal Sampling Algorithm

---

**Require:** Model output frame sequence $\{D_1, D_2, \ldots, D_L\}$, maximum temporal interval $M$
**Ensure:** Frame pair sampling set $S$
1: **// Stage 1: Temporal Sampling**
2: $S \leftarrow \emptyset$ {Initialize sampling set}
3: **for** $t = 1$ to $L - M$ **do**
4:     $k \leftarrow \text{RandomSample}(1, M)$ {Random sampling within $[1, M]$ range}
5:     $S \leftarrow S \cup \{(D_t, D_{t+k})\}$ {Add frame pair to sampling set}
6: **end for**
7: **return** $S$

---

B.2 CROSS-FRAME INSTANCE MATCHING

Algorithm 2 implements the cross-frame instance matching stage that establishes correspondence between map elements across temporal frames.

---
**Algorithm 2** Cross-Frame Instance Matching Algorithm
---
**Require:** Frame pair sampling set $S$
**Ensure:** Matched polyline pairs $\mathcal{M}$
 1: **// Stage 2: Cross-Frame Instance Matching**
 2: $\mathcal{M} \leftarrow \emptyset$ {Initialize matching result set}
 3: **for** each $(D_t, D_{t+k}) \in S$ **do**
 4:    **// Step 2.1: Frame-to-GT Matching**
 5:    $\text{matches}_t \leftarrow \text{HungarianMatching}(D_t, \text{GT}_t)$
 6:    $\text{matches}_{t+k} \leftarrow \text{HungarianMatching}(D_{t+k}, \text{GT}_{t+k})$
 7:    **// Step 2.2: GT-based Association**
 8:    $E \leftarrow \text{FindCommonGTInstances}(\text{matches}_t, \text{matches}_{t+k})$
 9:    **for** each $e \in E$ **do**
10:       $\text{poly}_t(e) \leftarrow \text{GetPolyline}(\text{matches}_t, e)$
11:       $\text{poly}_{t+k}(e) \leftarrow \text{GetPolyline}(\text{matches}_{t+k}, e)$
12:       $\mathcal{M} \leftarrow \mathcal{M} \cup \{(\text{poly}_{t+k}(e), \text{poly}_t(e))\}$
13:    **end for**
14: **end for**
15: **return** $\mathcal{M}$

---

The algorithm employs a two-step strategy: first matching predictions to ground truth within each frame using the algorithm 3, then establishing temporal correspondence through ground truth-based association.

---
**Algorithm 3** Hungarian Matching Sub-algorithm
---
**Require:** Prediction frame $D$, ground truth frame GT, cost function $C$
**Ensure:** Matching result matches
 1: $P \leftarrow \text{GetPolygons}(D)$ {Get prediction polygons}
 2: $G \leftarrow \text{GetPolygons}(\text{GT})$ {Get ground truth polygons}
 3: $n \leftarrow |P|, m \leftarrow |G|$
 4: **// Build cost matrix**
 5: $C_{\text{matrix}} \leftarrow \text{zeros}(n \times m)$
 6: **for** $i = 1$ to $n$ **do**
 7:    **for** $j = 1$ to $m$ **do**
 8:       $C_{\text{matrix}}[i, j] \leftarrow C(P[i], G[j])$ {Geometric and semantic similarity cost}
 9:    **end for**
10: **end for**
11: **// Execute Hungarian algorithm**
12: $\text{matches} \leftarrow \text{HungarianAlgorithm}(C_{\text{matrix}})$
13: **return** matches

---

This indirect matching approach leverages the consistency of ground truth annotations to overcome the inherent inconsistencies in model predictions, yielding a set of matched instance pairs for each frame pair.

B.3 GEOMETRIC ALIGNMENT AND RESAMPLING

Algorithm 4 implements the geometric alignment and resampling stage that ensures spatially consistent comparison between matched polylines.

The algorithm transforms historical polylines into the current frame's coordinate system, applies algorithm 5 to ensure evaluation consistency, and performs uniform resampling along the $x$-axis. This process guarantees spatially aligned and comparable point sets for subsequent stability analysis, returning a comprehensive set of matched and aligned polyline pairs, each annotated with the

---

**Algorithm 4** Geometric Alignment and Resampling Algorithm

---

**Require:** Matched polyline pairs $\mathcal{M}$, perception range $[x_{\min}, x_{\max}, y_{\min}, y_{\max}]$, resampling points $N$
**Ensure:** Matched and aligned polyline pairs $\mathcal{M}^{\text{sample}}$
1: **// Stage 3: Geometric Alignment and Resampling**
2: $\mathcal{M}^{\text{sample}} \leftarrow \emptyset$ {Initialize sampled polyline pairs set}
3: **for** each $(\text{poly}_{t+k}(e), \text{poly}_t(e)) \in \mathcal{M}$ **do**
4:     **// Step 3.1: Coordinate Transformation**
5:     $\text{poly}_{t \to t+k}(e) \leftarrow T_{\text{world} \to t+k} \cdot T_{t \to \text{world}} \cdot \text{poly}_t(e)$
6:     **// Step 3.2: Perception Range Filtering**
7:     $\text{poly}_{t \to t+k}(e) \leftarrow \text{ClipToPerceptionRange}(\text{poly}_{t \to t+k}(e), [x_{\min}, x_{\max}, y_{\min}, y_{\max}])$
8:     **// Step 3.3: Dynamic Axis Selection and Uniform Resampling**
9:     $\mathcal{I} \leftarrow \text{DynamicAxisSelection}(\text{poly}_{t+k}(e))$ {Divide polyline into intervals with primary sampling axis}
10:     $\{n_j\}_{j=1}^K \leftarrow \text{AllocateSamplingPoints}(\mathcal{I}, N)$ {Allocate $N$ points proportionally to intervals}
11:     $\text{poly}_{t+k}^{\text{sample}}(e) \leftarrow \emptyset, \text{poly}_t^{\text{sample}}(e) \leftarrow \emptyset$
12:     **for** each interval $I_j \in \mathcal{I}$ with $n_j$ sampling points **do**
13:         **if** $I_j$ is X-sampling interval **then**
14:             Generate $n_j$ uniform $x$-values in $[x_{\min}^j, x_{\max}^j]$
15:             Compute corresponding $y$-coordinates via linear interpolation
16:         **else**
17:             Generate $n_j$ uniform $y$-values in $[y_{\min}^j, y_{\max}^j]$
18:             Compute corresponding $x$-coordinates via linear interpolation
19:         **end if**
20:         Add sampled points to $\text{poly}_{t+k}^{\text{sample}}(e)$ and $\text{poly}_t^{\text{sample}}(e)$
21:     **end for**
22:     $\mathcal{M}^{\text{sample}} \leftarrow \mathcal{M}^{\text{sample}} \cup \{(\text{poly}_{t+k}^{\text{sample}}(e), \text{poly}_t^{\text{sample}}(e), e)\}$
23: **end for**
24: **return** $\mathcal{M}^{\text{sample}}$

---

corresponding map element identifier. The algorithm transforms historical polylines into the current frame's coordinate system, applies algorithm 5 to ensure evaluation consistency, and performs uniform resampling using a dynamic axis selection mechanism (detailed below). This process guarantees spatially aligned and comparable point sets for subsequent stability analysis, returning a comprehensive set of matched and aligned polyline pairs, each annotated with the corresponding map element identifier.

**Dynamic Axis Selection Mechanism.** For a map instance composed of a point sequence $(x_1, y_1), (x_2, y_2), \ldots, (x_M, y_M)$, we do not perform resampling along a fixed axis. Instead, we dynamically determine the primary sampling axis by analyzing its local geometric orientation. The specific procedure is as follows:

1. **Calculation of Local Direction Vectors**: For each consecutive segment $\vec{s_j}$ in the instance, formed by points $p_j$ and $p_{j+1}$, its direction vector is computed as $\vec{v_j} = (dx_j, dy_j) = (x_{j+1} - x_j, y_{j+1} - y_j)$.

2. **Determination of the Primary Sampling Axis**: We compute the absolute changes of the segment along each coordinate axis:

$$\Delta x_j = |dx_j|, \quad \Delta y_j = |dy_j|$$

   If $\Delta y_j > \Delta x_j$, the segment is considered **vertically oriented** and categorized as an **"X-sampling interval"**. Conversely, if $\Delta x_j > \Delta y_j$, it is considered **horizontally oriented** and categorized as a **"Y-sampling interval"**.

3. **Determination of the Number of Sampling Points per Interval**: The number of sampling points in each interval is determined based on its length. First, the instance is divided into $K$ intervals using the dynamic axis selection method described previously. For each interval $j$, we calculate its length along the **primary sampling axis**:

   - **For an X-sampling interval:** $L_j = x_{\max}^j - x_{\min}^j$
   - **For a Y-sampling interval:** $L_j = y_{\max}^j - y_{\min}^j$

Subsequently, sampling points are allocated proportionally based on the relative length of each interval. The initial number of sampling points $n_j$ for each interval is calculated as follows:

$$n_j = \text{round}\left(N \times \frac{L_j}{\sum_{t=1}^{K} L_t}\right)$$

where $\text{round}(\cdot)$ denotes rounding to the nearest integer.

4. **Total Adjustment Mechanism**: Since rounding may cause the total number of points $\sum_{i=1}^{K} n_i \neq N$, we introduce an adjustment mechanism:

   - If $\sum_{i=1}^{K} n_i < N$: Points are incrementally allocated to intervals in descending order of their length ratio $\frac{L_i}{\sum L_j}$ until the total number of points equals $N$.

   - If $\sum_{i=1}^{K} n_i > N$: Points are sequentially removed from intervals in ascending order of their length ratio until the total number of points equals $N$.

5. **Axial Resampling Execution**:

   - Within an **X-sampling interval**, we uniformly generate $n_s$ sample points $x_i$ across the x-value range $[x_{\min}^s, x_{\max}^s]$, and compute the corresponding $y$-coordinates via linear interpolation:

   $$x_i = x_{\min}^s + (i-1) \cdot \frac{x_{\max}^s - x_{\min}^s}{n_s - 1}, \quad i = 1, 2, \ldots, n_s$$

   $$\text{poly}^{\text{sample}} = \{(x_i, y(x_i))\}$$

   - Within a **Y-sampling interval**, we uniformly generate $n_s$ sample points $y_i$ across the y-value range $[y_{\min}^s, y_{\max}^s]$, and compute the corresponding $x$-coordinates via linear interpolation:

   $$y_i = y_{\min}^s + (i-1) \cdot \frac{y_{\max}^s - y_{\min}^s}{n_s - 1}, \quad i = 1, 2, \ldots, n_s$$

   $$\text{poly}^{\text{sample}} = \{(x(y_i), y_i)\}$$

This mechanism ensures that, regardless of the map element's orientation, uniform sampling is always performed along its locally dominant direction of extension, thereby establishing a **fair and robust benchmark** for point-to-point geometric comparison. Specifically designed for the **piecewise-smooth polyline structures** prevalent in online HD maps.

---

**Algorithm 5** Perception Range Filtering Sub-algorithm

---

**Require:** Transformed polyline $\text{poly}_{t \to t+k}(e)$, perception range $[x_{\min}, x_{\max}, y_{\min}, y_{\max}]$
**Ensure:** Filtered polyline $\text{poly}_{\text{filtered}}$
1: $\text{poly}_{\text{filtered}} \leftarrow \emptyset$
2: **for** each point $(x, y) \in \text{poly}_{t \to t+k}(e)$ **do**
3:     **if** $x_{\min} \leq x \leq x_{\max}$ AND $y_{\min} \leq y \leq y_{\max}$ **then**
4:         $\text{poly}_{\text{filtered}} \leftarrow \text{poly}_{\text{filtered}} \cup \{(x, y)\}$
5:     **end if**
6: **end for**
7: **return** $\text{poly}_{\text{filtered}}$

---

## B.4 STABILITY METRIC COMPUTATION

Algorithm 6 implements the core stability metric computation that quantifies temporal stability across three dimensions: presence, localization, and shape. The algorithm processes each matched polyline pair to compute individual stability scores, then aggregates these scores at the class and model levels. The presence stability evaluates detection consistency, the localization stability quantifies positional jitter, and the shape stability assesses geometric consistency through curvature comparison.

---

**Algorithm 6** Multi-dimensional Map Stability Evaluation Framework - Stability Metric Computation

---

**Require:** Matched and aligned polyline pairs $\mathcal{M}^{\text{sample}}$, detection threshold $\tau$, weighting parameter $\omega$, scaling parameter $\beta$
**Ensure:** Overall model stability score mAS
1: **// Stage 4: Stability Metric Computation**
2: $\mathcal{C} \leftarrow \text{GetAllClasses}(\mathcal{M}^{\text{sample}})$ {Get all classes}
3: **for** each class $\in \mathcal{C}$ **do**
4:     $\mathcal{I}_{\text{class}} \leftarrow \text{GetInstancesOfClass}(\mathcal{M}^{\text{sample}}, \text{class})$
5:     $\text{stability}_{\text{class}} \leftarrow 0$
6:     **for** each $(\text{poly}_{t+k}^{\text{sample}}(e), \text{poly}_t^{\text{sample}}(e), e) \in \mathcal{I}_{\text{class}}$ **do**
7:         **// Compute Presence Stability**
8:         **if** $\text{score}_{t+k}(e) \geq \tau$ AND $\text{score}_t(e) \geq \tau$ OR $\text{score}_{t+k}(e) < \tau$ AND $\text{score}_t(e) < \tau$ **then**
9:             $\text{Presence}(e) \leftarrow 1$
10:        **else**
11:           $\text{Presence}(e) \leftarrow 0.5$ {Flickering case}
12:        **end if**
13:        **// Compute Localization Stability**
14:        $\text{avg\_deviation} \leftarrow \frac{1}{N} \sum_{i=1}^{N} |y_{t+k}(x_i) - y_t(x_i)|$
15:        $\text{Loc}(e) \leftarrow \beta \cdot \text{avg\_deviation}$
16:        **// Compute Shape Stability**
17:        $\kappa_{t+k} \leftarrow \text{ComputeCurvature}(\text{poly}_{t+k}^{\text{sample}}(e))$
18:        $\kappa_t \leftarrow \text{ComputeCurvature}(\text{poly}_t^{\text{sample}}(e))$
19:        $\text{Shape}(e) \leftarrow 1 - \frac{|\kappa_{t+k} - \kappa_t|}{\pi}$
20:        **// Compute Comprehensive Stability**
21:        $\text{Stability}(e) \leftarrow \text{Presence}(e) \cdot [\omega \cdot \text{Loc}(e) + (1 - \omega) \cdot \text{Shape}(e)]$
22:        $\text{stability}_{\text{class}} \leftarrow \text{stability}_{\text{class}} + \text{Stability}(e)$
23:     **end for**
24:     $\text{Stability}_{\text{class}} \leftarrow \frac{\text{stability}_{\text{class}}}{|\mathcal{I}_{\text{class}}|}$
25: **end for**
26: **// Compute Overall Model Stability**
27: $\text{mAS} \leftarrow \frac{1}{|\mathcal{C}|} \sum_{\text{class} \in \mathcal{C}} \text{Stability}_{\text{class}}$
28: **return** mAS

---

The specific implementation of the ComputeCurvature function mentioned in Algorithm 6 is detailed in Algorithm 7, which approximates polyline curvature by computing the average angles between consecutive segments, thereby providing a robust geometric measurement method with translation and rotation invariance.

---

**Algorithm 7** Curvature Computation Sub-algorithm

---

**Require:** Resampled polyline $\text{poly}^{\text{sample}} = \{(x_i, y_i) \mid i = 1, 2, \ldots, N\}$
**Ensure:** Curvature $\kappa$
1: $\kappa \leftarrow 0$
2: **for** $j = 1$ to $N - 1$ **do**
3:     $\vec{v_j} \leftarrow (x_{j+1} - x_j, y_{j+1} - y_j)$ {Compute vector $\vec{v_j}$}
4:     $\vec{v_{j+1}} \leftarrow (x_{j+2} - x_{j+1}, y_{j+2} - y_{j+1})$ {Compute vector $\vec{v_{j+1}}$}
5:     $\theta_j \leftarrow \cos^{-1}\left(\frac{\vec{v_j} \cdot \vec{v_{j+1}}}{|\vec{v_j}| \cdot |\vec{v_{j+1}}|}\right)$ {Compute angle}
6:     $\kappa \leftarrow \kappa + \theta_j$
7: **end for**
8: $\kappa \leftarrow \frac{\kappa}{N-1}$ {Compute average curvature}
9: **return** $\kappa$

---

This curvature-based approach enables effective comparison of geometric consistency across temporal frames while capturing subtle shape variations that may indicate instability in the model's predictions.

Upon completion of all sub-metric evaluations, the final mean Average Stability (mAS) score is computed, providing a holistic measure of the model's temporal stability across all evaluated classes.

## C    Further Details on the Experimental Setup

### C.1    Supplemental Details on the Experimental Setup

Prior to conducting experiments, it is necessary to configure certain hyperparameters. This section primarily elaborates on the detailed configurations of these hyperparameters adopted in our study, along with the rationale for these choices:

- Maximum frame interval ($M$=2): The configuration of different frame intervals essentially represents distinct evaluation scenarios for stability assessment, each carrying unique implications. Therefore, in addition to the experiments with $M$=2 presented in the main text, as shown in chapter C.4.3, we have conducted supplementary experiments with $M$=3, $M$=5, and $M$=10 to provide as comprehensive a stability evaluation as possible for existing models.

- Number of resampling points ($N$=100): The purpose of resampling is to adjust the distribution of map points on two instance polylines to be identical, thereby facilitating the calculation of stability metrics. Thus, the value of N should not be set too small to avoid undue influence from individual outliers on the instances in subsequent computations. However, beyond a sufficient threshold, variations in N do not significantly affect stability evaluation outcomes. Conversely, excessively large values of N may substantially reduce computational efficiency. Balancing resampling granularity and computational cost, we ultimately set N = 100. This value can be appropriately adjusted in different experimental settings.

- Position Stability Scaling Factor ($\beta$=15.0): The specific implication of this scaling factor is that when the distance between two matched map points (i.e., points sampled at identical x-values on matched map instances) in adjacent frames equals $\beta$, their positional stability is considered zero. Consequently, the value of $\beta$ corresponds to the distance threshold representing extreme instability. Typically, we define such extreme cases using the shorter map radius (half the length of the map's shorter side). In prevailing map construction paradigms, the standard map range is generally defined as x $\in [-15, 15]$, y $\in [-30, 30]$. Therefore, in our experiments, $\beta$ is set to 15.0.

### C.2    Design Rationale for Stability Metrics and Parameter Selection

#### C.2.1    Shape Stability Design Motivation

**Design Motivation: Capturing Perceptual Shape Instability Rather Than Precise Differential Geometric Properties**    Our primary objective is to evaluate the temporal stability of map element shapes, rather than to compute their precise differential geometric curvature. In the context of online mapping, instability typically manifests as noticeable fluctuations or jitter in the polyline contours between consecutive frames. A typical example would be a smoothly curved road being temporarily predicted as an unnaturally sharp corner in certain frames. This type of macro-level shape flickering represents exactly the kind of instability that downstream planning modules are particularly sensitive to.

- **The Average Inter-Segment Angle Effectively Captures Macroscopic Instabilities**: The average inter-segment angle, defined as $\kappa(\text{poly}) = \frac{1}{N-1} \sum_{j=1}^{N-1} \theta_j$, essentially measures the cumulative rate of directional change along the entire curve. Any sudden and unstable sharp corner or jagged fluctuation will significantly increase the value of $\sum \theta_j$, and will therefore be effectively captured by our Shape Stability metric $(1 - |\Delta\kappa|/\pi)$.

- **Robustness to Microscopic Noise**: The sensitivity concern raised by the reviewer is substantially mitigated through our fixed and reasonable sampling strategy. By employing a **fixed sampling point count (N=100)**, we ensure **uniform and consistent sampling** for all instances. Under such uniform and consistent sampling conditions, minor variations in segment angles caused by sampling artifacts tend to be smoothed out through the averaging process, while genuine shape instabilities emerge as significant signals.

**Critical Balance Between Practicality and Computational Efficiency** Online HD mapping systems place high demands on evaluation efficiency, particularly in large-scale benchmarking. It is essential to strike a balance between the discriminative power of a metric and its computational cost.

Regarding the Fréchet distance, while it is an excellent metric for curve similarity, its computational complexity is relatively high as it typically requires dynamic programming. This becomes a bottleneck in benchmarking scenarios that require rapid evaluation of tens of thousands of instance pairs.

Concerning curvature signatures, these require denser sampling and more complex local computations. They present similar challenges in computational efficiency and implementation complexity, while also introducing the need for additional parameter selection such as determining the appropriate scale for Gaussian smoothing.

In contrast, our method based on the average inter-segment angle offers distinct advantages. It demonstrates high computational efficiency since it involves only vector dot products and inverse trigonometric calculations, with a complexity of O(N). This makes it highly suitable for large-scale evaluation. Furthermore, it maintains a clear physical interpretation as the output, represented by average angular change, is straightforward to understand and visualize, providing intuitive feedback for model diagnosis.

### C.2.2 PRESENCE STABILITY DESIGN RATIONALE

We wish to emphasize that **one of the core design objectives of our proposed presence metric is precisely to directly capture various types of prediction inconsistencies, including semantic flickering.**

First, the "semantic flickering" mentioned would lead to a **significant decrease in the presence metric**, thereby reducing the mAS. When a model exhibits semantic flickering. For example, if a map instance is recognized as a "lane divider" at frame $t$ but as a "road boundary" at frame $t + 1$. Our presence metric is heavily penalized once such a mismatch occurs.

Furthermore, the presence metric also accurately captures another form of instability: even after accounting for range effects, if a map instance is detected at frame $t$ but missed at frame $t + 1$, the presence metric similarly incurs a strong penalty.

### C.2.3 LOCALIZATION STABILITY PARAMETER SELECTION

Regarding the scaling coefficient $\beta$ in the localization stability formula: $\text{Loc}(e) = 1 - \frac{1}{\beta} \cdot \frac{1}{N} \sum_{i=1}^{N} |y_{t+k}(x_i) - y_t(x_i)|$. The mAS metric necessarily decreases monotonically as $\beta$ decreases. The selection of $\beta = 15$ in our experimental setup was based on the following reasoning:

First, we clarify the physical meaning of $\beta$. As shown in the formula, when the average distance between corresponding map points of two instances reaches $\frac{1}{N} \sum_{i=1}^{N} |y_{t+k}(x_i) - y_t(x_i)| = \beta$, then $\text{Loc}(e) = 0$. Therefore, $\beta$ represents the distance threshold at which our framework applies the maximum penalty to localization stability. Specifically, when the average distance between map points of the same instance across two frames reaches $\beta$, our metric considers this instance "completely unstable" between these frames.

This threshold corresponds to the extreme case of "complete instability." We therefore define this threshold as the map's short-range radius, which in the models we evaluated corresponds to $\beta = 15$. This value represents a distance at which localization errors would be considered critically significant for autonomous driving applications.

Experiments with different values of $\beta$ were conducted, and the results align with the theoretical reasoning, demonstrating that the mAS metric increases monotonically as the value of $\beta$ increases.

Based on the theoretical analysis and experimental results presented above, we argue that the value of $\beta$ should not be determined through experimental selection alone, but should instead be chosen as a physically meaningful and appropriately justified value.

### C.2.4 COMPREHENSIVE STABILITY INDEX PARAMETER SELECTION

Regarding the balancing coefficient $\omega$ used in the comprehensive stability calculation, it serves to balance the relative importance between Localization Stability and Shape Stability. Our selection of $\omega = 0.7$ was based on the following rationale: In autonomous driving, the localization stability of map elements (such as the lateral position of lane lines or boundaries) directly impacts path planning, vehicle control, and safety. For instance, localization jitter may cause the vehicle to deviate abruptly from its lane, whereas shape variations (such as subtle changes in curvature) generally have a lesser impact, unless severe distortion occurs. Therefore, we assign a higher weight (70%) to localization stability to reflect its greater importance in real-world driving scenarios.

### C.3 EVALUATION UNDER ADVERSE WEATHER CONDITIONS

To ensure a fair and consistent evaluation, we did not introduce a new dataset but instead followed the established practice from prior online HD mapping work, PivotNet, by curating a subset of challenging scenarios from the nuScenes dataset. Specifically, we selected scenes under overcast, rainy, and nighttime conditions, resulting in a total of 78 scenes for evaluation.

Table 7: Evaluation Results Under Adverse Weather Conditions.

| Modal | Temp | Backbone | BEV | Epoch | $mAS_{pre}$ | $mAS_{loc}$ | $mAS_{shape}$ | $mAS$ |
|-------|------|----------|-----|-------|-------------|-------------|---------------|-------|
| Only C | $\times$ | R18 | GKT | 24 | 0.5773 | 0.7035 | 0.8556 | 0.4457 |
| Only C | $\times$ | R18 | GKT | 110 | 0.5668 | 0.6958 | 0.8883 | 0.4442 |
| Only C | $\times$ | R50 | GKT | 24 | 0.5683 | 0.6062 | 0.8538 | 0.4149 |
| Only C | $\times$ | R50 | GKT | 110 | 0.5762 | 0.7033 | 0.874 | 0.446 |
| Only C | $\times$ | R50 | BEVFormer | 24 | 0.5683 | 0.7195 | 0.8543 | 0.4416 |
| Only C | $\times$ | R50 | BEVPool | 24 | 0.5639 | 0.699 | 0.8531 | 0.4319 |
| Only C | $\checkmark$ | R50 | GKT | 24 | 0.5624 | 0.7224 | 0.872 | 0.4419 |
| Only C | $\checkmark$ | R50 | BEVFormer | 24 | 0.5663 | 0.7324 | 0.8673 | 0.4475 |
| C+L | $\times$ | R50 | GKT | 24 | 0.5662 | 0.7173 | 0.858 | 0.4391 |

Based on the results in the table above, it can be observed that all stability metrics show a significant decrease under adverse weather conditions. Moreover, while the stability metrics decrease markedly, they also tend to converge across different models. This occurs because under adverse weather conditions, model instability manifests more prominently in terms of instance presence stability, that is, whether an instance is detected or persists over time, rather than in positional or shape variations. As a result, the presence stability metric drops significantly and becomes similar across models. This finding aligns with the expected performance drop of existing models under challenging weather conditions.

In our metric design, since the location and shape stability metrics can only reflect the stability of instances that are consistently present, we incorporated the presence stability as a weighting coefficient in the overall metric. This ensures a comprehensive and accurate representation of stability in terms of presence, location, and shape. Consequently, when the presence stability is low, variations in location and shape stability have a diminished impact on the overall metric. This design leads to the overall mAS scores exhibiting a distribution similar to that of the presence metric, showing lower values that are closely clustered across different models.

### C.4 ABLATION STUDIES ON HYPERPARAMETERS

### C.4.1 ABLATION STUDY ON THE LOCALIZATION STABILITY SCALING FACTOR $\beta$

Experiments with different values of $\beta$ were conducted, and the results align with the theoretical reasoning, demonstrating that the mAS metric increases monotonically as the value of $\beta$ increases.

### C.4.2 ABLATION STUDY ON THE RESAMPLING DENSITY $N$

Concerning the resampling density ($N$), the design principle is to ensure it is **significantly higher** than the number of points in the vast majority of original map instances. In current mainstream

Table 8: Ablation Study on the Localization Stability Scaling Factor $\beta$.

| Model | $\beta$ | Presence | Loc | Shape | mAS |
|---|---|---|---|---|---|
| MapTRv1 noTemp R18-GKT-24 | 5 | 0.911 | 0.4097 | 0.4097 | 0.6053 |
| | 10 | 0.8776 | 0.6768 | 0.886 | 0.697 |
| | 15 | 0.8776 | 0.7499 | 0.8852 | 0.7283 |
| | 20 | 0.8776 | 0.8012 | 0.886 | 0.7497 |
| MapTRv1 noTemp R50-GKT-24 | 5 | 0.8776 | 0.5601 | 0.886 | 0.6471 |
| | 10 | 0.911 | 0.561 | 0.9084 | 0.6737 |
| | 15 | 0.912 | 0.6544 | 0.9063 | 0.7158 |
| | 20 | 0.911 | 0.7052 | 0.9084 | 0.7393 |
| MapTRv1 Temp R50-GKT-24 | 5 | 0.8861 | 0.3955 | 0.5812 | 0.5812 |
| | 10 | 0.8861 | 0.5017 | 0.896 | 0.6265 |
| | 15 | 0.8861 | 0.5974 | 0.8928 | 0.6657 |
| | 20 | 0.8861 | 0.6325 | 0.896 | 0.6824 |
| MapTRv1 Temp R50-BevFormer-24 | 5 | 0.9035 | 0.4885 | 0.913 | 0.6382 |
| | 10 | 0.9035 | 0.6411 | 0.913 | 0.7091 |
| | 15 | 0.9035 | 0.6947 | 0.9121 | 0.73 |
| | 20 | 0.9035 | 0.7931 | 0.913 | 0.7751 |
| MapTRv2 noTemp R50-BevPool-24 | 5 | 0.9138 | 0.5231 | 0.9143 | 0.67 |
| | 10 | 0.9138 | 0.6251 | 0.9143 | 0.7144 |
| | 15 | 0.9147 | 0.6861 | 0.9095 | 0.7396 |
| | 20 | 0.9138 | 0.7255 | 0.9143 | 0.7579 |
| MGMap noTemp R50-Former-24 | 5 | 0.9213 | 0.5195 | 0.9256 | 0.6774 |
| | 10 | 0.9213 | 0.6645 | 0.9256 | 0.7428 |
| | 15 | 0.9221 | 0.7498 | 0.9234 | 0.7801 |
| | 20 | 0.9213 | 0.8008 | 0.9256 | 0.8029 |
| MapQR noTemp R50-Former-24 | 5 | 0.9184 | 0.5865 | 0.9257 | 0.7058 |
| | 10 | 0.9184 | 0.6937 | 0.9257 | 0.7553 |
| | 15 | 0.9182 | 0.7556 | 0.9156 | 0.7781 |
| | 20 | 0.9184 | 0.7981 | 0.9257 | 0.8021 |
| GEMap noTemp R50-Former-24 | 5 | 0.9227 | 0.4976 | 0.929 | 0.6657 |
| | 10 | 0.9227 | 0.6186 | 0.929 | 0.7198 |
| | 15 | 0.9227 | 0.6973 | 0.9258 | 0.7546 |
| | 20 | 0.9227 | 0.7603 | 0.929 | 0.7851 |

paradigms, this instance point count is typically fixed at 20. The core objective is to **avoid computational instability caused by an insufficient number of random samples**, thereby ensuring the **consistency and repeatability** of the stability metric. If N is too small, random variations in individual sample points can disproportionately influence the overall stability score. At the same time, we aim to prevent **computational inefficiency** from an excessively large $N$. Our experiments confirm that the map stability metric shows no significant variation once the resampling density reaches $N \geq 100$. Therefore, balancing sampling density against computational cost, we selected $N = 100$ to achieve the best overall performance.

### C.4.3 ABLATION STUDY ON THE TEMPORAL SAMPLING INTERVAL

Temporal sampling serves as the initial step in the stability evaluation benchmark. In the default configuration, the time interval $M$ is set to 2 to assess the granularity of map changes, as delineated in Table 1. To provide a more comprehensive illustration of our evaluation framework's performance across different temporal sampling intervals, we conducted further experiments with $M$ values of 3, 5, and 10. The results are presented in Tables 10, 11, and 12 respectively. All other experimental settings remain consistent with Table 1.

The ablation studies reveal several important patterns regarding temporal stability assessment. First, as the temporal interval $M$ increases from 2, 3, 5, 10, most models exhibit a progressive decline in stability scores across all metrics, particularly in Localization Stability. This pattern is consistent across different architectural paradigms and demonstrates the challenge of maintaining consistency over longer time horizons. For instance, MapTR (Liao et al., 2022) with camera-only input shows a reduction in mAS from 71.6 ($M$=2) to 61.9 ($M$=10), primarily driven by decreasing Localization Stability. Similar trends are observed for other non-temporal models, with BeMapNet (Qiao et al., 2023) maintaining superior Presence Stability but experiencing significant Localization Stability degradation from 65.8 ($M$=2) to 41.6 ($M$=10).

Table 9: Ablation Study on the Resampling Density $N$.

| Model | $N$ | Presence | Loc | Shape | mAS |
|---|---|---|---|---|---|
| MapTRv1 noTemp R18-GKT-24 | 30 | 0.88 | 0.7535 | 0.8862 | 0.7318 |
| | 50 | 0.8776 | 0.7499 | 0.8852 | 0.7283 |
| | 100 | 0.8794 | 0.7541 | 0.8867 | 0.7317 |
| | 150 | 0.8783 | 0.7533 | 0.8862 | 0.73 |
| | 200 | 0.8789 | 0.7524 | 0.887 | 0.7305 |
| MapTRv1 noTemp R50-GKT-24 | 30 | 0.9107 | 0.6534 | 0.9078 | 0.7145 |
| | 50 | 0.912 | 0.6544 | 0.9063 | 0.7158 |
| | 100 | 0.9111 | 0.6479 | 0.9052 | 0.7112 |
| | 150 | 0.9107 | 0.6473 | 0.9087 | 0.7119 |
| | 200 | 0.911 | 0.6488 | 0.905 | 0.7113 |
| MapTRv1 Temp R50-GKT-24 | 30 | 0.8964 | 0.5663 | 0.8973 | 0.6624 |
| | 50 | 0.8861 | 0.5974 | 0.8928 | 0.6657 |
| | 100 | 0.8859 | 0.5769 | 0.8957 | 0.6582 |
| | 150 | 0.8965 | 0.5702 | 0.8971 | 0.6625 |
| | 200 | 0.8967 | 0.569 | 0.8959 | 0.6614 |
| MapTRv1 Temp R50-BevFormer-24 | 30 | 0.8839 | 0.7169 | 0.9139 | 0.7284 |
| | 50 | 0.9035 | 0.6947 | 0.9121 | 0.73 |
| | 100 | 0.9012 | 0.7326 | 0.9142 | 0.7481 |
| | 150 | 0.902 | 0.7359 | 0.9146 | 0.7502 |
| | 200 | 0.903 | 0.7344 | 0.9146 | 0.7507 |
| MapTRv2 noTemp R50-BevPool-24 | 30 | 0.9156 | 0.6714 | 0.914 | 0.7342 |
| | 50 | 0.9147 | 0.6861 | 0.9095 | 0.7396 |
| | 100 | 0.9149 | 0.6853 | 0.9149 | 0.7414 |
| | 150 | 0.9147 | 0.6868 | 0.9152 | 0.7426 |
| | 200 | 0.9145 | 0.6848 | 0.9147 | 0.7411 |
| MGMap noTemp R50-Former-24 | 30 | 0.9188 | 0.7437 | 0.9249 | 0.7756 |
| | 50 | 0.9221 | 0.7498 | 0.9234 | 0.7801 |
| | 100 | 0.9217 | 0.7492 | 0.925 | 0.7802 |
| | 150 | 0.9212 | 0.7472 | 0.925 | 0.7789 |
| | 200 | 0.922 | 0.7476 | 0.9253 | 0.7801 |
| MapQR noTemp R50-Former-24 | 30 | 0.9184 | 0.7559 | 0.9247 | 0.7839 |
| | 50 | 0.9182 | 0.7556 | 0.9156 | 0.7781 |
| | 100 | 0.918 | 0.7539 | 0.9263 | 0.7826 |
| | 150 | 0.9173 | 0.7532 | 0.9261 | 0.7818 |
| | 200 | 0.918 | 0.7541 | 0.9259 | 0.7825 |
| GEMap noTemp R50-Former-24 | 30 | 0.9144 | 0.6985 | 0.9277 | 0.752 |
| | 50 | 0.9227 | 0.6973 | 0.9258 | 0.7546 |
| | 100 | 0.922 | 0.693 | 0.931 | 0.7499 |
| | 150 | 0.9233 | 0.6916 | 0.9315 | 0.7504 |
| | 200 | 0.9232 | 0.6937 | 0.9309 | 0.7504 |

Table 10: Ablation Study on the Temporal Sampling Interval ($M$=3)

| Method | Venue | Temp | Modal | BEV Encoder | Epoch | mAP↑ | Presence↑ | Loc↑ | Shape↑ | mAS↑ |
|---|---|---|---|---|---|---|---|---|---|---|
| MapTR | ICLR'23 | ✗ | C | GKT | 24 | 44.1 | 89.3 | 59.9 | 90.6 | 67.6 |
| MapTR | ICLR'23 | ✗ | C & L | GKT | 24 | 62.8 | 91.0 | 69.3 | 91.8 | 73.7 |
| BeMapNet | CVPR'23 | ✗ | C | IPM-PE | 30 | 61.4 | 100.0 | 58.8 | 97.7 | 78.2 |
| PivotNet | ICCV'23 | ✗ | C | PersFormer | 30 | 57.1 | 100.0 | 45.2 | 98.3 | 71.7 |
| MapTRv2 | IJCV'24 | ✗ | C | BEVPool | 24 | 61.4 | 90.6 | 59.5 | 91.8 | 69.0 |
| GeMap | ECCV'24 | ✗ | C | BEVFormer-1 | 24 | 51.3 | 90.9 | 66.6 | 92.9 | 73.3 |
| MGMap | CVPR'24 | ✗ | C | BEVFormer-1 | 24 | 57.9 | 91.8 | 68.8 | 92.3 | 74.4 |
| MapQR | ECCV'24 | ✗ | C | BEVFormer-3 | 24 | 66.4 | 89.4 | 66.9 | 91.1 | 73.4 |
| MapTR | ICLR'23 | ✓ | C | GKT | 24 | 51.3 | 86.8 | 55.2 | 89.1 | 62.9 |
| StreamMapNet | WACV'24 | ✓ | C | BEVFormer-1 | 30 | 63.3 | 96.9 | 96.6 | 95.8 | 93.2 |
| MapTracker | ECCV'24 | ✓ | C | BEVFormer-2 | 72 | 75.95 | 93.7 | 96.2 | 93.4 | 88.7 |
| HRMapNet | ECCV'24 | ✓ | C | BEVFormer-1 | 24 | 67.2 | 91.2 | 70.4 | 92.2 | 75.4 |

Table 11: Ablation Study on the Temporal Sampling Interval ($M$=5)

| Method | Venue | Temp | Modal | BEV Encoder | Epoch | mAP↑ | Presence↑ | Loc↑ | Shape↑ | mAS↑ |
|---|---|---|---|---|---|---|---|---|---|---|
| MapTR | ICLR'23 | ✗ | C | GKT | 24 | 44.1 | 89.0 | 64.7 | 90.0 | 68.8 |
| MapTR | ICLR'23 | ✗ | C & L | GKT | 24 | 62.8 | 89.4 | 69.2 | 91.0 | 72.1 |
| BeMapNet | CVPR'23 | ✗ | C | IPM-PE | 30 | 61.4 | 100.0 | 50.0 | 97.5 | 73.7 |
| PivotNet | ICCV'23 | ✗ | C | PersFormer | 30 | 57.1 | 100.0 | 41.5 | 98.3 | 70.0 |
| MapTRv2 | IJCV'24 | ✗ | C | BEVPool | 24 | 61.4 | 89.0 | 52.8 | 90.8 | 64.8 |
| GeMap | ECCV'24 | ✗ | C | BEVFormer-1 | 24 | 51.3 | 88.8 | 61.7 | 91.6 | 69.3 |
| MGMap | CVPR'24 | ✗ | C | BEVFormer-1 | 24 | 57.9 | 90.0 | 67.1 | 91.5 | 71.9 |
| MapQR | ECCV'24 | ✗ | C | BEVFormer-3 | 24 | 66.4 | 88.1 | 59.9 | 89.6 | 67.4 |
| MapTR | ICLR'23 | ✓ | C | GKT | 24 | 51.3 | 85.0 | 55.8 | 90.3 | 61.8 |
| HRMapNet | ECCV'24 | ✓ | C | BEVformer-1 | 24 | 67.2 | 89.6 | 70.0 | 92.1 | 73.8 |

Table 12: Ablation Study on the Temporal Sampling Interval ($M$=10)

| Method | Venue | Temp | Modal | BEV Encoder | Epoch | mAP↑ | Presence↑ | Loc↑ | Shape↑ | mAS↑ |
|---|---|---|---|---|---|---|---|---|---|---|
| MapTR | ICLR'23 | ✗ | C | GKT | 24 | 44.1 | 88.0 | 47.4 | 90.1 | 61.9 |
| MapTR | ICLR'23 | ✗ | C & L | GKT | 24 | 62.8 | 89.0 | 58.2 | 90.9 | 66.5 |
| BeMapNet | CVPR'23 | ✗ | C | IPM-PE | 30 | 61.4 | 100.0 | 41.6 | 97.5 | 69.5 |
| PivotNet | ICCV'23 | ✗ | C | PersFormer | 30 | 57.1 | 100.0 | 28.3 | 98.9 | 63.6 |
| MapTRv2 | IJCV'24 | ✗ | C | BEVPool | 24 | 61.4 | 82.5 | 49.9 | 90.3 | 58.8 |
| GeMap | ECCV'24 | ✗ | C | BEVFormer-1 | 24 | 51.3 | 85.4 | 61.2 | 92.3 | 65.6 |
| MGMap | CVPR'24 | ✗ | C | BEVFormer-1 | 24 | 57.9 | 89.7 | 57.5 | 92.4 | 68.1 |
| MapQR | ECCV'24 | ✗ | C | BEVFormer-3 | 24 | 66.4 | 84.9 | 43.8 | 95.4 | 59.0 |
| MapTR | ICLR'23 | ✓ | C | GKT | 24 | 51.3 | 92.8 | 41.0 | 92.8 | 62.3 |
| HRMapNet | ECCV'24 | ✓ | C | BEVFormer-1 | 24 | 67.2 | 83.8 | 55.8 | 92.8 | 62.4 |

The comparative analysis reveals distinctive robustness characteristics across representation paradigms. Models with inherent temporal modeling capabilities, such as StreamMapNet (Yuan et al., 2024) and MapTracker (Chen et al., 2024), demonstrate remarkable resilience to increasing temporal intervals. StreamMapNet maintains exceptional stability with mAS of 93.2 at $M$=3, significantly outperforming non-temporal counterparts. This performance advantage is particularly pronounced in Localization Stability, where temporal models consistently exceed 96.0 even at larger intervals, compared to the substantial degradation observed in static models.

The studies also reveal paradigm-specific sensitivity patterns. Geometric-prior-based models like BeMapNet (Qiao et al., 2023) and PivotNet (Ding et al., 2023) maintain perfect Presence Stability across all intervals but exhibit considerable vulnerability in Localization Stability. In contrast, learning-based BEV representation models show more balanced degradation across stability dimensions. The performance variations across intervals provide additional evidence that accuracy (mAP) and stability (mAS) represent independent evaluation dimensions, as models with comparable mAP scores exhibit dramatically different stability characteristics under extended temporal intervals.

These findings underscore the importance of evaluating temporal stability across multiple time scales, as different representation paradigms exhibit distinct degradation patterns. The comprehensive interval analysis reinforces our central thesis that temporal stability constitutes a fundamental performance dimension that requires explicit consideration in online HD mapping system design and evaluation.

# D COMPREHENSIVE EVALUATION RESULTS WITH FURTHER DETAILS

We present a detailed analysis of the performance of various online HD mapping models in terms of both accuracy (mAP) and temporal stability (mAS), based on the comprehensive results summarized in Tables 13 - Table 22. Our analysis highlights how different architectural choices, including backbone networks, temporal modeling, sensor modalities, and BEV encoders, affect these two critical performance dimensions.

Table 13: Evaluation Results of MapTR

| Backbone | Temp | Modal | BEV Encoder | Epoch | mAP↑ | Presence↑ | Loc↑ | Shape↑ | mAS↑ | Parameters↓ |
|----------|------|-------|-------------|-------|------|-----------|------|--------|------|-------------|
| R18 | ✗ | C | GKT | 24 | 32.4 | 87.8 | 75.0 | 88.5 | 72.8 | **15.4M** |
| R18 | ✗ | C | GKT | 110 | 45.5 | 86.0 | 71.7 | **94.8** | 71.8 | **15.4M** |
| R50 | ✗ | C | GKT | 24 | 44.1 | **91.2** | 65.4 | 90.6 | 71.6 | 36.2M |
| R50 | ✗ | C | GKT | 110 | 50.5 | 89.8 | 63.2 | 91.0 | 68.2 | 36.2M |
| R50 | ✗ | C | BEVFormer-1 | 24 | 41.6 | 89.6 | 69.7 | 90.6 | 71.3 | 36.3M |
| R50 | ✗ | C | BEVPool | 24 | 50.1 | 89.3 | 69.8 | 88.5 | 71.9 | 32.3M |
| R50 | ✓ | C | GKT | 24 | 51.3 | 88.6 | 59.7 | 89.3 | 66.6 | 36.2M |
| R50 | ✓ | C | BEVFormer-1 | 24 | 53.3 | 90.4 | 69.5 | 91.2 | 73.0 | 36.3M |
| R50 & Sec. | ✗ | C & L | GKT | 24 | **62.8** | 90.1 | **75.2** | 90.8 | **74.9** | 40.1M |

## D.1 COMPREHENSIVE ANALYSIS TOWARDS MAPTR

As presented in Table 13, the extensive variants of MapTR (Liao et al., 2022) provide a controlled setting to dissect how distinct representation paradigms influence model behavior. By altering key components while holding the core architecture constant, we can isolate their effects on both accuracy (mAP) and temporal stability (mAS).

**Temporal Fusion.** The effect of incorporating temporal fusion is not uniform but is mediated by the underlying BEV representation. When applied to the GKT-based (Chen et al., 2022) representation, temporal fusion disrupts its core strength. Presence Stability drops from 91.2 to 88.6, and Localization Stability plummets from 65.4 to 59.7, leading to a significant decrease in mAS (71.6 to 66.6). This indicates that the representation formed by GKT is not easily aligned or integrated across time; the temporal module may introduce noise rather than useful context. Based on our experimental data and model architecture analysis, **the core reason lies in the fundamental incompatibility between the static spatial features generated by the GKT encoder and the dynamic representations required for temporal fusion.**

Specifically, the decline in stability stems from the following three interrelated underlying mechanisms:

1. **The feature space solidifies the static scene representation, making dynamic alignment challenging.** The core strength of GKT lies in its efficient construction of single-frame BEV features through geometric priors. However, this makes its features highly dependent on the instantaneous coordinate system at the time of construction. When historical frame features are introduced, due to the cumulative error of ego-motion, these statically generated features from different coordinate systems cannot be precisely aligned in the BEV space. **This inherent misalignment at the feature level directly leads to representational confusion after fusion**, thereby causing detection jitter (drop in Presence stability) and positional jumps (drop in Localization stability) of map elements.

2. **Temporal aggregation of localized fine-grained features amplifies high-frequency jitter.** GKT excels at extracting local, high-resolution geometric details. However, when a simple temporal fusion module (e.g., basic convolution or attention mechanisms) attempts to aggregate these highly localized features, it **not only fails to smooth out their inherent inter-frame variations but even amplifies such high-frequency jitter**. This manifests as instability in the position and shape of map elements after decoding.

3. **The training objective optimized for single-frame performance conflicts with that of the temporal module.** As a non-temporal model, MapTR's GKT encoder and decoder are strictly optimized to achieve optimal accuracy on individual frames. When a fusion module without temporal consistency constraints is introduced, the entire network falls into an optimization dilemma: the single-frame supervision signal forces the features to maintain overfitting to instantaneous observations, while the newly added temporal parameters, lacking explicit guidance, **disrupt the originally well-optimized feature distribution for single frames**. This ultimately leads to performance degradation in the temporal dimension.

The validity of this analysis is **confirmed unequivocally** by our controlled experiments on MapTR-BEVFormer. As shown in Table 4 of the paper, BEVFormer inherently employs deformable atten-

tion for temporal modeling, meaning its **features are designed from the outset for cross-frame alignment and fusion**. Consequently, when a temporal fusion module is incorporated, its features can be integrated smoothly and consistently, leading to simultaneous improvements in both accuracy and stability (mAP increased from 41.6 to 53.3, while mAS rose from 71.3 to 73.0). This demonstrates that temporal fusion acts as a complementary enhancement when applied to representations that are already designed for spatiotemporal modeling. This demonstrates that temporal fusion is most effective when the base representation is inherently compatible with processing sequential data. The effectiveness of temporal fusion is not a standalone property but is contingent on the representational capacity of the BEV encoder. It amplifies the capabilities of a temporally-aware representation (BEVFormer) but can degrade the performance of a primarily spatially-focused one (GKT). Therefore, temporal fusion serves as a force multiplier for representations already predisposed to temporal modeling, but can be detrimental to those that are not.

**2D Backbone.** Changing the 2D backbone from ResNet-18 to ResNet-50 shifts the model's representational focus towards more complex visual patterns. This shift has a clear effect: it consistently improves mAP (32.4 to 44.1) by leveraging higher-capacity feature extraction. However, this comes with a redistribution of stability properties: Presence Stability improves (87.8 to 91.2), but Localization Stability worsens significantly (75.0 to 65.4). The deeper network appears to learn a representation that is more sensitive to semantic content but potentially more susceptible to per-frame variations in texture or lighting, which can harm geometric consistency. The backbone network influences the type of features that form the basis of the map representation. More powerful backbones enhance semantic discrimination but can introduce high-frequency noise that undermines geometric stability, suggesting that representations favoring stability may require features that are invariant to superficial appearance changes.

Table 14: Evaluation Results of BeMapNet

| Backbone | Temp | Modal | BEV Encoder | Epoch | mAP↑ | Presence↑ | Loc↑ | Shape↑ | mAS↑ | Parameters↓ |
|---|---|---|---|---|---|---|---|---|---|---|
| Effb0 | ✗ | C | IPM-PE | 30 | 60.7 | **100.0** | **67.9** | 97.9 | **82.9** | 55.4M |
| R50 | ✗ | C | IPM-PE | 30 | 61.4 | **100.0** | 65.8 | 97.9 | 81.9 | 73.8M |
| SwinT | ✗ | C | IPM-PE | 30 | 64.1 | **100.0** | 62.8 | 98.0 | 80.4 | 79.6M |
| R50 | ✗ | C | IPM-PE | 110 | 66.2 | **100.0** | 62.1 | **98.2** | 80.2 | 73.8M |
| SwinT | ✗ | C | IPM-PE | 110 | **68.3** | **100.0** | 64.0 | **98.2** | 81.1 | 79.6M |

Table 15: Evaluation Results of PivotNet

| Backbone | Temp | Modal | BEV Encoder | Epoch | mAP↑ | Presence↑ | Loc↑ | Shape↑ | mAS↑ | Parameters↓ |
|---|---|---|---|---|---|---|---|---|---|---|
| Effb0 | ✗ | C | PersFormer | 30 | 57.8 | **100.0** | 71.8 | 97.2 | 84.5 | **17.1M** |
| R50 | ✗ | C | PersFormer | 30 | 57.1 | **100.0** | 71.4 | 97.2 | 84.3 | 41.2M |
| Swin-T | ✗ | C | PersFormer | 30 | 61.6 | **100.0** | 71.6 | 97.2 | 84.4 | 44.8M |
| Swin-T | ✗ | C | PersFormer | 110 | **66.4** | **100.0** | **72.1** | **97.4** | **84.8** | 44.8M |

## D.2 In-Depth Analysis of BeMapNet and PivotNet

As shown in Table 14 and Table 15, both BeMapNet(Qiao et al., 2023) and PivotNet(Ding et al., 2023) demonstrate stable mAP and mAS performance, with BeMapNet achieving mAP scores of 60.7 to 68.3 and mAS values of 80.2 to 82.9, while PivotNet demonstrates mAP values of 57.1 to 66.4 and mAS scores of 84.3 to 84.8.This indicats that these two models are insensitive to backbone network selection and training epoch configurations. The difference lies in the fact that PivotNet achieves its highest mAS when using Swin-T(Liu et al., 2021) as the backbone, while BeMapNet attains its peak mAS value with an EfficientNet-B0(Tan & Le, 2019) backbone.

It should be specifically noted that both BeMapNet and PivotNet adopt a "dynamic vectorized sequence" representation for map encoding, which explains their consistently perfect presence metrics (100%). However, this representation format severely limits the localization stability of map instances, resulting in significantly lower performance compared to models like GeMap(Zhang et al., 2024c), MGMap(Liu et al., 2024a), and MapQR(Liu et al., 2024b).

Table 16: Evaluation Results of MapTRv2

| Backbone | Temp | Modal | BEV Encoder | Epoch | mAP↑ | Presence↑ | Loc↑ | Shape↑ | mAS↑ | Parameters↓ |
|---|---|---|---|---|---|---|---|---|---|---|
| R18 | ✗ | C | BEVPool | 24 | 57.2 | 91.0 | **73.2** | **91.2** | **75.6** | **27.9M** |
| R50 | ✗ | C | BEVPool | 24 | **61.4** | **91.5** | 68.6 | 91.0 | 74.0 | 40.6M |

## D.3 EVALUATION AND ANALYSIS OF MAPTRV2

MapTRv2 (Liao et al., 2025b) improves upon MapTR (Liao et al., 2022) with higher baseline mAP (57.2–61.4) and mAS (74.0–75.6). Interestingly, the R18 backbone achieves higher mAS (75.6) than R50 (74.0), despite a lower mAP (57.2 *vs.* 61.4), as illustrated in Table 13 and Table 16, reinforcing the independence of accuracy and stability.

Table 17: Evaluation Results of GeMap

| Backbone | Temp | Modal | BEV Encoder | Epoch | mAP↑ | Presence↑ | Loc↑ | Shape↑ | mAS↑ | Parameters↓ |
|---|---|---|---|---|---|---|---|---|---|---|
| R50 | ✗ | C | BEVFormer-1 | 24 | 51.3 | 92.3 | 69.7 | 92.6 | 75.5 | **44.1M** |
| R50 | ✗ | C | BEVFormer-1 | 110 | 62.7 | 91.1 | 67.5 | **94.5** | 74.7 | **44.1M** |
| Swin-T | ✗ | C | BEVFormer-1 | 110 | 72.0 | 92.2 | **74.9** | 93.2 | **78.1** | 50.5M |
| V2-99 | ✗ | C | BEVFormer-1 | 110 | 72.0 | 89.2 | 71.5 | 92.6 | 74.2 | 92.6M |
| V2-99(DD3D) | ✗ | C | BEVFormer-1 | 110 | **76.0** | **93.4** | 66.9 | 93.7 | 75.1 | 92.6M |
| R50 & second | ✗ | C & L | BEVFormer-1 | 110 | 66.5 | 89.1 | 66.3 | 92.7 | 71.8 | 48.0M |

## D.4 DISCUSSION ON GEMAP

As illustrated in Table 17, GeMap (Zhang et al., 2024c) presents a particularly instructive case for examining the complex relationship between accuracy and stability. The model demonstrates a strong capacity for high accuracy, with its mAP score scaling significantly from 51.3 to a top score of 76.0 when employing a powerful V2-99 (Lee et al., 2019) backbone and extended training. However, this pursuit of accuracy often introduces instability, as evidenced by its mAS scores, which range from a moderate 71.8 to a more competitive 78.1.

A critical observation is the divergent effect of LiDAR fusion. While integrating LiDAR data with a ResNet-50 (He et al., 2016) backbone yields a predictable improvement in mAP (+6.1%, from 62.7 to 66.5), it conversely leads to a decrease in mAS (-3.9%, from 74.7 to 71.8). This result challenges the conventional wisdom that more sensor data invariably leads to more robust perception. It suggests that GeMap's architecture, while effectively leveraging LiDAR for geometric precision in a single frame, may lack the necessary mechanisms to harmonize the potentially noisy or asynchronous multi-modal signals across time, leading to increased jitter or flickering.

Furthermore, the model exhibits high sensitivity to backbone design. The Swin-T (Liu et al., 2021) backbone strikes the most favorable balance, achieving the highest mAS (78.1) alongside a high mAP (72.0). In contrast, the larger V2-99 (Lee et al., 2019) backbone, despite achieving the peak mAP (76.0), produces a lower mAS (75.1). The degradation in Localization Stability (from 74.9 with Swin-T (Liu et al., 2021) to 66.9 with V2-99 (Lee et al., 2019)) is especially notable, implying that the increased representational power of the larger backbone may overfit to single frame features at the expense of temporal coherence. This pattern underscores that for stability, simply scaling up model capacity is not a sufficient strategy and may even be counterproductive without explicit temporal regularization.

Table 18: Evaluation Results of MGMap

| Backbone | Temp | Modal | BEV Encoder | Epoch | mAP↑ | Presence↑ | Loc↑ | Shape↑ | mAS↑ | Parameters↓ |
|---|---|---|---|---|---|---|---|---|---|---|
| R50 | ✗ | C | BEVFormer-1 | 24 | 58.0 | 92.2 | 75.0 | 92.3 | 78.0 | 55.9M |

## D.5 ANALYSIS FOR MGMAP

MGMap (Liu et al., 2024a) achieves a balanced profile (mAP: 58.0, mAS: 78.0) with a ResNet-50 backbone (He et al., 2016) and BEVFormer (Li et al., 2024b) encoder, as presented in Table 18. Its strong Localization and Shape Stability scores (75.0 and 92.3, respectively) suggest robustness against geometric jitter.

Table 19: Evaluation Results of MapQR

| Backbone | Temp | Modal | BEV Encoder | Epoch | mAP↑ | Presence↑ | Loc↑ | Shape↑ | mAS↑ | Parameters↓ |
|---|---|---|---|---|---|---|---|---|---|---|
| R18 | ✗ | C | BEVFormer-3 | 24 | 62.3 | 88.2 | 73.1 | 92.5 | 74.1 | **112.6M** |
| R50 | ✗ | C | BEVFormer-3 | 24 | 66.4 | 91.8 | 75.6 | 91.6 | 77.8 | 125.4M |
| R50 | ✗ | C | BEVFormer-3 | 110 | **72.6** | **92.4** | **75.9** | **96.4** | **80.3** | 125.4M |

## D.6 STUDY OF MAPQR

As shown in Table 19, MapQR (Liu et al., 2024b) shows a clear positive scaling trend: larger backbones and longer training improve both mAP (62.3 to 72.6) and mAS (74.1 to 80.3). This indicates that the model's architecture supports stable learning under increased capacity.

Table 20: Evaluation Results of StreamMapNet

| Backbone | Temp | Modal | BEV Encoder | Epoch | mAP↑ | Presence↑ | Loc↑ | Shape↑ | mAS↑ | Parameters↓ |
|---|---|---|---|---|---|---|---|---|---|---|
| R50 | ✗ | C | BEVFormer-1 | 30 | 51.7 | 87.0 | 97.8 | 95.1 | 83.8 | 56.0M |
| R18 | ✓ | C | BEVFormer-1 | 30 | 27.8 | 87.1 | 98.4 | 94.6 | 85.0 | **42.5M** |
| R50 | ✓ | C | BEVFormer-1 | 30 | **63.4** | 96.6 | 97.7 | 92.3 | 91.9 | 56.3M |
| R50 | ✓ | C | BEVFormer-1 | 24 | 51.2 | **97.0** | **98.5** | **96.1** | **94.4** | 56.3M |

## D.7 TOWARDS A COMPREHENSIVE ANALYSIS OF STREAMMAPNET

StreamMapNet (Yuan et al., 2024) stands out as the paradigm for temporally stable online mapping, achieving the highest mAS scores in our benchmark, ranging from 83.8 to an exceptional 94.4. This performance is primarily driven by its native temporal architecture, which is explicitly designed to model consistency across frames.

The most striking feature of StreamMapNet is its near-perfect Localization Stability (97.7–98.5), the highest among all models, as shown in Table 1 and Table 20. This indicates an exceptional ability to suppress the positional jitter of map elements over time, a critical factor for downstream planning tasks. The analysis clearly shows that temporal fusion is not merely an optional add-on but the core determinant of its performance. Enabling temporal modeling (comparing the R50, ✓ vs. R50, ✗ configurations) results in a dramatic improvement in both mAP (+22.6%, from 51.7 to 63.4) and mAS (+9.7%, from 83.8 to 91.9). This dual improvement confirms that effectively leveraging historical context can simultaneously enhance per-frame accuracy and inter-frame consistency.

Table 21: Evaluation Results of MapTracker

| Backbone | Temp | Modal | BEV Encoder | Epoch | mAP↑ | Presence↑ | Loc↑ | Shape↑ | mAS↑ | Parameters↓ |
|---|---|---|---|---|---|---|---|---|---|---|
| R18 | ✗ | C | BEVFormer-2 | 72 | 62.8 | **95.3** | 97.3 | 85.9 | 87.4 | **60.7M** |
| R50 | ✗ | C | BEVFormer-2 | 72 | 68.3 | 94.5 | 97.9 | 93.8 | 90.8 | 74.0M |
| R18 | ✓ | C | BEVFormer-2 | 48 | 69.3 | 94.8 | 98.2 | 94.8 | 91.5 | **60.7M** |
| R18 | ✓ | C | BEVFormer-2 | 72 | 71.9 | 92.9 | **98.5** | 94.8 | 89.9 | **60.7M** |
| R50 | ✓ | C | BEVFormer-2 | 48 | 73.0 | 91.7 | **98.5** | **96.0** | **91.7** | 74.0M |
| R50 | ✓ | C | BEVFormer-2 | 72 | **76.0** | 93.3 | 98.1 | 95.8 | 90.4 | 74.0M |

## D.8 A COMPREHENSIVE ANALYSIS OF MAPTRACKER

MapTracker (Chen et al., 2024) represents another strong temporal model that successfully balances state-of-the-art accuracy with high stability, which is shown in Table 1 and Table 21. It achieves the

highest overall mAP (76.0) in our benchmark while maintaining mAS scores above 87.4, peaking at 91.7.

Similar to StreamMapNet (Yuan et al., 2024), MapTracker's integration of temporal fusion ("Temp = ✓") consistently boosts mAP (e.g., from 62.8 to 69.3 for R18) while preserving high mAS. This reinforces the conclusion that architectures designed with temporal reasoning in mind from the ground up are essential for high-performance online mapping. The model also exhibits very strong Localization and Shape Stability, often exceeding 98.0 and 94.0, respectively, which is characteristic of models that effectively aggregate information over time.

However, MapTracker reveals a nuanced trade-off related to training duration. For both the R18 and R50 backbones, extending training from 48 to 72 epochs leads to a further increase in mAP but a slight decrease in mAS (e.g., R50: mAP 73.0 to 76.0, mAS 91.7 to 90.4). This pattern, which we term optimization sensitivity, suggests that as the model continues to minimize a primarily accuracy-oriented loss function, it may gradually overfit to single-frame details, thereby sacrificing some temporal smoothness. This highlights a key area for future research: the development of loss functions or regularization techniques that explicitly penalize temporal instability during training to prevent this erosion.

Table 22: Evaluation Results of HRMapNet

| Backbone | Temp | Initial Map | Modal | BEV Encoder | Epoch | mAP↑ | Presence↑ | Loc↑ | Shape↑ | mAS↑ | Parameters↓ |
|---|---|---|---|---|---|---|---|---|---|---|---|
| R50 | ✓ | ✗ | C | BEVFormer-1 | 24 | 67.2 | 92.3 | 70.5 | 91.5 | 75.9 | **47.3M** |
| R50 | ✓ | Testing Map | C | BEVFormer-1 | 24 | 73.0 | **94.9** | 71.4 | 93.0 | **78.4** | **47.3M** |
| R50 | ✓ | Training Map | C | BEVFormer-1 | 24 | **83.6** | 89.9 | **75.9** | **93.2** | 76.7 | **47.3M** |
| R50 | ✓ | ✗ | C | BEVFormer-1 | 110 | 73.5 | 90.5 | 74.1 | 92.7 | 75.9 | **47.3M** |

## D.9 DETAILED INVESTIGATION OF HRMAPNET

As presented in Table 22, HRMapNet (Zhang et al., 2024b) incorporates a distinct representation paradigm by integrating static map priors into a temporal mapping framework. The performance variations across its configurations provide critical insights into the interaction between dynamic sensory input and static prior knowledge.

The most pronounced effect is observed on single frame accuracy. Utilizing a map prior during training (e.g., linear priors (Li et al., 2024a)) yields a substantial improvement in mAP, elevating the score from 67.2 to 83.6. This result indicates that the model's representation effectively internalizes the structural constraints provided by the high quality offline map, leading to superior geometric precision in individual frames.

In contrast, the impact of map priors on temporal stability is more complex and less direct. The configuration employing a prior only during testing achieves the highest mAS of 78.4 and the highest Presence Stability of 94.9. This suggests that an externally provided prior can serve as a stabilizing reference during inference, enhancing detection consistency without being fully baked into the model parameters.

However, when the model is trained with the map prior, a different pattern emerges. While this configuration achieves the highest mAP, its mAS of 76.7 is lower than the testing prior variant. Notably, its Presence Stability decreases to 89.9. This indicates that deep integration of the static prior during training may lead to a representation that is overly reliant on persistent features, potentially at the expense of robustness to real world variations encountered in a temporal sequence. The model may become less adept at handling cases where the prior is imperfect or where dynamic scenes deviate from the stored map.

Furthermore, extending training to 110 epochs without any initial map prior improves mAP to 73.5 but leaves mAS unchanged at 75.9. This stability saturation effect underscores that prolonged training on a single frame accuracy objective has diminishing returns for temporal consistency. The gain in stability achieved through the intelligent use of a testing time prior surpasses that achieved by simply training the baseline model longer.

In summary, HRMapNet demonstrates that static map priors constitute a powerful representation for enhancing perceptual accuracy. Their utility for improving temporal stability, however, is contingent

on the method of integration. A prior used as a dynamic guidance signal at inference can bolster consistency, whereas deeply embedding the prior into the model through training may introduce a trade off, favoring accuracy over stability. This highlights that the effective fusion of dynamic and static representations remains a key challenge for robust online mapping.

### D.10    SUMMARY OF COMPREHENSIVE EVALUATION

The comprehensive evaluation of ten representative online HD mapping models demonstrates that different representation paradigms induce distinct performance characteristics along the accuracy-stability spectrum. Our analysis reveals that these two performance dimensions are independently influenced by specific architectural choices and their underlying representational biases.

Models incorporating strong geometric priors, such as BeMapNet (Qiao et al., 2023) and PivotNet (Ding et al., 2023), achieve exceptional Presence Stability due to their structure-aware representations. In contrast, architectures based on learned view transformations like BEVFormer (Li et al., 2024b) exhibit superior Localization Stability, benefiting from their spatially coherent bird's eye view representations.

Temporal modeling effectiveness shows fundamental dependence on representational compatibility. Architectures with native temporal designs demonstrate that explicit sequence modeling produces the highest stability scores. However, the integration of temporal modules requires careful alignment with the base representation, as evidenced by the varied outcomes when adding temporal components to different BEV encoders.

Multi-modal integration exhibits model-dependent effects on stability. While sensor fusion generally enhances accuracy, its impact on temporal consistency varies across architectures, indicating that effective multi-modal representation requires specialized design beyond simple feature combination.

The relationship between model capacity and performance reveals consistent patterns. Larger backbones produce substantial accuracy gains but yield inconsistent effects on stability, suggesting that representational capacity alone cannot address temporal consistency requirements.

The comparison between static priors and dynamic modeling highlights their complementary roles. While static priors significantly boost accuracy, dynamic temporal modeling proves essential for achieving temporal stability, indicating that these two approaches address distinct aspects of the mapping problem, and future priors can further enhance ahead-aware online mapping (Li et al., 2025b).

These findings collectively suggest that accuracy and stability are governed by different aspects of representation design. This understanding points to the need for future architectures that can simultaneously support high-fidelity spatial representation and robust temporal consistency through integrated design principles.

### D.11    ON THE SYNERGISTIC USE OF mAS AND mAP

We would like to clarify that the primary intention behind designing mAS was never to provide a standalone metric for evaluating map construction performance in isolation. Instead, it aims to reveal the "temporal stability" dimension that conventional mAP overlooks.

Crucially, any model employing strategies like copy pasting or blind temporal smoothing would inevitably suffer a significant drop in precision metrics like mAP, even if it achieved a high localization stability score. This is because both copy pasting and heavily smoothing predictions across frames introduce substantial deviations from the ground truth for the affected instances.

Therefore, when mAS is examined in conjunction with mAP, such pseudo stable models are easily exposed. They exhibit an anomalous profile of high mAS but very low mAP. Identifying and cautioning against this exact type of behavior is a key objective of our evaluation framework.

Thus, the value of mAS lies in its synergistic use with inference accuracy metrics like mAP. Together, they define a more comprehensive trustworthiness evaluation quadrant, guiding the community to pursue models that achieve both high accuracy and high stability, rather than over optimizing for either metric alone.

### D.12 COMPARISON WITH BASELINE METHODS

**Comparison with Chamfer Distance**   First, regarding "inter-frame Chamfer Distance," our proposed **mAS is a comprehensive evaluation metric**, whereas traditional geometric measures like Chamfer Distance typically capture only **a single dimension of stability**. As explained in Section 3.4 of the paper, mAS consists of three dimensions: **Presence Stability, Localization Stability, and Shape Stability**. Chamfer Distance essentially measures the overall positional deviation between two point sets, roughly corresponding to the "Localization Stability" in our framework. However, it **completely fails to detect "presence flickering"**, the critical instability where map elements appear and disappear intermittently, and **cannot effectively quantify temporal changes in geometric shape, such as curvature**. A model may exhibit low Chamfer Distance (i.e., minimal positional jitter), but if its predicted lane boundaries frequently appear or vanish (poor presence stability), it remains hazardous and unreliable for downstream planning tasks. mAS is specifically designed to comprehensively diagnose these different types of instabilities.

**Comparison with Temporal Consistency Loss**   Second, regarding "temporal consistency loss," **mAS and "temporal consistency loss" serve different purposes and are complementary**. The "temporal consistency loss" you mentioned is an **optimization objective used during model training**, aimed at penalizing temporal jitter in model outputs through regularization terms. In contrast, mAS is an **offline evaluation metric for assessing model performance**. Their roles and applications are fundamentally distinct. An optimized loss function does not automatically constitute a comprehensive evaluation system. In fact, our mAS metric **can provide precise evaluation guidance and feedback for designing and optimizing such temporal loss functions**. For example, by analyzing the three sub-metrics of mAS, we can determine whether a temporal loss function effectively improves localization stability or inadvertently sacrifices presence stability.

**Advantages of mAS Design**   Furthermore, **the design of mAS is more aligned with the safety requirements of autonomous driving**. The stability issues in online HD map construction manifest their risks in multiple aspects. As demonstrated in Figures 2, 6, and 7 of the paper, as well as the downstream task impact analysis in Appendix E.1, **the disruption caused by presence flickering and shape mutations to the planning module is often as critical as minor positional jitter**. The comprehensiveness of mAS enables it to **simultaneously alert against these three different types of risks**, whereas single geometric metrics like Chamfer Distance overlook key safety-related information.

In summary, we do not dismiss the value of traditional geometric metrics but highlight their **inherent limitations in evaluating complex temporal stability**. The proposal of mAS aims to establish a **more comprehensive and safety-aware evaluation benchmark**, capable of uncovering instability patterns that are overlooked by conventional methods.

## E   SUPPLEMENTAL VISUALIZATION AND ANALYSIS

### E.1   IMPACT OF UNSTABLE MAP PREDICTIONS ON DOWNSTREAM TASKS

In this part, we visualize how temporal fluctuations in online mapping predictions impact downstream tasks, as shown in Figures 6 and 7.

Figure 6 illustrates the effect of map changes between consecutive frames on these tasks.

In Scenario A at time t, the ego vehicle fails to detect an intersection ahead due to occlusion by a leading vehicle, leading it to predict the vehicle will turn left. At time t+1, the ego vehicle successfully identifies the intersection, resulting in a corrected prediction of the leading vehicle's trajectory.

In Scenario B, occlusion by three vehicles directly ahead prevents the ego vehicle from detecting the road boundary behind them, causing it to plan a straight path that would collide with the curb. At time t+1, after moving forward, the ego vehicle observes the previously hidden map element and correctly plans a right turn.

In Scenario C at time t, the vehicle does not observe the crosswalk ahead and plans to continue straight. After advancing and detecting the map element, the ego vehicle adjusts its plan accordingly.

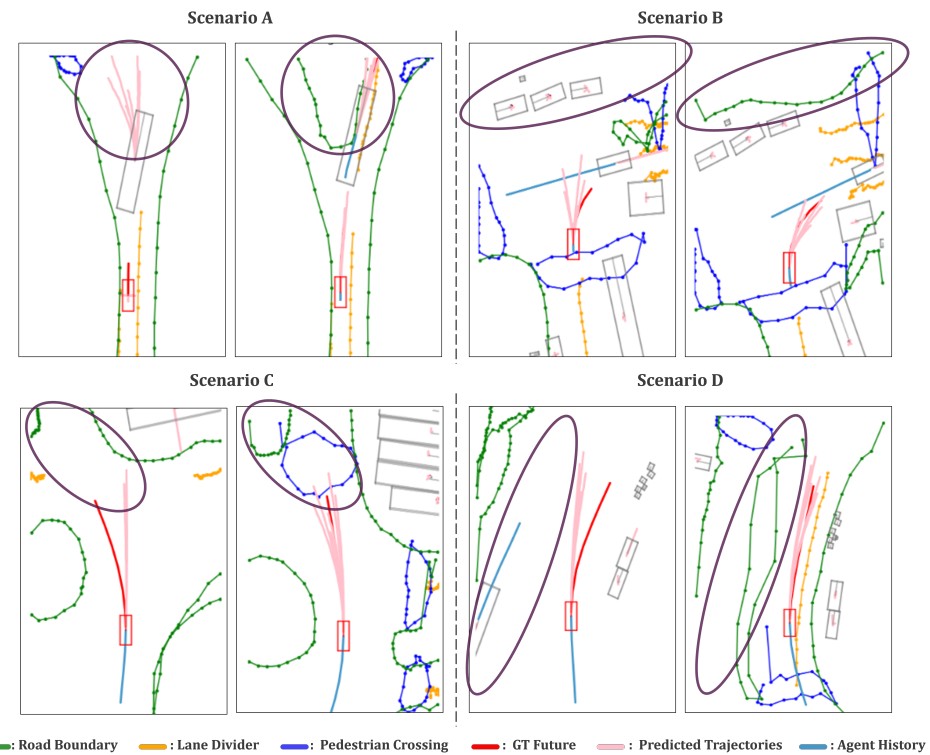

Figure 6: **Impact of Temporal Inconsistency in Map Element Presence on Downstream Tasks**

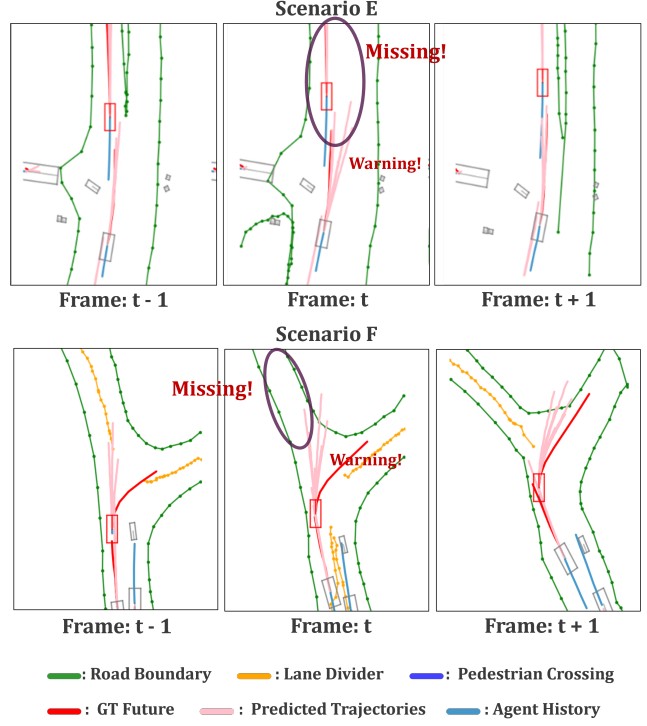

Figure 7: **Impact of Flickering in Predicted Map Elements on Downstream Tasks.**

In Scenario D's initial frame, occlusion by other vehicles prevents the ego vehicle from predicting the road boundary to its left, leading to a straight path plan that risks a curb collision. Upon detecting the map element on the left in the next frame, it adjusts its trajectory for safe navigation.

Based on the analysis of Figure 6, a key conclusion can be drawn: map elements are critical for autonomous systems to perform downstream tasks such as trajectory prediction and planning, including in VLA-based driving systems (Cai et al., 2025). Temporal instability in the perception of these elements can lead to significantly different and potentially unsafe predictions and plans.

**Quantitative Evaluation on Trajectory Prediction Tasks.** To quantitatively assess the impact of map stability on downstream tasks, we integrate map predictions from different models into trajectory prediction pipelines. We evaluate two representative trajectory prediction methods: HiVT (Zhou et al., 2022), which leverages map context for multi-agent trajectory forecasting, and DenseTNT (Gu et al., 2021), which uses map topology for goal-oriented trajectory prediction.

Table 23: Downstream Task Performance: Trajectory Prediction with DenseTNT.

| Methods | config | | | | | DenseTNT | | | mAS ↑ |
|---|---|---|---|---|---|---|---|---|---|
| | modality | Temporal | Backbone | BEV Encoder | Epoch | minADE ↓ | minFDE ↓ | MR ↓ | |
| MapTR | Only C | × | R50 | GKT | 24 | 1.1228 | 2.2151 | 0.3726 | 71.6 |
| StreamMapNet | Only C | × | R50 | BEVFormer | 24 | 1.0639 | 2.1430 | 0.3412 | 83.8 |

Table 24: Downstream Task Performance: Trajectory Prediction with HiVT.

| Methods | config | | | | | HiVT | | | mAS ↑ |
|---|---|---|---|---|---|---|---|---|---|
| | modality | Temporal | Backbone | BEV Encoder | Epoch | minADE ↓ | minFDE ↓ | MR ↓ | |
| MapTR | Only C | × | R50 | GKT | 24 | 0.4015 | 0.8404 | 0.0960 | 71.6 |
| StreamMapNet | Only C | × | R50 | BEVFormer | 24 | 0.3963 | 0.8223 | 0.0923 | 83.8 |

The experimental results, summarized in Tables 24 and 23, demonstrate a clear correlation between map stability (mAS) and downstream task performance. StreamMapNet, with its superior temporal stability (mAS: 83.8), consistently outperforms MapTR (mAS: 71.6) across all evaluation metrics in both trajectory prediction methods. Specifically:

- **HiVT Results:** StreamMapNet achieves lower minADE (0.3963 vs. 0.4015), minFDE (0.8223 vs. 0.8404), and Miss Rate (0.0923 vs. 0.0960), indicating more accurate and reliable trajectory predictions.
- **DenseTNT Results:** Similarly, StreamMapNet demonstrates superior performance with lower minADE (1.0639 vs. 1.1228), minFDE (2.1430 vs. 2.2151), and Miss Rate (0.3412 vs. 0.3726).

These quantitative results validate that temporal stability in map construction directly translates to improved performance in downstream autonomous driving tasks. Unstable map predictions introduce noise and inconsistency that propagate through the pipeline, degrading trajectory prediction accuracy and increasing the risk of unsafe maneuvers. This finding reinforces the critical importance of evaluating and optimizing temporal stability alongside traditional accuracy metrics.

As shown in Figure 7, we provide further visualization of how flickering map elements impact downstream tasks over time.

In Scenario E at frame t-1, the ego vehicle is proceeding normally. However, at frame t, a flicker occurs in the predicted road boundary to the right of the lead vehicle, caused by instability in the online mapping model. This leads the ego vehicle to perceive an opportunity to overtake on the right, resulting in a planning decision to steer right and attempt a pass. By frame t+1, the model correctly perceives the road boundary again, causing the ego vehicle to abort the maneuver and resume a straight path.

In Scenario F at frame t-1, the ego vehicle observes the lane divider ahead and plans a normal trajectory. At frame t, however, the predicted lane divider suddenly disappears, causing the planning module to become uncertain and unable to confidently decide between a lane change or continuing straight.

## E.2 MAP *vs.* MAS

In this section, we present additional cases where mAP proves to be a misleading indicator for evaluating temporal stability, whereas our proposed mAS correctly assesses temporal stability, as demonstrated in Figure 8 and Figure 9. These examples clearly show that mAP should not be used as a criterion for temporal stability evaluation, whereas mAS provides a more accurate assessment of temporal stability.

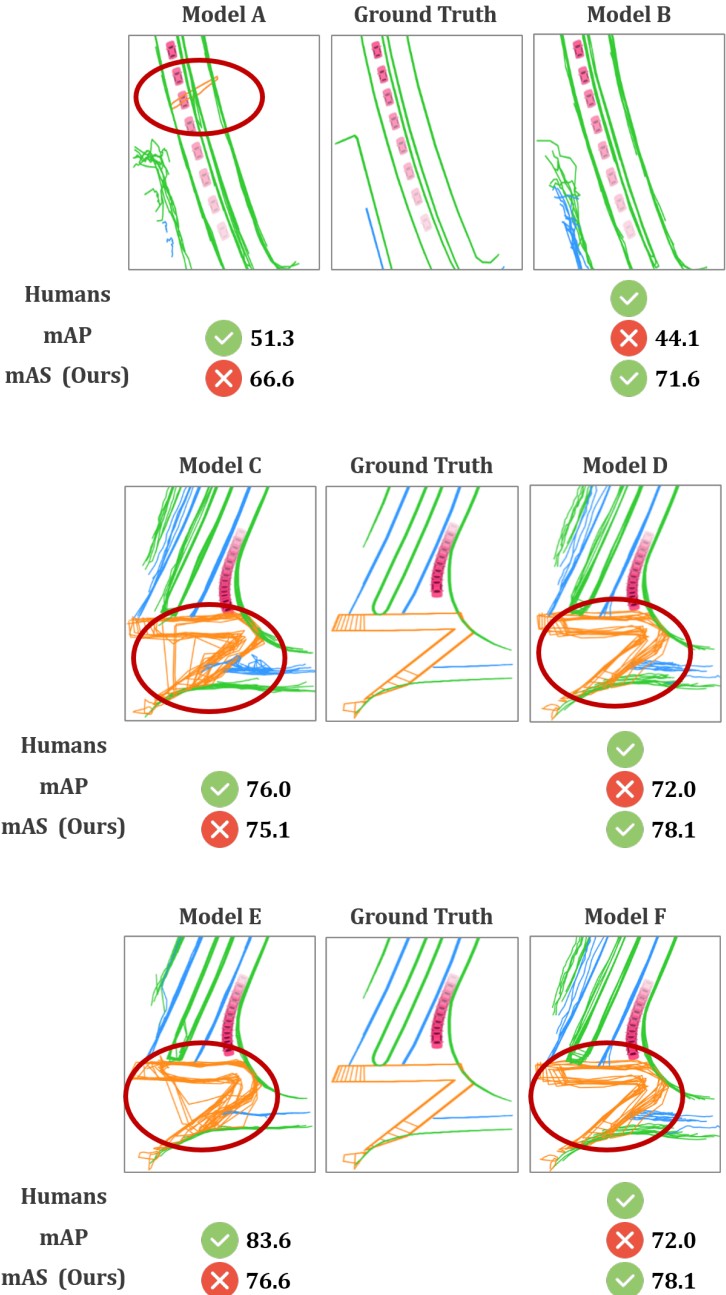

Figure 8: **Evaluating trustworthiness of online mapping models using human judgment, traditional mAP, and our mAS metric.**

As shown in Figure 8, model A represents the MapTR (Liao et al., 2022) model integrated with temporal features, employing a ResNet-50 backbone (He et al., 2016) and GKT encoder (Chen

et al., 2022), trained for 24 epochs. Model B represent the MapTR Liao et al. (2022) model without temporal features, utilizing the same ResNet-50 backbone (He et al., 2016) and GKT encoder (Chen et al., 2022), also trained for 24 epochs. Although model A achieves a relatively high mAP of 51.3, its stability is inferior to that of model B. Specifically, in the visualization results of model A, a crosswalk flickers into view on the road, and additionally, the leftmost lane divider visualized by model A flickers frequently. In contrast, although model B has a lower mAP compared to model A, it does not produce sudden flickering of other map elements in the middle of the road and is able to consistently predict the lane divider on the roadside in nearly every frame.

As illustrated in Figure 8, both model C and model D represent GeMap (Zhang et al., 2024c) models. Model C was trained for 110 epochs using the Swin-T (Liu et al., 2021) backbone network and the BEVFormer encoder. Model D was trained for 110 epochs using the V2-99 (DD3D) (Lee et al., 2019) backbone network and the BEVFormer encoder (Li et al., 2024b). Although the performance indicators of model C are superior to those of model D, our mAS evaluation indicates that the stability of model C is relatively poor. By visually comparing the outputs of the two, we find that model C occasionally detects non-existent pedestrian crossings in individual frames, which is a manifestation of poor field stability. This observation result confirms that the stability of model C is indeed weaker than that of model D, which is consistent with the mAS evaluation result.

As depicted in Figure 8, model E represents the HRMapNet (Zhang et al., 2024b) model with a training map as initial map, employing a ResNet-50 backbone (He et al., 2016) and BevFormer encoder (Li et al., 2024b), trained for 24 epochs. Model F represents the GeMap model (Zhang et al., 2024c), employing a Swin-T backbone (Liu et al., 2021) and BevFormer encoder (Li et al., 2024b), trained for 110 epochs. Model E achieves a higher mAP value than model F, yet according to our mAS metric evaluation, model E exhibits inferior stability compared to model F. A visual comparison of their inference results reveals that the crosswalks predicted by model E show more pronounced geometric jitter, while other instances remain similar between the two models. This observation confirms that model E's stability is indeed poorer than Model F's, consistent with the assessment provided by the mAS metric.

As can be seen from Figure 9, both model G and model H represent GeMap (Zhang et al., 2024c) models. Model G was trained for 110 epochs using the Swin-T backbone network (Liu et al., 2021) and the BEVFormer encoder (Li et al., 2024b), while model H was trained using the V2-99 (DD3D) backbone network (Lee et al., 2019) and the BEVFormer encoder (Li et al., 2024b). It was also trained for 110 epochs. Although the performance indicators of model G are superior to those of model H, our mAS evaluation indicates that the stability of model G is relatively poor. Through visual comparison of the outputs of the two, we find that model G has significant spatial offset and morphological fluctuation in the prediction of road boundary lines in consecutive frames, which is a manifestation of poor field stability. This observation result confirms that the stability of Model G is indeed weaker than that of model H, which is consistent with the mAS evaluation result.

As illustrated in Figure 9, both model I and model J represent GeMap models (Zhang et al., 2024c) . Model I was trained for 110 epochs using the Swin-T backbone network (Liu et al., 2021) and the BEVFormer encoder (Li et al., 2024b). Model J was trained for 110 epochs using the V2-99 (DD3D) backbone network (Lee et al., 2019) and the BEVFormer encoder (Li et al., 2024b). Model I achieves a higher mAP value than model J, yet according to our mAS metric evaluation, model I exhibits inferior stability compared to model J. Model J demonstrates superior delineation in the demarcated regions, with map instances exhibiting more precise spatial localization, while maintaining comparable performance to other models in remaining areas. This observed enhancement in output quality confirms model J's higher stability, which aligns consistently with the mAS evaluation results.

As shown in Figure 9, model K represents the MapTR (Liao et al., 2022) model integrated with temporal features, employing a ResNet-50 backbone (He et al., 2016) and GKT encoder (Chen et al., 2022), trained for 24 epochs. Model L represent the MapTR Liao et al. (2022) model without temporal features, utilizing the same ResNet-50 backbone (He et al., 2016) and GKT encoder (Chen et al., 2022), also trained for 24 epochs. Model K achieves a higher mAP value than model L, yet according to our mAS metric evaluation, model K exhibits inferior stability compared to model L. Model L produces clearer map results in the demarcated areas with more accurate spatial positioning of map instances, while maintaining similar performance to other models in remaining regions. This demonstrates Model L's superior stability, which is consistent with the mAS evaluation outcomes.

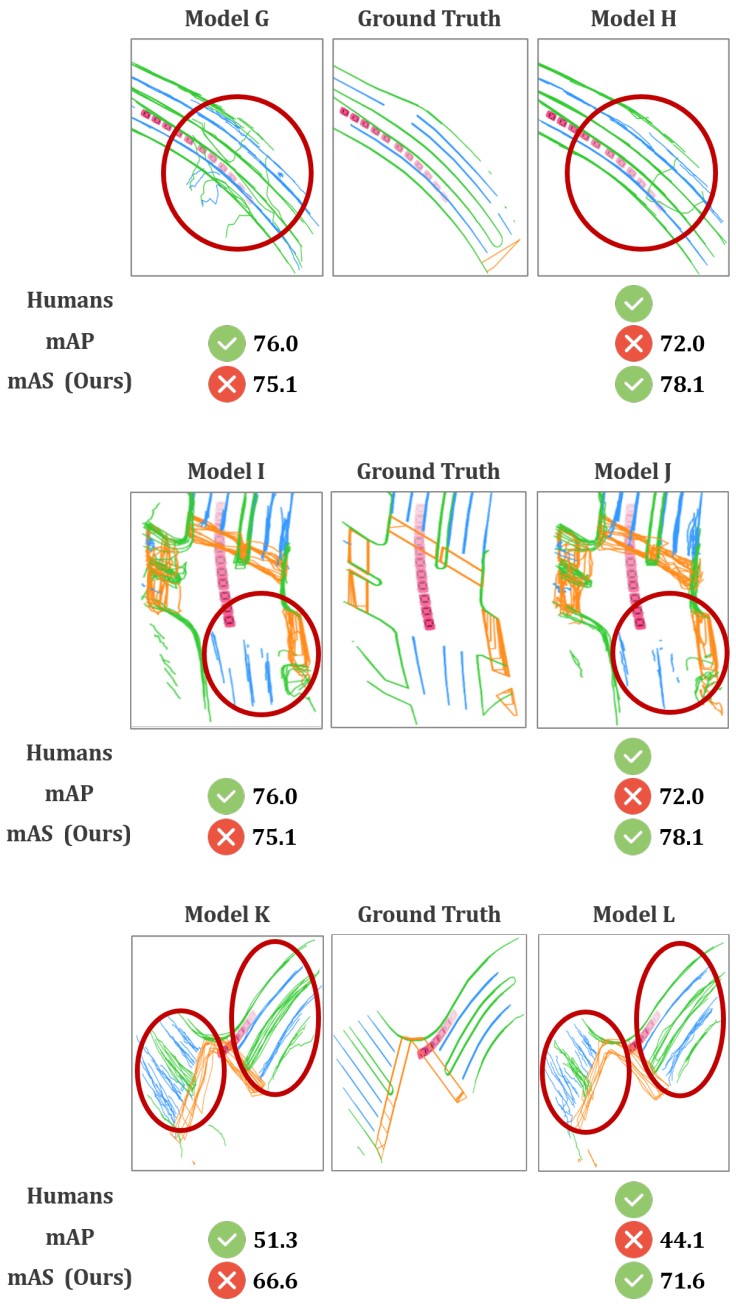

Figure 9: **Evaluating trustworthiness of online mapping models using human judgment, traditional mAP, and our mAS metric.**

## F  LIMITATIONS AND FUTURE WORK

This study presents the first dedicated benchmark for temporal stability evaluation in online HD mapping, yet several limitations indicate directions for future research. The current benchmark is constrained by the scope of existing datasets, particularly in representing complex real-world scenarios. Our evaluation primarily relies on standard driving sequences from the nuScenes dataset (Caesar et al., 2020), which lacks systematic coverage of challenging conditions such as extreme weather, adverse illumination, and intentional adversarial scenarios (Lou et al., 2025). Consequently, the current assessment may not fully reflect model stability under critical edge cases that are essential for safe autonomous driving.

Another limitation stems from the rapid evolution of this research field. While our benchmark encompasses 42 model variants representing major architectural paradigms, new methodologies continue to emerge at a rapid pace. The current static snapshot of model comparisons requires continuous updates to maintain relevance and comprehensiveness.

To address these limitations, we outline two primary directions for future work. First, we will establish a continuously maintained benchmark platform that systematically incorporates new research developments. This living benchmark will implement standardized evaluation protocols for emerging methodologies, ensuring fair comparisons and tracking progress over time. The platform will feature regular updates to model implementations, evaluation metrics, and dataset expansions, fostering community-wide collaboration (e.g., decentralized vehicular crowd sensing for online participant recruitment (Jiang et al., 2025)) and providing a reliable foundation for assessing advancements in temporal stability.

Second, we will expand the benchmark to include diverse challenging scenarios that better reflect real-world complexity. This expansion will incorporate data from multiple geographic regions with varying road infrastructures and traffic patterns. Specifically, we will integrate specialized datasets containing extreme weather conditions (heavy rain, snow, fog), low-light and night-time driving scenarios, and challenging urban environments with complex intersections and dense traffic. Furthermore, we will develop evaluation protocols for synthetic adversarial scenarios (e.g., occupancy-centric driving scene generation (Li et al., 2025a), controllable video generation (Ma et al., 2025) and diffusion transformer for image editing (Feng et al., 2025)) designed to stress-test model stability, such as sensor degradation simulations and challenging weather transitions(Zeng et al., 2025a; Xiao et al., 2025). These enhancements will provide a more comprehensive assessment of model robustness under critical conditions.

We believe these efforts will significantly advance the development of more reliable and robust online HD mapping systems.

