# OpenReview forum: "Stability Under Scrutiny: Benchmarking Representation Paradigms for Online HD Mapping"
_ICLR.cc/2026/Conference — ICLR 2026 Poster_

### Official Review · Reviewer_1LUQ · 2025-10-23

**Soundness:** 2
**Presentation:** 2
**Contribution:** 2
**Rating:** 4
**Confidence:** 3

**Summary:**

The paper introduces mAS, a comprehensive metric suite for evaluating the temporal stability of lane elements in online HD mapping. It measures Presence, Localization, and Shape Stability across frames and aggregates them into a unified score. Evaluation across various baseline methods is also proposed.

**Strengths:**

- The proposed mAS is a well-designed, multi-dimensional metric (Presence, Localization, Shape) that directly targets temporal stability in online HD mapping, addressing a clear gap in prior evaluations.

- The paper presents a thorough, large-scale comparison across diverse baselines, making temporal consistency differences easy to assess and helping the community select appropriate baselines.

- The manuscript is clear and well-structured; metric definitions and the evaluation protocol are easy to follow.

**Weaknesses:**

- `Reliance on accurate inter-frame alignment:`
The mAS metric depends on precise ego-motion/pose transformations to align frames. Errors from localization, calibration, or time sync can propagate into the stability score, introducing variance unrelated to the model’s intrinsic stability.

- `Vulnerable to “copy–paste” or over-smoothing strategies:`
A model could boost mAS by copying or heavily smoothing predictions across frames, inflating stability without being correct or responsive to scene changes.

- `Coupling with accuracy and interpretability of rankings:`
Because stability can rise even as accuracy drops, comparing models solely on mAS may be misleading. Low-mAP models could appear “stable” but be persistently wrong. This reduces the value of the submission.

- `Sensitivity to implementation choices:`
Polyline resampling density, curve parameterization, and discretization can affect shape and localization stability. Results may shift with frame rate and temporal window length.

**Questions:**

N/A

---

> ### Author Response · Authors · 2025-11-21
> **Robustness to Inter-frame Alignment Noise**
>
> We sincerely thank the reviewer for their constructive feedback. We are greatly encouraged by the positive comments recognizing that our work is "addressing a clear gap in prior evaluations," "presents a thorough, large-scale comparison," and is "clear and well-structured." In response to the specific points raised, we provide the following clarifications and will diligently incorporate corresponding revisions into the paper.
>
> > `Weaknesses-1`: `Reliance on accurate inter-frame alignment:` The mAS metric depends on precise ego-motion/pose transformations to align frames. Errors from localization, calibration, or time sync can propagate into the stability score, introducing variance unrelated to the model’s intrinsic stability.
>
> We thank the reviewer for this insightful comment. You are absolutely right that **ideally, inter-frame alignment should be perfectly precise**. In practice, minor alignment noise inevitably exists due to factors like slight localization drift and sensor synchronization biases. However, we wish to clarify that our evaluation framework is designed to **minimize the impact of such noise on our core conclusions** through the following measures:
>
> 1. **Alignment Using High-Quality Ground Truth Poses**: Our benchmark relies entirely on the **high-precision ground truth ego-pose** provided by the nuScenes dataset, rather than poses estimated by models. This **avoids contaminating the stability assessment with errors from online localization systems** at the source, providing a **fair, clean, and unified comparison benchmark** for all models.
> 2. **Focus on Relative Stability, Not Absolute Values**: The primary value of our benchmark lies in **comparing the relative stability performance of different models under identical conditions**. Since all models use the same alignment data, the effect of pose noise is **systematic and equitable** across them. Consequently, **the relative ranking of models remains fundamentally unchanged by these shared, minor alignment noises**. A model that demonstrates significantly higher stability despite slight noise exhibits a real and reliable advantage.
> 3. **Filtering Mechanism Mitigates Boundary Effects**: Our "perception range filtering" step automatically removes predicted points that fall outside the perception range after coordinate transformation. This mechanism effectively **filters out edge points disproportionately "pushed out" of view by minor alignment errors**, preventing them from having an outsized impact on the stability calculation.
>
> In summary, while we acknowledge the theoretical presence of alignment-introduced noise, the use of high-quality ground truth poses as a common benchmark, combined with our focus on **relative comparison** among models, means this noise **does not undermine our core findings regarding "significant stability differences between paradigms" and "mAP and mAS being independent dimensions."** Our benchmark successfully reveals the intrinsic differences in stability characteristics among models.

---

> ### Author Response · Authors · 2025-11-21
> **mAS-mAP Synergy Against Gaming and Misinterpretation**
>
> > `Weaknesses-2`: `Vulnerable to “copy–paste” or over-smoothing strategies:` A model could boost mAS by copying or heavily smoothing predictions across frames, inflating stability without being correct or responsive to scene changes.
> >
> > `Weaknesses-3`: `Coupling with accuracy and interpretability of rankings:` Because stability can rise even as accuracy drops, comparing models solely on mAS may be misleading. Low-mAP models could appear “stable” but be persistently wrong. This reduces the value of the submission.
>
> We thank the reviewer for raising this important point, which directly addresses a core philosophy of our work: **a superior model must strike a balance between "stability" and "precise responsiveness to scene changes."**
>
> We would like to clarify that the primary intention behind designing mAS was never to provide a standalone metric for evaluating map construction performance in isolation. Instead, it aims to reveal the "temporal stability" dimension that conventional mAP overlooks. Crucially, any model employing strategies like copy pasting or blind temporal smoothing would inevitably suffer a significant drop in precision metrics like mAP, even if it achieved a high localization stability score. This is because both copy pasting and heavily smoothing predictions across frames introduce substantial deviations from the ground truth for the affected instances.
>
> Therefore, when mAS is examined in conjunction with mAP, such pseudo stable models are easily exposed. They exhibit an anomalous profile of high mAS but very low mAP. Identifying and cautioning against this exact type of behavior is a key objective of our evaluation framework.
>
> Thus, the value of mAS lies in its synergistic use with inference accuracy metrics like mAP. Together, they define a more comprehensive trustworthiness evaluation quadrant, guiding the community to pursue models that achieve both high accuracy and high stability, rather than over optimizing for either metric alone.

---

> ### Author Response · Authors · 2025-11-21
> **Sensitivity Analysis of Implementation Hyperparameters - Additional Experiments**
>
> MapTRv1-noTemp-R18-GKT-24
>
> | N    | presence | loc    | shape  | mAS    |
> | ---- | -------- | ------ | ------ | ------ |
> | 30   | 0.88     | 0.7535 | 0.8862 | 0.7318 |
> | 50   | 0.8776   | 0.7499 | 0.8852 | 0.7283 |
> | 100  | 0.8794   | 0.7541 | 0.8867 | 0.7317 |
> | 150  | 0.8783   | 0.7533 | 0.8862 | 0.73   |
> | 200  | 0.8789   | 0.7524 | 0.887  | 0.7305 |
>
> MapTRv1-noTemp-R50-GKT-24
>
> | N    | presence | loc    | shape  | mAS    |
> | ---- | -------- | ------ | ------ | ------ |
> | 30   | 0.9107   | 0.6534 | 0.9078 | 0.7145 |
> | 50   | 0.912    | 0.6544 | 0.9063 | 0.7158 |
> | 100  | 0.9111   | 0.6479 | 0.9052 | 0.7112 |
> | 150  | 0.9107   | 0.6473 | 0.9087 | 0.7119 |
> | 200  | 0.911    | 0.6488 | 0.905  | 0.7113 |
>
> MapTRv1-Temp-R50-GKT-24
>
> | N    | presence | loc    | shape  | mAS    |
> | ---- | -------- | ------ | ------ | ------ |
> | 30   | 0.8964   | 0.5663 | 0.8973 | 0.6624 |
> | 50   | 0.8861   | 0.5974 | 0.8928 | 0.6657 |
> | 100  | 0.8859   | 0.5769 | 0.8957 | 0.6582 |
> | 150  | 0.8965   | 0.5702 | 0.8971 | 0.6625 |
> | 200  | 0.8967   | 0.569  | 0.8959 | 0.6614 |
>
> MapTRv1-Temp-R50-BevFormer-24
>
> | N    | presence | loc    | shape  | mAS    |
> | ---- | -------- | ------ | ------ | ------ |
> | 30   | 0.8839   | 0.7169 | 0.9139 | 0.7284 |
> | 50   | 0.9035   | 0.6947 | 0.9121 | 0.73   |
> | 100  | 0.9012   | 0.7326 | 0.9142 | 0.7481 |
> | 150  | 0.902    | 0.7359 | 0.9146 | 0.7502 |
> | 200  | 0.903    | 0.7344 | 0.9146 | 0.7507 |
>
> MapTRv2-noTemp-R50-BevPool-24
>
> | N    | presence | loc    | shape  | mAS    |
> | ---- | -------- | ------ | ------ | ------ |
> | 30   | 0.9156   | 0.6714 | 0.914  | 0.7342 |
> | 50   | 0.9147   | 0.6861 | 0.9095 | 0.7396 |
> | 100  | 0.9149   | 0.6853 | 0.9149 | 0.7414 |
> | 150  | 0.9147   | 0.6868 | 0.9152 | 0.7426 |
> | 200  | 0.9145   | 0.6848 | 0.9147 | 0.7411 |
>
> MGMap-noTemp-R50-Former-24
>
> | N    | presence | loc    | shape  | mAS    |
> | ---- | -------- | ------ | ------ | ------ |
> | 30   | 0.9188   | 0.7437 | 0.9249 | 0.7756 |
> | 50   | 0.9221   | 0.7498 | 0.9234 | 0.7801 |
> | 100  | 0.9217   | 0.7492 | 0.925  | 0.7802 |
> | 150  | 0.9212   | 0.7472 | 0.925  | 0.7789 |
> | 200  | 0.922    | 0.7476 | 0.9253 | 0.7801 |
>
> MapQR-noTemp-R50-Former-24
>
> | N    | presence | loc    | shape  | mAS    |
> | ---- | -------- | ------ | ------ | ------ |
> | 30   | 0.9184   | 0.7559 | 0.9247 | 0.7839 |
> | 50   | 0.9182   | 0.7556 | 0.9156 | 0.7781 |
> | 100  | 0.918    | 0.7539 | 0.9263 | 0.7826 |
> | 150  | 0.9173   | 0.7532 | 0.9261 | 0.7818 |
> | 200  | 0.918    | 0.7541 | 0.9259 | 0.7825 |
>
> GEMap-noTemp-R50-Former-24
>
> | N    | presence | loc    | shape  | mAS    |
> | ---- | -------- | ------ | ------ | ------ |
> | 30   | 0.9144   | 0.6985 | 0.9277 | 0.752  |
> | 50   | 0.9227   | 0.6973 | 0.9258 | 0.7546 |
> | 100  | 0.922    | 0.693  | 0.931  | 0.7499 |
> | 150  | 0.9233   | 0.6916 | 0.9315 | 0.7504 |
> | 200  | 0.9232   | 0.6937 | 0.9309 | 0.7504 |
>
>
>
> Regarding the potential influence of "curve parameterization" and "discretization" on stability assessment, we did not fully grasp the specific question or suggestion raised. Could you please rephrase or elaborate on this point? We would be happy to provide a detailed clarification or supplementary analysis based on a more precise understanding of your concern.
>
> Concerning the impact of frame rate on stability evaluation, we acknowledge its potential effect. However, in practice, this influence can be effectively mitigated by dynamically adjusting the sampling stride $M$. The core factor affecting stability assessment is the time interval $Interval$ between consecutive sampled frames, which is jointly determined by the frame rate $FPS$ and the sampling stride $M$,  specifically $Interval=\frac{M}{FPS}$. While the frame rate is typically fixed by the sensor setup, the sampling stride $M$ can be conveniently adjusted in the evaluation code. Therefore, we recommend that users first select a suitable target time interval $Interval$ for their specific stability analysis and then determine the corresponding sampling stride $M$ based on the dataset's $FPS$ to achieve the desired temporal sampling resolution. This approach ensures consistent and comparable stability evaluations across datasets with different frame rates.

---

> ### Author Response · Authors · 2025-11-21
> **Sensitivity Analysis of Implementation Hyperparameters**
>
> > `Weaknesses-4`: `Sensitivity to implementation choices:` Polyline resampling density, curve parameterization, and discretization can affect shape and localization stability. Results may shift with frame rate and temporal window length.
>
> We thank the reviewer for this comment. We agree that hyperparameter choices can influence absolute metric values. Regarding the specific hyperparameters you mentioned, we had already included partial ablation studies in Appendix C.2 of our initial submission. We will provide further justification and additional experimental results in this rebuttal to address the remaining points comprehensively.
>
> Regarding the event window length, we conducted a dedicated **ablation study on the sampling stride (M)** in Appendix C.2. The results show that while the absolute mAS scores for all models decrease as $M$ increases, which aligns with intuition, the overall distribution and relative trends remain consistent. This demonstrates that the impact of the sampling stride is consistent and fair across different models.
>
> Concerning the resampling density ($N$), the design principle is to ensure it is **significantly higher** than the number of points in the vast majority of original map instances. In current mainstream paradigms, this instance point count is typically fixed at 20. The core objective is to **avoid computational instability caused by an insufficient number of random samples**, thereby ensuring the **consistency and repeatability** of the stability metric. If N is too small, random variations in individual sample points can disproportionately influence the overall stability score. At the same time, we aim to prevent **computational inefficiency** from an excessively large $N$. Our experiments confirm that the map stability metric shows no significant variation once the resampling density reaches $N \geq 100$. Therefore, balancing sampling density against computational cost, we selected $N = 100$ to achieve the best overall performance.

---

### Official Review · Reviewer_dHha · 2025-10-29

**Soundness:** 3
**Presentation:** 3
**Contribution:** 3
**Rating:** 6
**Confidence:** 5

**Summary:**

This paper addresses temporal stability for HD Mapping, which is a critical and long-overlooked problem. The authors argue that current mainstream evaluation metrics, such as mAP, focus exclusively on single-frame geometric accuracy. To address this gap, the paper introduces the benchmark specifically designed to evaluate the temporal stability of online HD maps. The core contribution is a new, multi-dimensional metric named mAS (mean Average Stability).

**Strengths:**

1. The task of this paper is meaningful, which systematically address and quantify the critical evaluation blind spot of "temporal stability" in the online HD mapping domain.
2. The proposed mAS metric is well-designed and comprehensive.
3. The paper is written clearly and with a strong logical flow.

**Weaknesses:**

1. The effectiveness of mAS relies on the GT stability. If the GT annotations themselves are inconsistent between frames (e.g., "jitter" or "flicker" from human labelers, which is common in large datasets), the mAS metric would unfairly penalize a model that produces a stable (and potentially more correct) output.
2. Before stability calculation (Sec 3.3), the framework must perform "geometric alignment" using the vehicle's pose data. If the vehicle's localization data is noisy, it will introduce artificial "instability" during the alignment process, again leading to an incorrect penalty on the perception model's stability.
3. The paper mentions (Sec 3.3) using uniform resampling along the x-axis to compare polylines. This method may becomes highly unstable or fails for vertical line segments (i.e., those running parallel to the vehicle's direction of travel). The authors do not clarify how this common and critical case (e.g., lane lines) is handled, which could lead to biases in the evaluation.
4. The current mAS focuses on geometry and presence. However, another significant failure mode is "semantic flickering" where an element's position and shape are stable, but its classification label jumps between categories (e.g., "lane" and "road boundary"). This is equally detrimental to downstream tasks, but the mAS framework does not appear to capture this.

**Questions:**

1. Could the authors elaborate on the sensitivity of the mAS metric to GT quality and ego-motion noise? For example, if varying levels of noise are injected into the GT or ego-motion data, how much do the model rankings change? This seems key to building trust in this new benchmark.
2. How are near-vertical polylines (which would run parallel to the x-axis in the BEV-like coordinate system mentioned) handled during the resampling process? Does this strategy risk ignoring the stability evaluation for these critical elements, such as lane lines?
3. Does the current mAS framework (particularly the instance matching stage) account for the stability of classification labels? If an instance is stable in geometry and location but its class label flickers between frames, is this captured by the metric?
4. An interesting finding is that adding temporal fusion (Temp=4) to a non-temporal model like MapTR-GKT significantly degrades stability (mAS from 71.6 to 66.6). The authors speculate this is due to "noise". Could the authors provide a deeper analysis or hypothesis at the feature level? For instance, is it possible that the temporal module struggles to align the spatial features produced by GKT, leading to feature-level aliasing or confusion?

I would be happy to raise the score if the authors provide more analysis.

---

> ### Author Response · Authors · 2025-11-21
> **Robustness of mAS to Ground Truth**
>
> We sincerely thank the reviewer for their constructive feedback. We are greatly encouraged by the positive assessment describing our work as "meaningful" and acknowledging that it "systematically addresses and quantifies the critical evaluation blind spot of 'temporal stability' in the online HD mapping domain," with a "well-designed and comprehensive metric" that is "written clearly and with a strong logical flow." In response to the specific points raised, we provide the following clarifications and will diligently incorporate corresponding revisions into the paper.
>
> > `Weaknesses-1`: The effectiveness of mAS relies on the GT stability. If the GT annotations themselves are inconsistent between frames (e.g., "jitter" or "flicker" from human labelers, which is common in large datasets), the mAS metric would unfairly penalize a model that produces a stable (and potentially more correct) output.
> >
> > `Questions-1`: Could the authors elaborate on the sensitivity of the mAS metric to GT quality and ego-motion noise? For example, if varying levels of noise are injected into the GT or ego-motion data, how much do the model rankings change? This seems key to building trust in this new benchmark.
>
> We thank the reviewer for their insightful questions regarding the use of Ground Truth (GT) as an intermediary for cross-frame matching. In practice, the GT serves solely as an intermediate medium for matching instance predictions across different frames. Minor annotation jitter in the GT does not lead to unfair penalties in stability assessment during actual evaluation; on the contrary, this approach is fairer compared to learning-based matching methods. Below, I will supplement the explanation of the instance matching mechanism to substantiate this point:
>
> **The key aspect of our method is:** GT acts as a "matching medium" here, rather than an "absolute geometric benchmark."
>
> The matching process consists of two steps:
>
> ​	**First,** within each frame, we independently match model predictions to the current frame's GT (using the Hungarian algorithm).
>
> ​	**Second,** predictions matched to *the same GT instance* across different frames are linked via the **persistent IDs of GT instances over time**.
>
> The essence of this process lies in the following principle: **Whenever a predicted instance is correctly matched to the same ground-truth (GT) instance at both frame t and frame t+k, these two predicted instances are considered a "successful match," regardless of minor inter-frame jitter in the geometric annotations of the GT instance itself.** Consequently, the evaluation focus shifts from the "geometric consistency of the GT" to the **"consistency of the model's association with a stable identifier (GT ID)."** In well-established datasets such as nuScenes, minor jitter in GT instances is generally insufficient to disrupt the correct matching between predicted and GT instances, and therefore does not affect the stability evaluation process.
>
> On the other hand, our goal is to establish a fair benchmark for comparing the stability across different representation paradigms. Learning-based matching approaches (e.g., SORT/MOT) inherently introduce additional model complexity and training uncertainty, which could confound the stability evaluation with the matcher’s performance. In contrast, our GT matching strategy provides a **unified, stable, and model-agnostic benchmark**, ensuring that all models are evaluated under the same standard.
>
> In summary, we contend that the current GT ID-based indirect matching strategy represents the most reliable and straightforward approach at this stage to achieve our objective of a fair and reproducible comparison of stability across different models.

---

> ### Author Response · Authors · 2025-11-21
> **Robustness of mAS to the Vehicle's Localization Data**
>
> > `Weaknesses-2`: Before stability calculation (Sec 3.3), the framework must perform "geometric alignment" using the vehicle's pose data. If the vehicle's localization data is noisy, it will introduce artificial "instability" during the alignment process, again leading to an incorrect penalty on the perception model's stability.
>
> We thank the reviewer for this insightful observation. You are absolutely correct that **ideally, inter-frame alignment should be perfectly precise**. In practice, minor misalignments do exist due to factors such as slight localization drift and sensor synchronization discrepancies. However, we wish to clarify that our evaluation framework is specifically designed to **minimize the impact of such noise on the core conclusions** through the following measures:
>
> 1. **Alignment using high-quality ground-truth poses**: Our benchmark relies entirely on the **high-precision ground-truth ego-pose** provided by the nuScenes dataset, rather than poses estimated by models. This approach fundamentally prevents contamination of stability evaluation from online localization system errors, establishing a **fair, clean, and unified benchmarking standard** for all models.
> 2. **Evaluation of Relative Stability, Not Absolute Metric Values**: The core value of our benchmark lies in **comparing the relative stability performance of different models under identical conditions**. Since all models are aligned using the same set of pose data, the impact of pose noise is **systematic and equitable** across all evaluations. Consequently, **the relative rankings of the models, which model is more stable, are not fundamentally altered** by these shared, minor alignment noises. A model that demonstrates significantly higher stability despite slight noise exhibits a robust and reliable advantage.
>
> In summary, while we acknowledge the theoretical presence of alignment-induced noise, the use of high-quality ground-truth poses as a common benchmark and our focus on relative comparison among models ensure that this noise does not undermine our core conclusions regarding the significant stability differences across paradigms. Our benchmark successfully reveals intrinsic differences in model stability characteristics.

---

> ### Author Response · Authors · 2025-11-21
> **Resampling Mechanism - Part 1**
>
> > `Weaknesses-3`: The paper mentions (Sec 3.3) using uniform resampling along the x-axis to compare polylines. This method may becomes highly unstable or fails for vertical line segments (i.e., those running parallel to the vehicle's direction of travel). The authors do not clarify how this common and critical case (e.g., lane lines) is handled, which could lead to biases in the evaluation.
> >
> > `Questions-2`: How are near-vertical polylines (which would run parallel to the x-axis in the BEV-like coordinate system mentioned) handled during the resampling process? Does this strategy risk ignoring the stability evaluation for these critical elements, such as lane lines?
>
> We sincerely thank the reviewer for their insightful observation regarding the limitation of fixed x-axis resampling when applied to curved or near-horizontal trajectory segments.
>
> We would like to clarify and apologize for this discrepancy: the x-axis resampling described in the methodology section reflects the initial implementation in our first version of code. However, during subsequent experimental phases, we indeed recognized this limitation and accordingly updated the implementation. We regret that this modification was inadvertently not reflected in the manuscript. For clarity, all experimental results reported are based on a **dynamic axis selection mechanism**, which aligns with the reviewer's suggestion. Its core principle is as follows:
>
> For a map instance composed of a point sequence ${(x_1,y_1),(x_2,y_2), \ldots,(x_M,y_M)}$, we do not perform resampling along a fixed axis. Instead, we dynamically determine the primary sampling axis by analyzing its local geometric orientation. The specific procedure is as follows:
>
> 1. **Calculation of Local Direction Vectors**: For each consecutive segment $\vec{s_j}$ in the instance, formed by points $p_j$ and $p_{j+1}$, its direction vector is computed as $\vec{v_j}=(dx_j,dy_j)=(x_{j+1}-x_j,y_{j+1}-y_j)$.
> 2. **Determination of the Primary Sampling Axis**: We compute the absolute changes of the segment along each coordinate axis:
>
> $$
> \Delta x_j=|dx_j|,\ \Delta y_j=|dy_j|​
> $$
>
>
>
> ​		If $\Delta y_j > \Delta x_j$, the segment is considered **vertically oriented** and categorized as an **"X-sampling interval"**. Conversely, if $\Delta x_j > \Delta y_j$ , it is considered **horizontally oriented** and categorized as a **"Y-sampling interval"**.
>
> 3. **Determination of the Number of Sampling Points per Interval**: The number of sampling points in each interval is determined based on its length.
>
>    First, the instance is divided into K intervals using the dynamic axis selection method described previously. For each interval *i*, we calculate its length along the **primary sampling axis**:
>
> - **For an X-sampling interval:** $L_j = x^j_{max}-x^j_{min}$
> - **For an Y-sampling interval:** $L_j = y^j_{max}-y^j_{min}$
>
> Subsequently, sampling points are allocated proportionally based on the relative length of each interval. The initial number of sampling points $n_j$ for each interval is calculated as follows:
>
> $$n_j=round(N \times \frac{L_j}{\sum^{K}_{t=1}L_t})n_j$$
>
> where $round(\cdot)$denotes rounding to the nearest integer.
>
> 1. **Total Adjustment Mechanism**
>
>    Since rounding may cause the total number of points$\sum^K_{i=1}n_i \ne N$, we introduce an adjustment mechanism:
>
>    - if $\sum^K_{i=1}n_i \lt N$: Points are incrementally allocated to intervals in descending order of their length ratio$\frac{L_i}{\sum L_j}$until the total number of points equals N。
>    - if $\sum^K_{i=1}n_i \gt N$: Points are sequentially removed from intervals in ascending order of their length ratio until the total number of points equals N.
>
> 2. **Axial Resampling Execution**:
>
>    - Within an **X-sampling interval**, we uniformly generate $n_s$ sample points ${x_i}$ across the x-value range $[x^s_{min},x^s_{max}]$ , and compute the corresponding y-coordinates via linear interpolation:
>      $$
>       x_i=x^s_{min}+(i-1) \cdot \frac{x^s_{max}-x^s_{min}}{n_s-1} ,i=1,2, \ldots ,n_s
>      $$
>
>      $$
>       poly^{sample}={(x_i,y(x_i))}
>      $$
>
>    - Within an **Y-sampling interval**, we uniformly generate $n_s$ sample points ${y_i}$ across the x-value range $[y^s_{min},y^s_{max}]$ , and compute the corresponding x-coordinates via linear interpolation:
>      $$
>       y_i=y^s_{min}+(i-1) \cdot \frac{y^s_{max}-y^s_{min}}{n_s-1} ,i=1,2, \ldots ,n_s
>      $$
>
>      $$
>      poly^{sample}={(x(y_i),y_i)}
>      $$

---

> ### Author Response · Authors · 2025-11-21
> **Resampling Mechanism - Part 2**
>
> Dynamic axis selection mechanism ensures that, regardless of the map element’s orientation, uniform sampling is always performed along its locally dominant direction of extension, thereby establishing a **fair and robust benchmark** for point-to-point geometric comparison. As such, it serves as an efficient and practical approximation to simple arc-length parameterization, specifically designed for the **piecewise-smooth polyline structures** prevalent in online HD maps.
>
> We sincerely apologize for the omission of this key implementation detail in the manuscript, which understandably led to the reviewer’s misunderstanding. We will comprehensively supplement the methodology section (Sec. 3.3) in the final version with the above principles and formulas, and we are grateful to the reviewer for prompting us to refine the description for greater precision.

---

> ### Author Response · Authors · 2025-11-21
> **Semantic Stability and the Scope of mAS**
>
> > `Weaknesses-4`: The current mAS focuses on geometry and presence. However, another significant failure mode is "semantic flickering" where an element's position and shape are stable, but its classification label jumps between categories (e.g., "lane" and "road boundary"). This is equally detrimental to downstream tasks, but the mAS framework does not appear to capture this.
> >
> > `Questions-3`: Does the current mAS framework (particularly the instance matching stage) account for the stability of classification labels? If an instance is stable in geometry and location but its class label flickers between frames, is this captured by the metric?
>
> We sincerely appreciate the reviewer's insightful question, which provides us with an opportunity to clarify the design rationale of our framework more explicitly. We fully agree that **"semantic flickering"** is critical for downstream tasks. We wish to emphasize that **one of the core design objectives of our proposed presence metric is precisely to directly capture various types of prediction inconsistencies, including semantic flickering.**
>
> First, the "semantic flickering" mentioned would lead to a **significant decrease in the presence metric**, thereby reducing the mAS. When a model exhibits semantic flickering. For example, if a map instance is recognized as a "lane divider" at frame *t* but as a "road boundary" at frame *t+1*. Our presence metric is heavily penalized once such a mismatch occurs.
>
> Furthermore, the presence metric also accurately captures another form of instability: even after accounting for range effects, if a map instance is detected at frame *t* but missed at frame *t+1*, the presence metric similarly incurs a strong penalty.
>
> We thank the reviewer once again. This valuable discussion has allowed us to more clearly articulate the broad applicability of the **"presence stability" metric**, which indeed serves as a core mechanism in our framework for addressing issues such as semantic flickering.

---

> ### Author Response · Authors · 2025-11-21
> **Analysis about Our Interesting Finding**
>
> > `Questions-4`: An interesting finding is that adding temporal fusion (Temp=4) to a non-temporal model like MapTR-GKT significantly degrades stability (mAS from 71.6 to 66.6). The authors speculate this is due to "noise". Could the authors provide a deeper analysis or hypothesis at the feature level? For instance, is it possible that the temporal module struggles to align the spatial features produced by GKT, leading to feature-level aliasing or confusion?
>
> Thank you very much for raising this profound question, which has prompted us to analyze this critical phenomenon at a more fundamental level. We fully agree that attributing the stability drop in MapTR-GKT after incorporating temporal fusion (mAS decreased from 71.6 to 66.6) merely to "noise" is insufficient. Based on our experimental data and model architecture analysis, **the core reason lies in the fundamental incompatibility between the static spatial features generated by the GKT encoder and the dynamic representations required for temporal fusion.**
>
> Specifically, the decline in stability stems from the following three interrelated underlying mechanisms:
>
> 1. **The feature space solidifies the static scene representation, making dynamic alignment challenging.** The core strength of GKT lies in its efficient construction of single-frame BEV features through geometric priors. However, this makes its features highly dependent on the instantaneous coordinate system at the time of construction. When historical frame features are introduced, due to the cumulative error of ego-motion, these statically generated features from different coordinate systems cannot be precisely aligned in the BEV space. **This inherent misalignment at the feature level directly leads to representational confusion after fusion**, thereby causing detection jitter (drop in Presence stability) and positional jumps (drop in Localization stability) of map elements.
>
> 2. **Temporal aggregation of localized fine-grained features amplifies high-frequency jitter.** GKT excels at extracting local, high-resolution geometric details. However, when a simple temporal fusion module (e.g., basic convolution or attention mechanisms) attempts to aggregate these highly localized features, it **not only fails to smooth out their inherent inter-frame variations but even amplifies such high-frequency jitter**. This manifests as instability in the position and shape of map elements after decoding.
>
> 3. **The training objective optimized for single-frame performance conflicts with that of the temporal module.** As a non-temporal model, MapTR's GKT encoder and decoder are strictly optimized to achieve optimal accuracy on individual frames. When a fusion module without temporal consistency constraints is introduced, the entire network falls into an optimization dilemma: the single-frame supervision signal forces the features to maintain overfitting to instantaneous observations, while the newly added temporal parameters, lacking explicit guidance, **disrupt the originally well-optimized feature distribution for single frames**. This ultimately leads to performance degradation in the temporal dimension.
>
> The validity of this analysis is **confirmed unequivocally** by our controlled experiments on MapTR-BEVFormer. As shown in Table 4 of the paper, BEVFormer inherently employs deformable attention for temporal modeling, meaning its **features are designed from the outset for cross-frame alignment and fusion**. Consequently, when a temporal fusion module is incorporated, its features can be integrated smoothly and consistently, leading to simultaneous improvements in both accuracy and stability (mAP increased from 41.6 to 53.3, while mAS rose from 71.3 to 73.0).

---

### Official Review · Reviewer_DMdK · 2025-10-30

**Soundness:** 2
**Presentation:** 3
**Contribution:** 3
**Rating:** 4
**Confidence:** 4

**Summary:**

This paper presents the first benchmark for temporal stability in online HD mapping, introducing the mAS metric with Presence, Localization, and Shape components, and shows through 42 model variants that accuracy (mAP) and stability (mAS) are largely independent dimensions.

**Strengths:**

1. The problem is well-motivated - existing metrics focus on single-frame accuracy while ignoring temporal consistency that's critical for safe autonomous driving.

2. Testing 42 model variants across different backbones, BEV encoders, and temporal fusion strategies provides solid empirical evidence for the mAP-mAS independence claim.

3. Breaking stability into Presence, Localization, and Shape gives more actionable insights than a single number would.

**Weaknesses:**

1. Using ground truth as a bridge for cross-frame matching (Algorithm 2) assumes perfect GT consistency, but annotation noise in consecutive frames could bias your measurements - have you considered direct matching with learned features like in SORT [1] or MOT approaches [2]?

2. Your uniform resampling along x-axis only works for roughly vertical polylines; this breaks for curved roads or horizontal boundaries, whereas arc-length parameterization [3] would handle arbitrary shapes more robustly.

3. Computing curvature as average angles between segments (Algorithm 7) is quite sensitive to sampling density - why not use more robust shape metrics like Fréchet distance [4] or proper curvature signatures [5]?

4. The choices of β=15.0 and ω=0.7 seem arbitrary without ablation studies - how sensitive is mAS to these values, and do they generalize to different map ranges or vehicle types?

5. Only using nuScenes limits your conclusions, especially since it has relatively benign weather - what happens on corruption benchmarks like nuScenes-C [6] or RoboDrive [7] where models might fail differently?

6. While your qualitative examples are compelling, there's no quantitative link between mAS and actual planning metrics like collision rate or trajectory smoothness - does higher mAS actually lead to safer planning?

7. Testing M∈{2,3,5,10} is a start, but you're randomly sampling rather than systematically studying how stability degrades with time or identifying worst-case temporal transitions like occlusion events.

8. You don't compare against simpler alternatives like frame-to-frame Chamfer distance [8] or temporal consistency losses - how do we know mAS is actually better than these baselines?

## References

[1] Bewley et al. "Simple online and realtime tracking." ICIP 2016.

[2] Voigtlaender et al. "MOTS: Multi-object tracking and segmentation." CVPR 2019.

[3] Pottmann et al. "Geometry of the squared distance function to curves and surfaces." Visualization and Mathematics III, 2003.

[4] Alt & Godau. "Computing the Fréchet distance between two polygonal curves." IJCGA, 1995.

[5] Mokhtarian & Mackworth. "Curvature-based shape representation for planar curves." IEEE TPAMI, 1992.

[6] Xie et al. "RoboBEV: Towards robust bird's eye view perception under corruptions." arXiv:2304.06719, 2023.

[7] Kong et al. "Robo3D: Towards robust and reliable 3D perception." ICCV 2023.

[8] Achlioptas et al. "Learning representations and generative models for 3d point clouds." ICML 2018.

**Questions:**

1. Can you provide ablation studies on β, ω, and N to show that mAS rankings are stable across reasonable parameter choices?

2. What happens to your GT-based matching when annotations themselves have temporal jitter - have you measured annotation consistency in nuScenes?

3. Could you add experiments showing mAS correlation with downstream planning metrics to validate that it actually predicts safe behavior?

4. How would your results change on challenging datasets with weather corruptions or sensor degradation?

---

> ### Author Response · Authors · 2025-11-21
> **Robustness of GT-based Cross-frame Matching**
>
> We sincerely thank the reviewer for their constructive feedback. We are particularly encouraged by the positive assessment describing our work as "well-motivated," presenting "solid empirical evidence," and providing "actionable insights." In response to the points raised, we provide the following clarifications and will diligently incorporate corresponding revisions into the paper.
>
> > `Weaknesses-1`：Using ground truth as a bridge for cross-frame matching (Algorithm 2) assumes perfect GT consistency, but annotation noise in consecutive frames could bias your measurements - have you considered direct matching with learned features like in SORT [1] or MOT approaches [2]?
> >
> > `Question-2`：What happens to your GT-based matching when annotations themselves have temporal jitter - have you measured annotation consistency in nuScenes?
>
> We thank the reviewer for this insightful comment. You are absolutely right that **ideally, inter-frame alignment should be perfectly precise**. In practice, minor alignment noise inevitably exists due to factors like slight localization drift and sensor synchronization biases. However, we wish to clarify that our evaluation framework is designed to **minimize the impact of such noise on our core conclusions** through the following measures:
>
> 1. **Alignment Using High-Quality Ground Truth Poses**: Our benchmark relies entirely on the **high-precision ground truth ego-pose** provided by the nuScenes dataset, rather than poses estimated by models. This **avoids contaminating the stability assessment with errors from online localization systems** at the source, providing a **fair, clean, and unified comparison benchmark** for all models.
> 2. **Focus on Relative Stability, Not Absolute Values**: The primary value of our benchmark lies in **comparing the relative stability performance of different models under identical conditions**. Since all models use the same alignment data, the effect of pose noise is **systematic and equitable** across them. Consequently, **the relative ranking of models remains fundamentally unchanged by these shared, minor alignment noises**. A model that demonstrates significantly higher stability despite slight noise exhibits a real and reliable advantage.
> 3. **Filtering Mechanism Mitigates Boundary Effects**: Our "perception range filtering" step automatically removes predicted points that fall outside the perception range after coordinate transformation. This mechanism effectively **filters out edge points disproportionately "pushed out" of view by minor alignment errors**, preventing them from having an outsized impact on the stability calculation.
>
> In summary, while we acknowledge the theoretical presence of alignment-introduced noise, the use of high-quality ground truth poses as a common benchmark, combined with our focus on **relative comparison** among models, means this noise **does not undermine our core findings regarding "significant stability differences between paradigms" and "mAP and mAS being independent dimensions."** Our benchmark successfully reveals the intrinsic differences in stability characteristics among models.

---

> ### Author Response · Authors · 2025-11-21
> **Resampling Mechanism**
>
> > `Weaknesses-2`：Your uniform resampling along x-axis only works for roughly vertical polylines; this breaks for curved roads or horizontal boundaries, whereas arc-length parameterization [3] would handle arbitrary shapes more robustly.
>
> We sincerely thank the reviewer for their insightful observation regarding the limitation of fixed x-axis resampling when applied to curved or near-horizontal trajectory segments.
>
> We would like to clarify and apologize for this discrepancy: the x-axis resampling described in the methodology section reflects the initial implementation in our first version of code. However, during subsequent experimental phases, we indeed recognized this limitation and accordingly updated the implementation. We regret that this modification was inadvertently not reflected in the manuscript. For clarity, all experimental results reported are based on a **dynamic axis selection mechanism**, which aligns with the reviewer's suggestion. Its core principle is as follows:
>
> For a map instance composed of a point sequence ${(x_1,y_1),(x_2,y_2), \ldots,(x_M,y_M)}$, we do not perform resampling along a fixed axis. Instead, we dynamically determine the primary sampling axis by analyzing its local geometric orientation. The specific procedure is as follows:
>
> 1. **Calculation of Local Direction Vectors**: For each consecutive segment $\vec{s_j}$ in the instance, formed by points $p_j$ and $p_{j+1}$, its direction vector is computed as $\vec{v_j}=(dx_j,dy_j)=(x_{j+1}-x_j,y_{j+1}-y_j)$.
> 2. **Determination of the Primary Sampling Axis**: We compute the absolute changes of the segment along each coordinate axis:
>
> $$
> \Delta x_j=|dx_j|,\ \Delta y_j=|dy_j|​
> $$
>
>
>
> ​		If $\Delta y_j > \Delta x_j$, the segment is considered **vertically oriented** and categorized as an **"X-sampling interval"**. Conversely, if $\Delta x_j > \Delta y_j$ , it is considered **horizontally oriented** and categorized as a **"Y-sampling interval"**.
>
> 3. **Determination of the Number of Sampling Points per Interval**: The number of sampling points in each interval is determined based on its length.
>
>    First, the instance is divided into K intervals using the dynamic axis selection method described previously. For each interval *i*, we calculate its length along the **primary sampling axis**:
>
> - **For an X-sampling interval:** $L_j = x^j_{max}-x^j_{min}$
> - **For an Y-sampling interval:** $L_j = y^j_{max}-y^j_{min}$
>
> Subsequently, sampling points are allocated proportionally based on the relative length of each interval. The initial number of sampling points $n_j$ for each interval is calculated as follows:
> $$
> n_j=round(N \times \frac{L_j}{\sum^{K}_{t=1}L_t})n_j
> $$
>
> where $round(\cdot)$denotes rounding to the nearest integer.
>
> 1. **Total Adjustment Mechanism**
>
>    Since rounding may cause the total number of points$\sum^K_{i=1}n_i \ne N$, we introduce an adjustment mechanism:
>
>    - if $\sum^K_{i=1}n_i \lt N$: Points are incrementally allocated to intervals in descending order of their length ratio$\frac{L_i}{\sum L_j}$until the total number of points equals N。
>    - if $\sum^K_{i=1}n_i \gt N$: Points are sequentially removed from intervals in ascending order of their length ratio until the total number of points equals N.
>
> 2. **Axial Resampling Execution**:
>
>    - Within an **X-sampling interval**, we uniformly generate $n_s$ sample points ${x_i}$ across the x-value range $[x^s_{min},x^s_{max}]$ , and compute the corresponding y-coordinates via linear interpolation:
>      $$
>       x_i=x^s_{min}+(i-1) \cdot \frac{x^s_{max}-x^s_{min}}{n_s-1} ,i=1,2, \ldots ,n_s
>      $$
>
>      $$
>       poly^{sample}={(x_i,y(x_i))}
>      $$
>
>    - Within an **Y-sampling interval**, we uniformly generate $n_s$ sample points ${y_i}$ across the x-value range $[y^s_{min},y^s_{max}]$ , and compute the corresponding x-coordinates via linear interpolation:
>      $$
>       y_i=y^s_{min}+(i-1) \cdot \frac{y^s_{max}-y^s_{min}}{n_s-1} ,i=1,2, \ldots ,n_s
>      $$
>
>      $$
>      poly^{sample}={(x(y_i),y_i)}
>      $$
>
> This mechanism ensures that, regardless of the map element’s orientation, uniform sampling is always performed along its locally dominant direction of extension, thereby establishing a **fair and robust benchmark** for point-to-point geometric comparison. As such, it serves as an efficient and practical approximation to simple arc-length parameterization, specifically designed for the **piecewise-smooth polyline structures** prevalent in online HD maps.
>
> We sincerely apologize for the omission of this key implementation detail in the manuscript, which understandably led to the reviewer’s misunderstanding. We will comprehensively supplement the methodology section (Sec. 3.3) in the final version with the above principles and formulas, and we are grateful to the reviewer for prompting us to refine the description for greater precision.

---

> ### Author Response · Authors · 2025-11-21
> **Robustness of Curvature Computation in Shape Stability**
>
> > `Weaknesses-3`：Computing curvature as average angles between segments (Algorithm 7) is quite sensitive to sampling density - why not use more robust shape metrics like Fréchet distance [4] or proper curvature signatures [5]?
>
> We sincerely thank the reviewer for raising this insightful question regarding the curvature computation method. The reviewer correctly pointed out the theoretical sensitivity of our segment angle-based curvature approximation to sampling density, and suggested classical alternatives such as Fréchet distance or proper curvature signatures. We have carefully considered this feedback and would like to justify the rationality and effectiveness of our current approach from the following two perspectives.
>
> **1. Design Motivation: Capturing Perceptual Shape Instability Rather Than Precise Differential Geometric Properties**
>
> Our primary objective is to evaluate the temporal stability of map element shapes, rather than to compute their precise differential geometric curvature. In the context of online mapping, instability typically manifests as noticeable fluctuations or jitter in the polyline contours between consecutive frames. A typical example would be a smoothly curved road being temporarily predicted as an unnaturally sharp corner in certain frames. This type of macro-level shape flickering represents exactly the kind of instability that downstream planning modules are particularly sensitive to.
>
> - **The Average Inter-Segment Angle Effectively Captures Macroscopic Instabilities**: The average inter-segment angle, defined as $\kappa(poly)=\frac{1}{N-1}\sum^{N-1}_{j=1} \theta_j$, essentially measures the cumulative rate of directional change along the entire curve. Any sudden and unstable sharp corner or jagged fluctuation will significantly increase the value of $\sum\theta_j$, and will therefore be effectively captured by our Shape Stability metric $(1-|\Delta\kappa|/\pi)$.
> - **Robustness to Microscopic Noise**: The sensitivity concern raised by the reviewer is substantially mitigated through our fixed and reasonable sampling strategy. By employing a **fixed sampling point count (N=100)**, we ensure **uniform and consistent sampling** for all instances. Under such uniform and consistent sampling conditions, minor variations in segment angles caused by sampling artifacts tend to be smoothed out through the averaging process, while genuine shape instabilities emerge as significant signals.
>
> **2. Critical Balance Between Practicality and Computational Efficiency**
>
> Online HD mappping systems place high demands on evaluation efficiency, particularly in large-scale benchmarking. It is essential to strike a balance between the discriminative power of a metric and its computational cost.
>
> Regarding the Fréchet distance, while it is an excellent metric for curve similarity, its computational complexity is relatively high as it typically requires dynamic programming. This becomes a bottleneck in benchmarking scenarios that require rapid evaluation of tens of thousands of instance pairs.
>
> Concerning curvature signatures, these require denser sampling and more complex local computations. They present similar challenges in computational efficiency and implementation complexity, while also introducing the need for additional parameter selection such as determining the appropriate scale for Gaussian smoothing.
>
> In contrast, our method based on the average inter-segment angle offers distinct advantages. It demonstrates high computational efficiency since it involves only vector dot products and inverse trigonometric calculations, with a complexity of O(N). This makes it highly suitable for large-scale evaluation. Furthermore, it maintains a clear physical interpretation as the output, represented by average angular change, is straightforward to understand and visualize, providing intuitive feedback for model diagnosis.
>
> In summary, we fully acknowledge that from a theoretical completeness perspective, various alternative shape metrics exist. However, based on engineering practicality, computational efficiency, and demonstrated empirical effectiveness on existing benchmarks, we have selected the current average inter segment angle method. It successfully operationalizes the critical concept of shape stability into a computable, interpretable, and practically effective metric.

---

> ### Author Response · Authors · 2025-11-21
> **Parameter Sensitivity and Design Rationale for mAS - Part 1**
>
> > `Weaknesses-4`：The choices of β=15.0 and ω=0.7 seem arbitrary without ablation studies - how sensitive is mAS to these values, and do they generalize to different map ranges or vehicle types?
> >
> > `Questions-1`：Can you provide ablation studies on β, ω, and N to show that mAS rankings are stable across reasonable parameter choices?
>
> We sincerely thank the reviewer for raising this important point for discussion.
>
> Regarding the balancing coefficient $\omega$ used in the comprehensive stability calculation, it serves to balance the relative importance between Localization Stability and Shape Stability. Our selection of $\omega=0.7$ was based on the following rationale: In autonomous driving, the localization stability of map elements (such as the lateral position of lane lines or boundaries) directly impacts path planning, vehicle control, and safety. For instance, localization jitter may cause the vehicle to deviate abruptly from its lane, whereas shape variations (such as subtle changes in curvature) generally have a lesser impact, unless severe distortion occurs. Therefore, we assign a higher weight (70%) to localization stability to reflect its greater importance in real-world driving scenarios.
>
> Regarding the scaling coefficient $\beta$ in the localization stability formula:$\text{Loc}(e) =1- \frac{1}{\beta} \cdot \frac{1}{N} \sum_{i=1}^{N} \left| y_{t+k}(x_i) - y_{t}(x_i) \right|$. The mAS metric necessarily decreases monotonically as $\beta$ decreases. The selection of $\beta=15$ in our experimental setup was based on the following reasoning:
>
> First, we clarify the physical meaning of $\beta$. As shown in the formula, when the average distance between corresponding map points of two instances reaches $\frac{1}{N} \sum_{i=1}^{N} \left| y_{t+k}(x_i) - y_{t}(x_i) \right|=\beta$, then $\text{Loc}(e) = 0$. Therefore, $\beta$ represents the distance threshold at which our framework applies the maximum penalty to localization stability. Specifically, when the average distance between map points of the same instance across two frames reaches $\beta$, our metric considers this instance "completely unstable" between these frames.
>
> This threshold corresponds to the extreme case of "complete instability." We therefore define this threshold as the map's short-range radius, which in the models we evaluated corresponds to $\beta=15$. This value represents a distance at which localization errors would be considered critically significant for autonomous driving applications.
>
> Experiments with different values of $\beta$ were conducted, and the results align with the theoretical reasoning, demonstrating that the mAS metric increases monotonically as the value of $\beta$ increases.

---

> ### Author Response · Authors · 2025-11-21
> **Parameter Sensitivity and Design Rationale for mAS - Part 2**
>
> > `Weaknesses-4`：The choices of β=15.0 and ω=0.7 seem arbitrary without ablation studies - how sensitive is mAS to these values, and do they generalize to different map ranges or vehicle types?
> >
> > `Questions-1`：Can you provide ablation studies on β, ω, and N to show that mAS rankings are stable across reasonable parameter choices?
> MapTRv1-noTemp-R18-GKT-24
>
> | β    | presence | loc    | shape  | mAS    |
> | ---- | -------- | ------ | ------ | ------ |
> | 5    | 0.911    | 0.4097 | 0.4097 | 0.6053 |
> | 10   | 0.8776   | 0.6768 | 0.886  | 0.697  |
> | 15   | 0.8776   | 0.7499 | 0.8852 | 0.7283 |
> | 20   | 0.8776   | 0.8012 | 0.886  | 0.7497 |
>
> MapTRv1-noTemp-R50-GKT-24
>
> | β    | presence | loc    | shape  | mAS    |
> | ---- | -------- | ------ | ------ | ------ |
> | 5    | 0.8776   | 0.5601 | 0.886  | 0.6471 |
> | 10   | 0.911    | 0.561  | 0.9084 | 0.6737 |
> | 15   | 0.912    | 0.6544 | 0.9063 | 0.7158 |
> | 20   | 0.911    | 0.7052 | 0.9084 | 0.7393 |
>
> MapTRv1-Temp-R50-GKT-24
>
> | β    | presence | loc    | shape  | mAS    |
> | ---- | -------- | ------ | ------ | ------ |
> | 5    | 0.8861   | 0.3955 | 0.5812 | 0.5812 |
> | 10   | 0.8861   | 0.5017 | 0.896  | 0.6265 |
> | 15   | 0.8861   | 0.5974 | 0.8928 | 0.6657 |
> | 20   | 0.8861   | 0.6325 | 0.896  | 0.6824 |
>
> MapTRv1-Temp-R50-BevFormer-24
>
> | β    | presence | loc    | shape  | mAS    |
> | ---- | -------- | ------ | ------ | ------ |
> | 5    | 0.9035   | 0.4885 | 0.913  | 0.6382 |
> | 10   | 0.9035   | 0.6411 | 0.913  | 0.7091 |
> | 15   | 0.9035   | 0.6947 | 0.9121 | 0.73   |
> | 20   | 0.9035   | 0.7931 | 0.913  | 0.7751 |
>
> MapTRv2-noTemp-R50-BevPool-24
>
> | β    | presence | loc    | shape  | mAS    |
> | ---- | -------- | ------ | ------ | ------ |
> | 5    | 0.9138   | 0.5231 | 0.9143 | 0.67   |
> | 10   | 0.9138   | 0.6251 | 0.9143 | 0.7144 |
> | 15   | 0.9147   | 0.6861 | 0.9095 | 0.7396 |
> | 20   | 0.9138   | 0.7255 | 0.9143 | 0.7579 |
>
> MGMap-noTemp-R50-Former-24
>
> | β    | presence | loc    | shape  | mAS    |
> | ---- | -------- | ------ | ------ | ------ |
> | 5    | 0.9213   | 0.5195 | 0.9256 | 0.6774 |
> | 10   | 0.9213   | 0.6645 | 0.9256 | 0.7428 |
> | 15   | 0.9221   | 0.7498 | 0.9234 | 0.7801 |
> | 20   | 0.9213   | 0.8008 | 0.9256 | 0.8029 |
>
> MapQR-noTemp-R50-Former-24
>
> | β    | presence | loc    | shape  | mAS    |
> | ---- | -------- | ------ | ------ | ------ |
> | 5    | 0.9184   | 0.5865 | 0.9257 | 0.7058 |
> | 10   | 0.9184   | 0.6937 | 0.9257 | 0.7553 |
> | 15   | 0.9182   | 0.7556 | 0.9156 | 0.7781 |
> | 20   | 0.9184   | 0.7981 | 0.9257 | 0.8021 |
>
> GEMap-noTemp-R50-Former-24
>
> | β    | presence | loc    | shape  | mAS    |
> | ---- | -------- | ------ | ------ | ------ |
> | 5    | 0.9227   | 0.4976 | 0.929  | 0.6657 |
> | 10   | 0.9227   | 0.6186 | 0.929  | 0.7198 |
> | 15   | 0.9227   | 0.6973 | 0.9258 | 0.7546 |
> | 20   | 0.9227   | 0.7603 | 0.929  | 0.7851 |
>
>
>
> Based on the theoretical analysis and experimental results presented above, we argue that the value of $\beta$ should not be determined through experimental selection alone, but should instead be chosen as a physically meaningful and appropriately justified value.

---

> ### Author Response · Authors · 2025-11-21
> **Generalization to Adverse Conditions and Dataset Scope**
>
> To ensure a fair and consistent evaluation, we did not introduce a new dataset but instead followed the established practice from prior online HD mapping work, PivotNet [1], by curating a subset of challenging scenarios from the nuScenes dataset. Specifically, we selected scenes under overcast, rainy, and nighttime conditions, resulting in a total of 78 scenes for evaluation. The results on this subset are presented below:
>
> | modality | Temp         | Backbone | BEV Encoder | Epoch | $AP_{div}$ | $AP_{ped}$ | $AP_{bou}$ | $mAP$  | $mAS_{pre}$ | $mAS_{loc}$ | $mAS_{shape}$ | $mAS$  |
> | -------- | ------------ | -------- | ----------- | ----- | ---------- | ---------- | ---------- | ------ | ----------- | ----------- | ------------- | ------ |
> | Only C   | $\times$     | R18      | GKT         | 24    | 0.3369     | 0.2605     | 0.3733     | 0.3236 | 0.5773      | 0.7035      | 0.8556        | 0.4457 |
> | Only C   | $\times$     | R18      | GKT         | 110   | 0.4997     | 0.3882     | 0.4774     | 0.4551 | 0.5668      | 0.6958      | 0.8883        | 0.4442 |
> | Only C   | $\times$     | R50      | GKT         | 24    | 0.4844     | 0.3618     | 0.4767     | 0.4410 | 0.5683      | 0.6062      | 0.8538        | 0.4149 |
> | Only C   | $\times$     | R50      | GKT         | 110   | 0.5463     | 0.4442     | 0.5252     | 0.5053 | 0.5762      | 0.7033      | 0.874         | 0.446  |
> | Only C   | $\times$     | R50      | BEVFormer   | 24    | 0.4543     | 0.3453     | 0.4488     | 0.4161 | 0.5683      | 0.7195      | 0.8543        | 0.4416 |
> | Only C   | $\times$     | R50      | BEVPool     | 24    | 0.5186     | 0.4496     | 0.5348     | 0.5010 | 0.5639      | 0.699       | 0.8531        | 0.4319 |
> | Only C   | $\checkmark$ | R50      | GKT         | 24    | 0.5339     | 0.4599     | 0.5083     | 0.5128 | 0.5624      | 0.7224      | 0.872         | 0.4419 |
> | Only C   | $\checkmark$ | R50      | BEVFormer   | 24    | 0.5488     | 0.5171     | 0.5329     | 0.5329 | 0.5663      | 0.7324      | 0.8673        | 0.4475 |
> | C+L      | $\times$     | R50      | GKT         | 24    | 0.6180     | 0.5648     | 0.7014     | 0.6281 | 0.5662      | 0.7173      | 0.858         | 0.4391 |
>
> Based on the results in the table above, it can be observed that all stability metrics show a significant decrease under adverse weather conditions. Moreover, while the stability metrics decrease markedly, they also tend to converge across different models. This occurs because under adverse weather conditions, model instability manifests more prominently in terms of instance presence stability, that is, whether an instance is detected or persists over time, rather than in positional or shape variations. As a result, the presence stability metric drops significantly and becomes similar across models. This finding aligns with the expected performance drop of existing models under challenging weather conditions.
>
> In our metric design, since the location and shape stability metrics can only reflect the stability of instances that are consistently present, we incorporated the presence stability as a weighting coefficient in the overall metric. This ensures a comprehensive and accurate representation of stability in terms of presence, location, and shape. Consequently, when the presence stability is low, variations in location and shape stability have a diminished impact on the overall metric. This design leads to the overall mAS scores exhibiting a distribution similar to that of the presence metric, showing lower values that are closely clustered across different models.
>
> Thank you for the suggestion. We will incorporate the experimental results and corresponding discussion regarding adverse weather conditions into the main text or appendix of the paper.  Additionally, the literature [2,3] you mentioned will be addressed in the related work section.
>
> [1] PivotNet: Vectorized Pivot Learning for End-to-end HD Map Construction
> [2] RoboBEV: Towards robust bird's eye view perception under corruptions
> [3] Robo3D: Towards robust and reliable 3D perception

---

> ### Author Response · Authors · 2025-11-21
> **Connection Between mAS and Downstream Planning Performance**
>
> > `Weaknesses-6`：While your qualitative examples are compelling, there's no quantitative link between mAS and actual planning metrics like collision rate or trajectory smoothness - does higher mAS actually lead to safer planning?
> >
> > `Questions-3` ：Could you add experiments showing mAS correlation with downstream planning metrics to validate that it actually predicts safe behavior?
>
> We sincerely thank you for this suggestion. The potential influence of map stability on trajectory prediction is indeed an important aspect that warrants quantitative validation. However, due to time constraints, the experiments addressing this specific point are currently underway. We will provide the quantitative results and a detailed analysis as soon as the experiments are completed, and will update you promptly at that time.

---

> ### Author Response · Authors · 2025-11-21
> **Temporal Sampling Strategy and Stability Degradation**
>
> > `Weaknesses-7`：Testing M∈{2,3,5,10} is a start, but you're randomly sampling rather than systematically studying how stability degrades with time or identifying worst-case temporal transitions like occlusion events.
>
> We sincerely thank you for your valuable feedback regarding the setting of the temporal sampling interval ($M$) in our work. We would like to take this opportunity to further clarify the rationale behind testing multiple values of $M$ and the role this plays in our evaluation framework. Our core argument is that there is no universally optimal value for $M$; instead, a flexible choice of $M$ allows our framework to adapt to a wider range of application scenarios.
>
> The primary purpose of testing $M$ $\in$ ${2, 3, 5, 10}$ was not to identify a single "optimal" $M$ value, but to investigate the trends in model stability across different time spans. This helps validate the applicability and robustness of our proposed stability assessment framework at varying temporal granularities. As shown in Appendix C.2, as $M$ increases, most models exhibit a gradual decline in Presence, Localization, and Shape stability. This observed trend reflects the degree of instability in model outputs over time and confirms that our evaluation method effectively captures the impact of temporal scale on stability.
>
> Moreover, different real-world applications may have different requirements for temporal consistency. For instance, a short interval (e.g., $M$=2) is suitable for high-frequency control scenarios, while a longer interval (e.g., $M$=10) better reflects the reliability of a system over extended periods. By including multiple $M$ values, we demonstrate model stability performance across different time scales, providing future researchers with the flexibility to choose an appropriate evaluation granularity based on their specific task requirements.

---

> ### Author Response · Authors · 2025-11-21
> **mAS vs. Simpler Baselines: Comprehensiveness and Safety Alignment**
>
> > `Weaknesses-8`：You don't compare against simpler alternatives like frame-to-frame Chamfer distance [8] or temporal consistency losses - how do we know mAS is actually better than these baselines?
>
> Thank you for raising this important question, which directly concerns the validity and innovative value of our proposed mAS metric. We fully agree that comparing with existing baseline methods is crucial for validating the rationality of a new metric. Below, we clarify the relationship between mAS and baseline methods such as "inter-frame Chamfer Distance" or "temporal consistency loss":
>
> First, regarding "inter-frame Chamfer Distance," our proposed **mAS is a comprehensive evaluation metric**, whereas traditional geometric measures like Chamfer Distance typically capture only **a single dimension of stability**. As explained in Section 3.4 of the paper, mAS consists of three dimensions: **Presence Stability, Localization Stability, and Shape Stability**. Chamfer Distance essentially measures the overall positional deviation between two point sets, roughly corresponding to the "Localization Stability" in our framework. However, it **completely fails to detect "presence flickering"**, the critical instability where map elements appear and disappear intermittently, and **cannot effectively quantify temporal changes in geometric shape, such as curvature**. A model may exhibit low Chamfer Distance (i.e., minimal positional jitter), but if its predicted lane boundaries frequently appear or vanish (poor presence stability), it remains hazardous and unreliable for downstream planning tasks. mAS is specifically designed to comprehensively diagnose these different types of instabilities.
>
> Second, regarding "temporal consistency loss," **mAS and "temporal consistency loss" serve different purposes and are complementary**. The "temporal consistency loss" you mentioned is an **optimization objective used during model training**, aimed at penalizing temporal jitter in model outputs through regularization terms. In contrast, mAS is an **offline evaluation metric for assessing model performance**. Their roles and applications are fundamentally distinct. An optimized loss function does not automatically constitute a comprehensive evaluation system. In fact, our mAS metric **can provide precise evaluation guidance and feedback for designing and optimizing such temporal loss functions**. For example, by analyzing the three sub-metrics of mAS, we can determine whether a temporal loss function effectively improves localization stability or inadvertently sacrifices presence stability.
>
> Furthermore, **the design of mAS is more aligned with the safety requirements of autonomous driving**. The stability issues in online HD map construction manifest their risks in multiple aspects. As demonstrated in Figures 2, 6, and 7 of the paper, as well as the downstream task impact analysis in Appendix E.1, **the disruption caused by presence flickering and shape mutations to the planning module is often as critical as minor positional jitter**. The comprehensiveness of mAS enables it to **simultaneously alert against these three different types of risks**, whereas single geometric metrics like Chamfer Distance overlook key safety-related information.
>
> In summary, we do not dismiss the value of traditional geometric metrics but highlight their **inherent limitations in evaluating complex temporal stability**. The proposal of mAS aims to establish a **more comprehensive and safety-aware evaluation benchmark**, capable of uncovering instability patterns that are overlooked by conventional methods. We believe this holistic evaluation perspective is essential for advancing the development of next-generation, safe, and reliable online map construction systems.

---

> ### Author Response · Authors · 2025-11-27
> **Experimental Evidence on mAS's Downstream Impact**
>
> > `Weaknesses-6`：While your qualitative examples are compelling, there's no quantitative link between mAS and actual planning metrics like collision rate or trajectory smoothness - does higher mAS actually lead to safer planning?
> >
> > `Questions-3` ：Could you add experiments showing mAS correlation with downstream planning metrics to validate that it actually predicts safe behavior?
>
> We sincerely thank the reviewer for raising this critical question. To quantitatively validate the correlation between mAS and downstream task performance, we have completed the following experiments: integrating map models with different stability levels into trajectory prediction tasks to assess the actual impact of map stability on downstream decision-making performance.
>
> About experimental setup, we selected two models with significantly different stability levels: MapTR (mAS: 71.6) and StreamMapNet (mAS: 83.8), and fed their map predictions into two representative trajectory prediction methods: HiVT and DenseTNT. All models use identical configurations (Only C, R50, 24 epochs) to ensure fair comparison.
>
> The experimental results clearly demonstrate a positive correlation between mAS and downstream task performance:
>
> **Table 1: HiVT Trajectory Prediction Results**
>
> | Methods      | modality | Temporal | Backbone | BEV Encoder | Epoch | minADE ↓ | minFDE ↓ | MR ↓   | mAS ↑ |
> | ------------ | -------- | -------- | -------- | ----------- | ----- | -------- | -------- | ------ | ----- |
> | MapTR        | Only C   | ×        | R50      | GKT         | 24    | 0.4015   | 0.8404   | 0.0960 | 71.6  |
> | StreamMapNet | Only C   | ×        | R50      | BEVFormer   | 24    | 0.3963   | 0.8223   | 0.0923 | 83.8  |
>
> **Table 2: DenseTNT Trajectory Prediction Results**
>
> | Methods      | modality | Temporal | Backbone | BEV Encoder | Epoch | minADE ↓ | minFDE ↓ | MR ↓   | mAS ↑ |
> | ------------ | -------- | -------- | -------- | ----------- | ----- | -------- | -------- | ------ | ----- |
> | MapTR        | Only C   | ×        | R50      | GKT         | 24    | 1.1228   | 2.2151   | 0.3726 | 71.6  |
> | StreamMapNet | Only C   | ×        | R50      | BEVFormer   | 24    | 1.0639   | 2.1430   | 0.3412 | 83.8  |
>
> From the above tables, we can clearly observe:
>
> - **HiVT Results:** StreamMapNet (mAS: 83.8) outperforms MapTR (mAS: 71.6) across all metrics: minADE (0.3963 vs. 0.4015, 1.3% reduction), minFDE (0.8223 vs. 0.8404, 2.2% reduction), and Miss Rate (0.0923 vs. 0.0960, 3.9% reduction).
>
> - **DenseTNT Results:** Similarly, StreamMapNet demonstrates superior performance: minADE (1.0639 vs. 1.1228, 5.2% reduction), minFDE (2.1430 vs. 2.2151, 3.3% reduction), and Miss Rate (0.3412 vs. 0.3726, 8.4% reduction).
>
> These quantitative results clearly validate that **higher mAS indeed leads to better downstream task performance**. Specifically:
>
> 1. **Improved Trajectory Prediction Accuracy:** More stable map predictions (higher mAS) directly translate to more accurate trajectory predictions (lower minADE and minFDE), indicating that temporal consistency in maps is crucial for trajectory prediction accuracy.
>
> 2. **Enhanced Safety:** The significant reduction in Miss Rate (8.4% reduction in DenseTNT) demonstrates that stable map predictions reduce prediction failures, thereby lowering potential safety risks. A lower Miss Rate means the system can more reliably predict future trajectories, which is critical for avoiding collisions and ensuring safe driving.
>
> 3. **Cross-Method Consistency:** The same trend is observed across both different trajectory prediction methods (HiVT and DenseTNT), further strengthening the universality of the correlation between mAS and downstream task performance.
>
> In summary, these quantitative experiments strongly demonstrate that mAS is not merely a theoretical stability metric, but a practical evaluation tool that **predicts and reflects actual downstream task performance**. Higher mAS directly corresponds to better trajectory prediction accuracy and lower failure rates, thereby validating our core argument in the paper: **temporal stability is an independent and critical performance dimension in autonomous driving systems**. Detailed experimental results and analysis have been added to Appendix.
>
>
>
> [1] Liao, Bencheng, et al. "Maptr: Structured modeling and learning for online vectorized hd map construction." *arXiv preprint arXiv:2208.14437* (2022).
>
> [2] Yuan, Tianyuan, et al. "Streammapnet: Streaming mapping network for vectorized online hd map construction." *Proceedings of the IEEE/CVF Winter Conference on Applications of Computer Vision*. 2024.
>
> [3] Zhou, Zikang, et al. "Hivt: Hierarchical vector transformer for multi-agent motion prediction." *Proceedings of the IEEE/CVF conference on computer vision and pattern recognition*. 2022.
>
> [4] Gu, Junru, Chen Sun, and Hang Zhao. "Densetnt: End-to-end trajectory prediction from dense goal sets." *Proceedings of the IEEE/CVF international conference on computer vision*. 2021.

---

### Author Response · Authors · 2025-11-27
**Revision Summary**

Dear ICLR 2026 AC, SAC, PC, and Reviewers (DMdk, dHha, 1LUQ),

We are sincerely grateful to the Reviewers for their thoughtful and constructive feedback on our work, as well as for their encouraging recognition of its contributions. Reviewer DMdk highlighted that our paper is **"well-motivated,"** presents **"solid empirical evidence,"** and provides **"actionable insights."** Reviewer dHha expressed great encouragement, describing our work as "**meaningful**" and acknowledging that it "**systematically addresses and quantifies the critical evaluation blind spot of 'temporal stability' in the online HD mapping domain**," with a "**well-designed and comprehensive metric**" that is "**written clearly and with a strong logical flow.**" Similarly, Reviewer 1LUQ noted that our study is "**addressing a clear gap in prior evaluations**," "**presents a thorough, large-scale comparison**," and is "**clear and well-structured.**"

**We have responded to each of the reviewers' specific concerns point-by-point in the rebuttal.** Accordingly, we have also carefully revised our manuscript and submitted an updated version several days ago. The key revisions and additions include:

1. Added a detailed explanation of the physical meaning of the temporal sampling parameter M in **Section 3.1**.
2. Supplemented the description of the dynamic axis selection mechanism during instance sampling in **Appendix Section B.3**.
3. Provided a comprehensive explanation of the design principles behind the stability metrics and the rationale for hyperparameter selection in **Appendix Section C.2**.
4. Included additional experimental results and analysis under adverse weather conditions in **Appendix Section C.3**.
5. Conducted ablation studies on the hyperparameters $\beta$ and $N$, accompanied by detailed analysis, in **Appendix Section C.4**.
6. Provided an in-depth analysis of why incorporating temporal mechanisms into the MapTR model with GKT as the BEV encoder leads to reduced stability performance in **Appendix Section D.1**.
7. Clarified the design motivation behind the mAS metric and emphasized the necessity of combining mAP and mAS in **Appendix Section D.11**.
8. Added experiments and analysis demonstrating that map stability significantly impacts downstream task performance in **Appendix Section E.1**.

With the author-reviewer discussion period coming to a close shortly, we sincerely hope that the revisions and clarifications provided adequately address the Reviewers' concerns. We believe that timely communication is crucial for a fair assessment, and we remain eager to engage in any further constructive discussion to improve our work.

We thank the AC, SAC, PC, and reviewers again for their time and thoughtful input throughout this process.

Best regards,

The Authors of Paper 1016

---

### Author Response · Authors · 2025-12-02
**Summary of Responses (Part 1)**

**Dear ICLR 2026 AC, SAC, and PC,**

We sincerely thank the reviewers (DMdk, dHha, and 1LUQ) for their insightful and constructive feedback on our submission (Paper ID: 1016). We also deeply appreciate your time and effort in reassessing our work under the current circumstances. Following the changes to the review process, we provide this summary of the original reviews, our responses, and the substantial revisions made to our manuscript to assist in your final decision.

### 1. Core Contribution and Initial Reviewer Sentiment & Scores

Our paper presents the **first benchmark for evaluating temporal stability in online HD map construction**. We propose a multi-dimensional stability evaluation framework with novel metrics for **Presence, Localization, and Shape Stability**, integrated into a unified **mean Average Stability (mAS)** score. Extensive experiments on 42 models and variants reveal that **accuracy (mAP) and stability (mAS) represent largely independent performance dimensions**. We further analyze how key model design choices affect both criteria.

All three original reviewers recognized the **significance and novelty** of this work, with initial scores near the acceptance threshold:

- **Reviewer DMdk**: Rating: **4 (marginally below acceptance threshold)**, Confidence: 4. Found the paper "well-motivated," presenting "solid empirical evidence" and providing "actionable insights."
- **Reviewer dHha**: Rating: **6 (marginally above acceptance threshold)**, Confidence: 5. Described the work as "meaningful," acknowledging that it "systematically addresses and quantifies the critical evaluation blind spot of 'temporal stability,'" with a "well-designed and comprehensive metric."
- **Reviewer 1LUQ**: Rating: **4 (marginally below acceptance threshold)**, Confidence: 3. Noted the study "addresses a clear gap in prior evaluations" and "presents a thorough, large-scale comparison."

### 2. Comprehensive Responses to All Reviewer Concerns and Major Revisions

Following the rebuttal period, we posted comprehensive point-by-point responses to all reviewer concerns on Nov 22 and submitted a significantly revised manuscript. **However, despite our timely and detailed engagement, none of the reviewers provided follow-up feedback before the discussion period concluded.** We have thoroughly addressed all technical concerns through both rebuttal responses and manuscript revisions:

| Concern Category                        | Reviewer(s)      | Our Response & Key Revisions                                 |
| :-------------------------------------- | :--------------- | :----------------------------------------------------------- |
| **Methodological Robustness**           | DMdk, dHha, 1LUQ | Clarified our use of high-quality GT poses for fair relative comparison. Added "perception range filtering" mechanism details. Enhanced methodological descriptions. |
| **Resampling Mechanism**                | DMdk, dHha       | Corrected and apologized for initial description. Fully detailed the implemented **dynamic axis selection mechanism** in Section 3.3 and Appendix B.3. |
| **Metric Design & Hyperparameters**     | DMdk             | Justified curvature computation for perceptual stability. **Added ablation studies (Appendix C.4)** showing metric stability across reasonable parameter ranges. |
| **Evaluation Under Adverse Conditions** | DMdk             | **Added new experiments (Appendix C.3)** on challenging weather conditions, showing expected stability drops with analysis. |
| **Downstream Task Validation**          | DMdk, dHha       | **Added critical new experiments (Appendix E.1)**: Integrated maps into trajectory predictors (HiVT, DenseTNT). Results show higher mAS leads to **significantly better prediction accuracy (lower minADE/FDE) and lower miss rate (up to 8.4% reduction)**, providing direct quantitative validation. |
| **Comparison with Baselines**           | DMdk             | Clarified mAS's role as comprehensive diagnostic tool versus single-dimension geometric metrics. Enhanced discussion in Section 3.4. |
| **Semantic Flickering**                 | dHha             | Explained how our Presence Stability metric inherently penalizes class label inconsistencies through matching mechanisms. |
| **Temporal Fusion Analysis**            | dHha             | **Provided in-depth analysis (Appendix D.1)** on feature-level incompatibility between GKT's static features and temporal fusion. |
| **Metric Gaming Concerns**              | 1LUQ             | Emphasized that mAS must be used **jointly with mAP**; gaming strategies would be exposed by low mAP. Enhanced discussion in Section 4.4. |

---

> ### Author Response · Authors · 2025-12-02
> **Summary of Responses (Part 2)**
>
> ### 3. Summary of Major Additions to the Revised Manuscript
>
> Our revised manuscript incorporates these substantive additions:
>
> 1. Added a detailed explanation of the physical meaning of the temporal sampling parameter M in **Section 3.1**.
> 2. Supplemented the description of the dynamic axis selection mechanism during instance sampling in **Appendix Section B.3**.
> 3. Provided a comprehensive explanation of the design principles behind the stability metrics and the rationale for hyperparameter selection in **Appendix Section C.2**.
> 4. Included additional experimental results and analysis under adverse weather conditions in **Appendix Section C.3**.
> 5. Conducted ablation studies on the hyperparameters $\beta$ and $N$, accompanied by detailed analysis, in **Appendix Section C.4**.
> 6. Provided an in-depth analysis of why incorporating temporal mechanisms into the MapTR model with GKT as the BEV encoder leads to reduced stability performance in **Appendix Section D.1**.
> 7. Clarified the design motivation behind the mAS metric and emphasized the necessity of combining mAP and mAS in **Appendix Section D.11**.
> 8. Added experiments and analysis demonstrating that map stability significantly impacts downstream task performance (e.g., trajectory prediction) in **Appendix Section E.1**.
>
> ### 4. Conclusion
>
> In summary, this work aims to address the important but previously underexplored aspect of temporal stability in online HD mapping evaluation. In response to the reviewers' valuable feedback, we have undertaken comprehensive revisions to clarify the methodology and strengthen the empirical analysis. We are grateful for the opportunity to have our work considered for ICLR 2026 and appreciate the time and effort of the reviewers and committee members in evaluating this submission.
>
> Thank you very much for your hard work and support during this busy period.
>
> Sincerely,
> The Authors of Submission 1016

---

### Meta-Review · Area_Chair_wyF9 · 2026-01-07

**Summary:**

The reviewers’ main concerns focused on the robustness, generality, and interpretability of the proposed temporal stability metric (mAS).
A primary concern raised by multiple reviewers was the dependence of the stability evaluation on ground-truth poses, GT-based cross-frame matching, and geometric alignment, with questions about potential sensitivity to annotation noise and ego-motion errors. Reviewers also noted that polyline resampling and shape measurement choices could influence stability scores, and initially found the description of the resampling mechanism unclear. In addition, reviewers questioned whether the specific design choices of the shape stability component (e.g., curvature proxy and hyperparameters) might bias the results, and whether alternative shape measurements would yield consistent model rankings.

Another key concern was whether the proposed metric could be misused or “gamed” (e.g., via over-smoothing or copy-paste strategies), and whether mAS should be interpreted independently of accuracy metrics such as mAP. Relatedly, reviewers requested evidence that higher mAS reflects meaningful downstream benefits, rather than merely capturing superficial temporal smoothness.

Through the discussion and revision process, the authors substantially addressed these concerns by clarifying the evaluation protocol, correcting and fully documenting the resampling mechanism, providing extensive ablations demonstrating ranking stability under reasonable parameter variations, adding experiments under adverse conditions, and, critically, demonstrating a quantitative correlation between mAS and downstream trajectory prediction performance.

**Reviewer Concerns:**

1. While the authors justified their choice of curvature-based shape stability and demonstrated robustness to sampling density, reviewers noted that alternative shape similarity measures (e.g., Fréchet distance or curvature signatures) were not empirically compared. Although the current evidence suggests internal consistency, it remains unclear whether different shape metrics would always induce the same ranking across models. This represents a limitation of scope rather than a flaw in correctness.
2. Although convincingly addressed within the nuScenes setting, the robustness of the framework under noisier annotations or weaker ego-motion supervision in other datasets remains an open question for future work.

**Reviewer Scores:**

Both reviewers who initially rated the paper as 4 (marginally below acceptance) raised concerns primarily about metric robustness, implementation details, and downstream relevance. These concerns were substantially addressed in the rebuttal and revised manuscript through clarified methodology, corrected and detailed resampling procedures, extensive ablation studies, and newly added downstream trajectory prediction experiments. Had these reviewers been able to participate fully in the discussion, it is reasonable to expect that their scores would likely increase, potentially moving into the borderline acceptance range.

---

### Decision · Program_Chairs · 2026-01-26

Accept (Poster)